# AcE: Attribution-Controlled Knowledge Editing for Multi-hop Factual Recall

**Jiayu YANG** [*]
HKUST(GZ)
Deep Interdisciplinary Intelligence Lab
jyang729@connect.hkust-gz.edu.cn

**Yuxuan FAN** [*]
HKUST(GZ)
yfan546@connect.hkust-gz.edu.cn

**Songning LAI** [*]
HKUST(GZ)
Deep Interdisciplinary Intelligence Lab
songninglai@hkust-gz.edu.cn

**Shengen WU**
HKUST(GZ)
Deep Interdisciplinary Intelligence Lab
wu.shengen@outlook.com

**Jiaqi TANG**
HKUST
jtang092@connect.hkust-gz.edu.cn

**Chun KANG**
BUAA
kicoforstudy@gmail.com

**Zhijiang GUO** [†]
HKUST(GZ)
HKUST
zhijiangguo@hkust-gz.edu.cn

**Yutao YUE** [†]
HKUST(GZ)
Institute of Deep Perception Technology, JITRI
Deep Interdisciplinary Intelligence Lab
yutaoyue@hkust-gz.edu.cn

## Abstract

Large Language Models (LLMs) require efficient knowledge editing (KE) to update factual information, yet existing methods exhibit significant performance decay in multi-hop factual recall. This failure is particularly acute when edits involve intermediate implicit subjects within reasoning chains. Through causal analysis, we reveal that this limitation stems from an oversight of how chained knowledge is dynamically represented and utilized at the neuron level. We discover that during multi-hop reasoning, implicit subjects function as "query neurons", which sequentially activate corresponding "value neurons" across transformer layers to accumulate information toward the final answer—a dynamic prior KE work has overlooked. Guided by this insight, we propose **AcE** (**A**ttribution-**C**ontrolled Knowledge **E**diting), a framework that leverages neuron-level attribution to identify and edit these critical query-value (Q-V) pathways. AcE provides a mechanistically grounded solution for multi-hop KE, empirically outperforming state-of-the-art methods by **9.44%** on GPT-J and **37.46%** on Qwen3-8B. Our analysis further reveals more fine-grained activation patterns in Qwen3 and demonstrates that the semantic interpretability of value neurons is orchestrated by query-driven accumulation. These findings establish a new pathway for advancing KE capabilities based on the principled understanding of internal reasoning mechanisms.

## 1 Introduction

Large Language Models (LLMs) have demonstrated remarkable capabilities in storing and retrieving vast amounts of factual knowledge (Hu et al., 2024; Mousavi et al., 2025), underpinning their success in diverse downstream tasks such as question answering and reasoning (Mamaghan et al.,

---

[*]Equal contribution

[†]Correspondence to Zhijiang GUO and Yutao YUE {zhijiangguo@hkust-gz.edu.cn},{yutaoyue@hkust-gz.edu.cn}

2024; Ke et al., 2025). However, the knowledge encapsulated within these models is static and can become outdated or incorrect, necessitating mechanisms for efficient updates. Full retraining is computationally prohibitive, motivating the development of Knowledge Editing (KE) techniques, which aim to modify specific factual associations within LLMs cost-effectively and with minimal impact on unrelated knowledge (Yao et al., 2023; Mazzia et al., 2024).

While the locate-then-edit paradigm, exemplified by methods like ROME (Meng et al., 2022a) and MEMIT (Meng et al., 2022b), has proven effective for editing facts in LMs by successfully targeting components like Feed-Forward Networks (FFNs), its efficacy significantly diminishes when confronting multi-hop factual recall (Zhong et al., 2023; Zhang et al., 2024b). This challenge is particularly acute when the edited knowledge involves an implicit subject – an intermediate entity in a reasoning chain that leads to the final answer's explicit subject (Zhong et al., 2023). As illustrated in Figure 1, a multi-hop query like "*What country is Mark Trumbo's sport originates from?*" requires the model to react as a chain: starting with the initial subject "*Mark Trumbo*", it must first identify

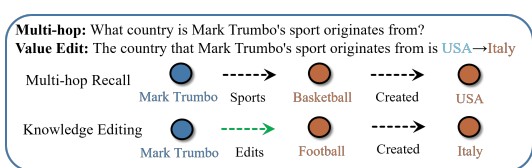

Figure 1: Illustration of a multi-hop factual recall. A multi-hop query requires traversing multiple facts. The diagram shows the original knowledge path (e.g., *Mark Trumbo → Basketball → USA*) and how a knowledge edit (green arrow) can target an intermediate fact, which then requires the model to follow a potentially new chain (*Mark Trumbo → Football → Italy*). The intermediate entity "*Football*" serves as the implicit subject.

his "sport" (the implicit subject, e.g., "*Basketball*" in the original knowledge) and then recall the country where that sport originated (the explicit subject, e.g., "*USA*"). Figure 1 depicts editing the knowledge so that Mark Trumbo's sport becomes "*Football*", which is then linked to originating from "*Italy*". Standard single-hop editing methods, often focusing on deeper FFN layers (Meng et al., 2022a;b; Li et al., 2024). Although recent work like IFMET (Zhang et al., 2024b) has demonstrated improved multi-hop performance, particularly for implicit subjects, by editing deeper FFN layers via constructing multi-hop prompts, the underlying mechanisms explaining why these deeper edits are crucial for correctly retrieving and utilizing the implicit subject information remain unexplored.

A key limitation of existing KE methods in multi-hop scenarios stems from an incomplete understanding of how knowledge—particularly intermediate reasoning steps—is dynamically represented and accessed at the neuron level. Through systematic causal analysis, we discover that successful multi-hop recall relies on coordinated interactions between neurons across layers, where implicit subjects trigger cascading activations that progressively accumulate information through query-value interactions. Our experiments reveal two critical properties: **(i)** In multi-hop factual recall, intermediate implicit subjects function as query neurons, sequentially accumulating and activating relevant value neurons for subsequent reasoning hops. **(ii)** LLMs store semantically analogous knowledge in structurally similar transformer components, with query and value neurons for specific knowledge types exhibiting consistent localization patterns across layers. Building upon these two initial properties, we systematically analyze the mechanisms of multi-hop reasoning through experiments in Section 4. These properties provide clear answers to two fundamental questions: *"How LLMs Store the Semantics Knowledge?"* and *"How is the information accumulated?"*.

These mechanistic insights motivate AcE (**A**ttribution-**C**ontrolled Knowledge **E**diting), a framework that moves from layer-level heuristics to neuron-level interventions. As shown in Figure 4, AcE employs novel attribution techniques to identify and edit critical query-value pathways. Extensive experiments demonstrate that AcE outperforms the state-of-the-art method PMET by 9.44% on GPT-J and 37.46% on Qwen3-8B in terms of multi-hop accuracy. Ablation studies confirm the necessity of both components: skipping query layers causes a 16.51% performance drop, while omitting value layers leads to a more severe 40.45% decrease. Our analysis reveals that existing KE methods fail in multi-hop reasoning by overlooking both deeper value layers and critical query-layer activation patterns. We further discover distinct architectural differences: while GPT-J maintains fixed layer separation, Qwen3-8B exhibits dynamic, domain-specific alignment. Crucially, correct predictions depend on sparse interpretable neurons—ablating just 27 critical neurons causes an accuracy drop to 3.2%, demonstrating the neural coordination required for multi-hop reasoning.

## 2 RELATED WORK

**Knowledge Editing and Multi-hop Reasoning in LLMs.**  To avoid the high cost of retraining, recent work focuses on efficient knowledge editing by directly modifying model weights (Zhang et al., 2024a; Yao et al., 2023). The locate-then-edit paradigm identifies key parameters storing target facts, with methods like ROME Meng et al. (2022a) and MEMIT Meng et al. (2022b) updating FFN weights via causal tracing and closed-form optimization, while PMET Li et al. (2024) distinguishes MHSA for general patterns and FFNs for factual content. Other approaches include tuning small subsets (Mitchell et al., 2021) or using hypernetworks (Gupta et al., 2024). However, these methods face significant challenges in multi-hop reasoning scenarios, where editing intermediate facts can break reasoning chains (Yao et al., 2023; Cohen et al., 2024). The MQuAKE benchmark (Zhong et al., 2023) highlights the failure of existing methods to propagate such edits effectively. While IFMET (Zhang et al., 2024b) advances multi-hop performance through layer-level interventions, ACE addresses these limitations by introducing a neuron-level mechanism that leverages activation dynamics for robust multi-hop edits.

**Mechanistic Interpretability and Knowledge Localization.**  Effective knowledge editing depends on understanding how LLMs store information. FFN layers have been shown to act as key-value stores, with values encoding semantic content (Geva et al., 2020), and recent work has localized factual knowledge at finer granularity (Dai et al., 2021; Hernandez et al., 2023). Yu & Ananiadou (2023) revealed that predictions are driven by interactions between query and value neurons, with value neurons exhibiting consistent relational semantics. Building on this, ACE employs a neuron-level approach to trace and modulate these activation pathways, enabling targeted and interpretable editing in multi-hop reasoning.

## 3 PRELIMINARIES

### 3.1 MODELING AND TASK

**Definition of Neurons in LLMs.**  Autoregressive decoder-only LLMs $\mathcal{F}_\theta$ process input $\mathbf{x}$ into tokens $X = [t_1, \ldots, t_T]$, predicting next-token distributions over $V$. Tokens are embedded via $E$ to $h_i^0$, then processed through $L$ transformer layers (MHSA + FFN). Layer $l$ hidden states are:

$$h_i^l = h_i^{l-1} + A_i^l + F_i^l, \tag{1}$$

where $h_i^{l-1}, A_i^l, F_i^l$ mean the previous layer's output, the attention output, and the FFN output. Finally, the last position's $Lth$ layer output is used to compute the probability distribution $y$ with unembedding matrix $E_u \in \mathbb{R}^{B \times d}$:

$$y = \text{softmax}(E_u h_T^L). \tag{2}$$

The attention layer outputs a weighted sum across $H$ heads, while the FFN applies a nonlinear activation $\sigma$ to two linear transformations.

$$A_i^l = \sum_{j=1}^{H} f_{ATTN}{}_j^l(h_1^{l-1}, h_2^{l-1}, ..., h_T^{l-1}), \tag{3}$$

$$F_i^l = W_{fc2}^l \sigma(W_{fc1}^l(h_i^{l-1} + A_i^l)), \tag{4}$$

where $W_{fc1}^l \in \mathbb{R}^{N \times d}$ and $W_{fc2}^l \in \mathbb{R}^{d \times N}$ are two matrices. Geva et al. (2020) finds that FFN output can be expressed as a weighted sum of FFN neurons:

$$F_i^l = \sum_{k=1}^{N} m_{i,k}^l \cdot fc2_k^l, \tag{5}$$

$$m_{i,k}^l = \sigma(fc1_k^l \cdot (h_i^{l-1} + A_i^l)). \tag{6}$$

Following the same notation as in (Yu & Ananiadou, 2023), $fc2_k^l$ is the subvalue of FFN which is the column $kth$ of $W_{fc2}^l$, and its coefficient score $m_{i,k}^l$ is calculated by residual output $h_i^{l-1} + A_i^l$ and FFN subkey $fc1_k^l$, the $k-th$ row of $W_{fc1}^l$. Similarly, the attention output $A_i^l$ can be represented as:

$$A_i^l = \sum_{j=1}^{H} \sum_{p=1}^{T} \alpha_{i,j,p}^l W_{j,l}^o(W_{j,l}^v h_p^{l-1}), \tag{7}$$

$$\alpha^l_{i,j,p} = \text{softmax}(W^q_{j,l} h^{l-1}_i \cdot W^k_{j,l} h^{l-1}_p), \tag{8}$$

where $W^q_{j,l}, W^k_{j,l}, W^v_{j,l}, W^o_{j,l}$ are the query, key, value and output matrices. As discussed in Eq. 5-6, the $kth$ **FFN neuron** is the $kth$ subvalue $fc2^l_k$ and its corresponding subkey $fc1^l_k$, including all the query and value neurons mentioned later. Similar to FFN neurons, we regard the $kth$ column of $W^o_{j,l}$ as the $kth$ attention subvalue, whose subkey is the $kth$ row of $W^v_{j,l}$.

**Factual Recall Tasks.** Define knowledge set $\mathcal{K} = \{(s, r, o)\} \subseteq \mathcal{E} \times \mathcal{R} \times \mathcal{E}$, where $\mathcal{E}$ (entities) and $\mathcal{R}$ (relations) form triplets $(s, r, o)$ indicating subject $s$ relates to object $o$ via $r$. An edit instance $e = (s, r, o \rightarrow o^*)$ represents replacing $o$ with $o^*$.

We evaluate model $\mathcal{M}$ through factual recall tasks, which assess its ability to answer both single-hop and multi-hop factual questions $Q$ requiring $\geq 1$ reasoning steps. A reasoning chain is formally defined as $\mathcal{C} = (s_1, r_1, o_1) \oplus \cdots \oplus (s_n, r_n, o_n)$, starting from an explicit subject $s_1$ and concluding with the target answer $o_n$. To illustrate, consider the two-hop question: "What country is Mark Trumbo's sport originates from?" This corresponds to the knowledge chain $(Mark\ Trumbo, Sport, Basketball) \oplus (Basketball, Created, USA)$. We employ two question formats in our evaluation: **Cloze Format** ($Q_{cloze}$): "The country that Mark Trumbo's sport originates from is ____" and **QA Format** ($Q_{qa}$): "What country is Mark Trumbo's sport originates from?".

The multi-hop recall process consists of two distinct phases: the *explicit recall step* $(s_1, r_1, o_1)$ to retrieve the initial fact, followed by *implicit recall steps* $\{(s_i, r_i, o_i)\}^n_{i=2}$ to traverse subsequent hops. A factual recall is considered successful if the model generates the correct final answer, i.e., $\mathcal{M}(Q_{cloze}) = o_n$ or $\mathcal{M}(Q_{qa}) = o_n$. Figure 1 shows the overall process of the task. Factual Recall Tasks evaluate if the post-edited model can utilize updated knowledge for multi-hop reasoning. Given an edit $e = (s, r, o \rightarrow o^*)$, edit prompt $T_e$, and fact chain $\mathcal{C}_e$ containing $(s, r, o)$, the model must answer multi-hop queries using the updated fact $(s, r, o^*)$. For example: After editing $(Mark\ Trumbo, Sport, Basketball \rightarrow Football)$, the multi query "The country that Mark Trumbo's sport originates from is" should shift from *USA* to *Italy*.

## 3.2 ATTRIBUTION METRICS

**Distribution Change.** The final hidden state $h^L_T$ aggregates critical information for token prediction through summation of neuron-level vectors, implying that essential predictive signals are encoded in specific neurons. By decomposing $h^l_T$ as $h^l_T = v + x$ (where $v$ denotes a target neuron's contribution and $x = h^l_T - v$ represents residual components), we quantify $v$'s causal influence via probability distribution shift $\Delta p(w) = p(w|x+v) - p(w|x)$ for token $w$. This framework enables systematic identification of neuronal components that maximally amplify $\Delta p(w)$, establishing a methodology for pivotal neurons in a static way, measuring the importance level of neurons.

**Importance Score for Value Neurons and Layers.** To jointly account for both the variable neuron $v$ and the conditioning variable $x$ in the probabilistic framework. Based on Yu & Ananiadou (2023), we find log probability increase could efficiently evaluate the model's distribution change by a neuron. Define the log probability increase as importance score $\mathcal{I}$ for vectors, which satisfies $\mathcal{I}(x+v) \approx \mathcal{I}(x) + \mathcal{I}(v)$. If $v^l$ is a vector in $lth$ attention layer, the importance score of $v^l$ is:

$$\mathcal{I}(v^l) = \log\big(p(w \mid v^l + h^{l-1})\big) - \log\big(p(w \mid h^{l-1})\big), \mathcal{I}(l) = \sum_{v \in l} \mathcal{I}(v^l), \tag{9}$$

where the probability is computed from vocabulary in Eq.2, layer $l$ denotes the index set of varying neuron $v$. When vector $v^l$ in $lth$ FFN layer, $\mathcal{I}$ is computed by replacing $h^{l-1}$ as $h^{l-1} + A^l$.

**Importance Score for Query Neurons and Layers.** We find that query neurons exist in the transformer while solving multi-hop tasks aims to activate value neurons, even if they do not directly contain information about the target token $w$. We use the inner product between its subkey and itself to measure the importance of the query neuron, showing the ability activating value neurons:

$$\mathcal{I}_{query} = v \cdot fc1^l_k, \mathcal{I}_{query}(l) = \sum_{v \in l} \mathcal{I}(v^l), \tag{10}$$

where layer $l$ denotes the index set of varying neuron $v$, since the $fc2$ vectors do not change, the coefficient scores will be the only varying element. Therefore, if a query neuron exhibits a larger inner product with the subkey, it activates the value neurons more.

# 4 MECHANISM OF MULTI-HOP REASONING

How do language models store and retrieve knowledge when performing complex multi-hop reasoning? We begin by examining how semantically related knowledge is organized within transformer components (Section 4.1), revealing consistent patterns that challenge conventional wisdom about knowledge storage. Building on these structural insights, we then trace how information propagates through reasoning chains (Section 4.2), uncovering a sophisticated query-value activation mechanism that progressively accumulates evidence toward final answers. Our analysis spans both GPT-J (Wang & Komatsuzaki, 2021) and the more advanced Qwen3-8B (Team, 2025), with the latter exhibiting even more fine-grained activation patterns that offer new insights into reasoning dynamics.

## 4.1 HOW LLMS STORE THE SEMANTICS KNOWLEDGE?

A fundamental question underpinning effective knowledge editing is: *How do LLMs internally store knowledge?* Addressing this question is crucial for clarifying the models' reasoning logic in multi-hop scenarios and enhancing their internal interpretability. By systematically investigating knowledge representation mechanisms, we can uncover how information propagates through transformer architectures, thereby identifying the root causes of existing KE methods' failures in multi-hop reasoning. This understanding provides the foundation for developing more effective KE techniques.

To this end, we use dataset MQuAKE-3K (Zhong et al., 2023) to explore the mechanism of semantics knowledge storage. MQuAKE is a challenging knowledge editing benchmark which comprises over 3000 multi-hop edit instances. Considering the challenging factual recall queries, we extract a subset from original dataset by systematically evaluating the vanilla GPT-J model then scale to Qwen3-8B, using all single-hop factual recall queries to characterize its inherent knowledge retention capabilities. This controlled experimentation protocol isolates the model's capacity to store the knowledge. Both attention and FFN layers exhibit inherent capabilities for knowledge storage. We use GPT-4o (Hurst et al., 2024) to classify the knowledge types in the dataset, and conducting analysis across eight semantic categories: Nationality (NN), Continent (CT), Language (LG), Capital (CP), Leadership (LS), Author (AT), Sports Team (ST), and Company Founder (CF).

Table 1: Top 9 important attention layers (left block) and FFN layers (right block) in GPT-J.

| | **Top 9 important attention layers and FFN layers** | | | | | | | | | | | | | | | | | |
|---|---|---|---|---|---|---|---|---|---|---|---|---|---|---|---|---|---|---|
| NN | $a_{27}$ | $a_{26}$ | $a_7$ | $a_{10}$ | $a_9$ | $a_{25}$ | $a_8$ | $a_{11}$ | $a_5$ | $f_{20}$ | $f_{24}$ | $f_{16}$ | $f_{18}$ | $f_{15}$ | $f_{22}$ | $f_{23}$ | $f_{26}$ | $f_{25}$ |
| CT | $a_{27}$ | $a_{26}$ | $a_7$ | $a_{10}$ | $a_8$ | $a_5$ | $a_9$ | $a_{11}$ | $a_{25}$ | $f_{22}$ | $f_{24}$ | $f_{16}$ | $f_{21}$ | $f_{15}$ | $f_{17}$ | $f_{26}$ | $f_{25}$ | $f_{23}$ |
| LG | $a_{27}$ | $a_7$ | $a_5$ | $a_6$ | $a_8$ | $a_4$ | $a_1$ | $a_{26}$ | $a_9$ | $f_{27}$ | $f_7$ | $f_5$ | $f_6$ | $f_8$ | $f_4$ | $f_1$ | $f_{26}$ | $f_9$ |
| CP | $a_{27}$ | $a_{26}$ | $a_7$ | $a_{10}$ | $a_8$ | $a_9$ | $a_{12}$ | $a_5$ | $a_6$ | $f_{27}$ | $f_{26}$ | $f_7$ | $f_{10}$ | $f_{12}$ | $f_8$ | $f_9$ | $f_5$ | $f_6$ |
| LS | $a_{27}$ | $a_{26}$ | $a_{25}$ | $a_7$ | $a_{10}$ | $a_6$ | $a_8$ | $a_6$ | $a_5$ | $f_{27}$ | $f_{26}$ | $f_{25}$ | $f_{24}$ | $f_7$ | $f_{10}$ | $f_6$ | $f_8$ | $f_5$ |
| AT | $a_{27}$ | $a_{26}$ | $a_{25}$ | $a_7$ | $a_9$ | $a_8$ | $a_{10}$ | $a_5$ | $a_6$ | $f_7$ | $f_9$ | $f_8$ | $f_{10}$ | $f_{27}$ | $f_{26}$ | $f_{25}$ | $f_5$ | $f_6$ |
| ST | $a_{26}$ | $a_{27}$ | $a_7$ | $a_{10}$ | $a_8$ | $a_6$ | $a_5$ | $a_{25}$ | $a_4$ | $f_{26}$ | $f_{27}$ | $f_7$ | $f_{10}$ | $f_8$ | $f_6$ | $f_5$ | $f_{25}$ | $f_4$ |
| CF | $a_{26}$ | $a_{27}$ | $a_{24}$ | $a_{25}$ | $a_7$ | $a_8$ | $a_9$ | $a_6$ | $a_5$ | $f_{26}$ | $f_{27}$ | $f_{24}$ | $f_{25}$ | $f_7$ | $f_8$ | $f_9$ | $f_6$ | $f_5$ |

To find critical neurons and layers, we performed forward passes on all single-hop questions in the dataset and computed the sum of the importance scores at last position in the residual stream. We select top 9 most important layers to analysis the knowledge attribution as Table 1. Our analysis reveals that information with similar semantics tends to be stored in proximate neural modules, while semantically unrelated information is distributed across disparate modules. Specifically, MHSA components are activated in similar positions, for instance, $a_{27}, a_{26}, a_7$ ranks top in all knowledge. This suggests that **MHSA stores general knowledge and capabilities in LLMs. Differently, FFN layers tends to primarily extracts its own knowledge**. For instance, $f_{24}, f_{27}, f_{16}$ ranks top for similar semantics (NN, CT, LG, CP, LS), dissimilar semantics (AT, ST, CF) resides in distinct layers.

We set the top critical semantic-related neurons to zero in GPT-J and Qwen3-8B, Figure 2 shows that only 1% intervention on important neurons causes over 90% accuracy decrease in semantic-related subsets. Compared to 1% random intervention, only 9.47% and 8.19% accuracy decrease in GPT-J and Qwen3-8B. Based on our observations, we formalize **Takeaway 1: LLMs tend to store semantically analogous knowledge in structurally similar components.**

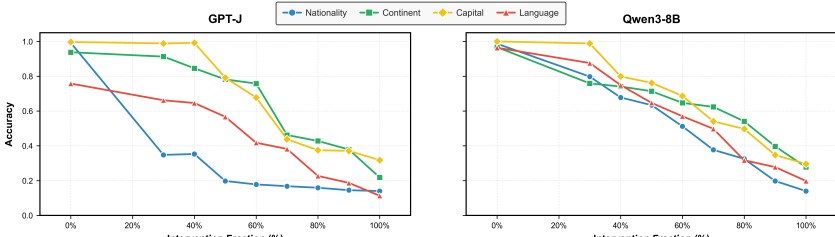

Figure 2: The Impact of Causal Intervention with semantic-related requests upon LLMs of most important layer, including Nationality, Continent, Capital and Language requests.

## 4.2 How is the Information Accumulated?

Building on Takeaway 1, which identifies the locations and distribution patterns of knowledge storage in LLMs, we now investigate the dynamic process of information accumulation during multi-hop reasoning. Specifically, we aim to address the question: *How is the information pertaining to the final answer accumulated through implicit subjects?* To this end, we conduct causal interventions and statistical analyses on the critical layers identified previously, examining the interactions between query and value neurons across the reasoning chain.

Building on the observation that implicit subject information accumulates through shallow FFN layers (Zhang et al., 2024b), we investigate the mechanistic details of this accumulation process. While Hou et al. (2023) suggests that models process multi-hop reasoning by segmenting it into individual single-hop recalls, the specific neural mechanisms underlying information integration remain unclear. This gap leads us to examine two fundamental questions: *How is information pertaining to the final answer progressively accumulated through implicit subjects? Through which transformer components is this cumulative effect primarily mediated?*

Our investigation begins with an analysis of value neuron distributions in GPT-J's FFN layers. Contrary to the prevailing assumption that deeper layers predominantly determine model outputs, we observe a more complex pattern. Value neurons exhibit peak density in middle-to-deeper layers, with distribution maxima not aligning with the final residual stream positions. Most strikingly, we find an abrupt depletion of neurons in the deepest layers—**a finding that challenges conventional understanding**. Further analysis reveals a systematic relationship between query and value neuron activation patterns. As shown in Figure 3, the significance variation of query FFN layers closely tracks the

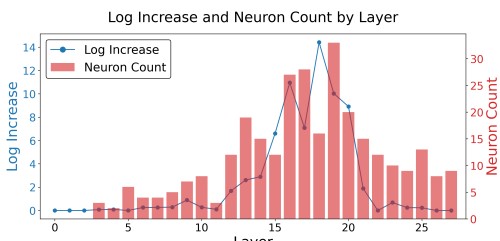

Figure 3: Query layers' log increase and value neurons count by layers in GPT-J. Layer log increase is the importance score calculated by logarithmic difference in Eq. 9.

progressive accumulation of implicit subject information during reasoning. Layer-wise probing demonstrates that query neuron activation (blue curve) consistently precedes value neuron activation (red histogram) by 1-2 layers, indicating a structured information flow mechanism.

To validate the functional importance of these patterns, we conducted targeted interventions on query neurons. When we ablated the top 100 query neurons from two layers showing peak activation for 2-hop requests ($f_{q16}$ and $f_{q18}$), model capability decreased by 46.2% and 61.9%, respectively. Moreover, the number of activated value neurons in subsequent layers ($f_{17,18,19}$) dropped dramatically from (28,16,33) to (6,4,7), while other value neurons showed minimal change (12 neurons decreased). We show more details on Qwen3 while we explore the forward processes in Appendix G.

These experimental results lead us to two key observations: (1) Final answer information is progressively encoded through early-stage query-value activation pairs throughout the reasoning chain; (2) Implicit subjects functionally operate as query neurons that orchestrate the activation of value neurons for subsequent reasoning steps. Based on these consistent experimental findings, we formulate

**Takeaway 2: Information of the final answer is accumulated through implicit query neurons that sequentially activate corresponding value neurons across the reasoning chain.**

## 5 ACE: ATTRIBUTION-CONTROLLED KNOWLEDGE EDITING

Based on our findings and validations concerning the mechanisms of LLMs' knowledge storage and reasoning in multi-hop factual recall tasks, we introduce **A**ttribution-**C**ontrolled Knowledge **E**diting (ACE) for Multi-hop Factual Recall. As Figure 4 demonstrated, ACE extends the established locate-then-edit paradigm through three sequential operations: **first** identifying latent query neurons to verify critical query FFN layers that activate explicit subjects' value neurons; **second**, applying model editing via multi-hop prompts to modify explicit subject knowledge by targeting FFN value components in deeper layers; and third, executing complementary edits targeting FFN query mechanisms in middle-to-shallow layers to adjust the implicit reasoning path originating from the updated explicit fact. ACE strategically emphasizes the significance of FFN query mechanisms for successful multi-hop reasoning while also properly considering the FFN values within the residual stream computation.

**Stage 1: Identifying.** Using the definition of neurons and importance in Section 3, we employ importance score $\mathcal{I}, \mathcal{I}_{query}$ in Eq. 9 and 10 to identify critical q/v neurons and their corresponding layers. In the identifying process, we performed forward passes on all multi-hop questions in the dataset and computed the sum of the importance scores at the last token position in the residual stream. After the identifying, we rank the query and value layers and select the top layers to edit.

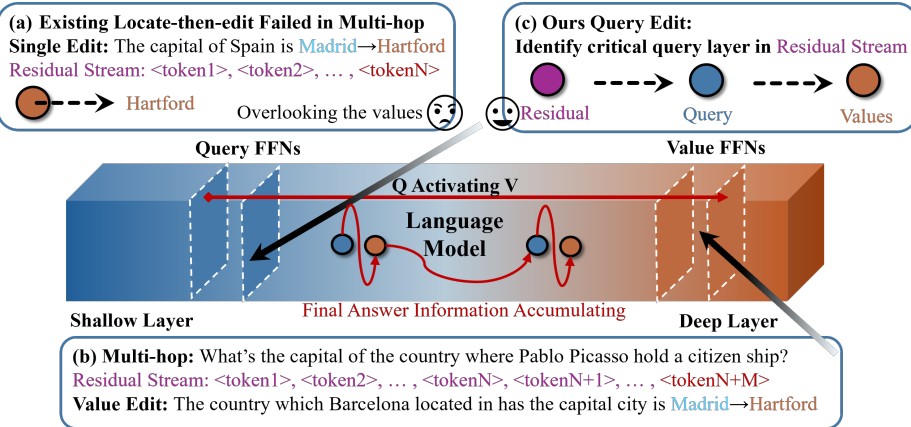

Figure 4: ACE edits Q-V neurons via attribution: (a) The existing locate-then-edit KE method updates new fact using a single-hop prompt; (b) For multi-hop factual recall tasks, traditional locate-then-edit failed to correct edit the knowledge on query layers, overlooking value neurons; (c) Our **ACE** identifies critical query layers which activates the value neurons most to edit the knowledge.

**Stage 2: Locate-then-edit.** Building upon identified critical layers, we conducted sequential knowledge editing to demonstrate the model's enhanced knowledge acquisition through intensified activation patterns. Based on previous locate-then-edit paradigm (Li et al., 2024), we apply editing on FFNs and keep attention heads unchanged, while the general semantic information saved in attention heads should not be changed. For $l$th layer FFN, it's output of the $i$th token 's would be $W_{fc2}^l \sigma(W_{fc2}^l h_i^{l-1})$, and there's no attention input in GPT-J, where $\sigma$ is the non-linear activation function. $\sigma(W_{fc2}^l h_i^{l-1})$ take the responsibility as keys, denoted as $k_i$. Whole FFN output is the value $v_i = W_{fc2}^l \sigma(W_{fc2}^l h_i^{l-1})$, whose subvalue matrix $W_{fc2}^l$ denotes the weight of the model's values needs to be modified. So we aim to modify subvalue matrix $W_{fc2}^l$ s.t. $W_{fc2}^l k = v^*$, where $v^*$ represents the new values (factual knowledge). We use backbone PMET (Li et al., 2024) to complete editing process, details shown in Algorithm D.

The ACE framework accomplishes knowledge editing through two sequential stages. In Stage 1, we identify critical query layers and value layers within the model architecture. The critical query and value layers represent the precise locations where target knowledge is stored. In Stage 2, we edit these components enables efficient integration of new factual information into the model's parameters. In all, ACE incorporates complementary edits to the often-overlooked query layers, ensuring that the

Table 2: Multi-hop accuracy comparison of different KE methods on the MQuAKE-3K dataset in a few-shot setting, Base shows the model's performance on the unedited answers and edited model's performance on edited answers. Our model outperformances than other models significantly.

| Editor | Avg.(GPT/Qwen) | GPT-J / # Edits = | | | | Qwen3-8B / # Edits = | | | |
|---|---|---|---|---|---|---|---|---|---|
| | | 1-edit | 2-edit | 3-edit | 4-edit | 1-edit | 2-edit | 3-edit | 4-edit |
| Base | 98.42 / 99.17 | 99.7 | 95.48 | 97.51 | 97.23 | 99.81 | 97.46 | 98.14 | 97.64 |
| FT | 3.54 / 2.18 | 4.17 | 2.63 | 0.00 | 0.00 | 3.14 | 2.79 | 0.00 | 0.00 |
| ROME | 35.04 / 28.79 | 44.51 | 38.93 | 17.52 | 5.06 | 35.09 | 32.48 | 18.97 | 7.08 |
| MEMIT | 38.58 / 18.67 | 64.30 | 16.87 | 17.25 | 8.16 | 29.84 | 19.18 | 12.91 | 4.20 |
| PMET | 37.01 / 20.78 | 49.26 | 36.30 | 24.34 | 17.01 | 28.64 | 14.08 | 12.56 | 11.20 |
| **ACE** | **46.45 / 58.24** | 45.26 | **50.24** | **36.17** | **43.29** | **60.22** | **59.48** | **51.62** | **47.61** |

updated knowledge can be properly activated and traversed during multi-step reasoning processes, making information accumulates progressively via query-value interactions.

## 6 EXPERIMENTS

### 6.1 EXPERIMENTAL SETUP

**Dataset.** MQuAKE-3K (Zhong et al., 2023) is a benchmark dataset for evaluating large language models' multi-hop fact recall capability after knowledge editing. Each instance contains a multi-hop fact recall chain (interconnected triples) and corresponding textual questions, requiring models to maintain coherent reasoning after applying single-hop knowledge edits via edit prompts. This design simulates cascading effects of real-world knowledge updates, providing a systematic framework to assess models' dynamic knowledge management. To faithfully evaluate the model's capabilities and assess its capacity to assimilate edited knowledge, we applied filtering to the original dataset, resulting in a curated subset where the base model consistently achieves accurate reasoning.

**Baselines.** Baselines are **Base**, refers to the original GPT-J (6B) and Qwen3-8B without any edit; **FT** refers to fine-tuning method, **ROME** Meng et al. (2022a) refers to the vanilla locote-then-edit method; **MEMIT** Meng et al. (2022b), extends ROME with updating weights by a set of facts, and **PMET** Li et al. (2024) claims optimizations on FFN layers based on MEMIT.

**Metric and Setup.** We apply our ACE on the model GPT-J (6B) and Qwen3-8B. We use Multi-hop requests answering accuracy as the metric to evaluate the performance of our edited model. We use PMET as our model's primary backbone for editing. More settings details see in Appendix H.

### 6.2 MAIN RESULTS

Table 2 demonstrates the general performance of various established methods and **ACE** under the different classes of data settings on our subset of MQuAKE-3K. To enable precise identification of critical layers store the knowledge and activation patterns during model editing, we systematically integrated in-context priming and Chain-of-Thought reasoning across all editing prompts. This dual-strategy architecture ensures optimal knowledge editing efficacy through targeted activation within the model's parameter space. **# Edits** refers to the number of how many individual facts in the reasoning chain are edits, its maximum equals to the number of hops in the dataset. As evidenced in Table 2, the proposed **ACE** framework demonstrates consistent superiority over existing methodologies across various evaluation metrics. Our method outperformances $9.44\%$ and $37.46\%$ on GPT-J and Qwen3-8B. Traditional Locate-then Edit paradigm performances even worse on Qwen3-8B due to the fixed editing positions, and ACE shows much flexibility on reasoning models due to the unfixed q-v positions as discussed in Section 6.4.

We also show detailed metrics of our experiments in Table 3. **Efficacy** metric, measures whether the model can successfully answer the single-hop fact recall prompt, **Paraphrase** metric, measures the model can answer the same original question in different statements. **Specificity** metric, measures the whether the edit of a specific fact affects other facts stored within the model. ACE demonstrates superior performance over SOTA across multiple metrics, highlighting its multiple capabilities.

Table 3: The detailed results of more metrics in experiments on GPT-J and Qwen3-8B.

| Editor | Efficacy | Paraphrase | Specificity |
|---|---|---|---|
| FT (GPT-J/Qwen3) | 98.4 / 97.1 | 74.5 / 73.2 | **83.8** / 79.6 |
| ROME (GPT-J/Qwen3) | 64.2 / 51.8 | 61.6 / 49.3 | 66.8 / 57.2 |
| MEMIT (GPT-J/Qwen3) | 62.8 / 53.6 | 66.2 / 61.8 | 70.0 / 64.7 |
| PMET (GPT-J/Qwen3) | 81.6 / 75.6 | 65.8 / 68.9 | 74.6 / 64.4 |
| ACE (GPT-J/Qwen3) | **99.8 / 99.4** | **91.2 / 94.2** | 79.2 / **81.8** |

Table 4: The results of ablation experiments on GPT-J-6B and Qwen3-8B model. The column **Editor** shows which layer(s) are **skipped** in the editing process, the index **#** of the layer refers to the importance rank. The percentage of decrease($\downarrow$) is calculated relative to ACE as the baseline.

| Editor | Avg. | Efficacy | Paraphrase | Specificity |
|---|---|---|---|---|
| $f_q^{\#1}$ (GPT-J-6B) | 43.26($\downarrow 6.87\%$) | 96.2 $\downarrow$ | 90.4 $\downarrow$ | 77.6 $\downarrow$ |
| $f_q^{\#1,2}$ (GPT-J-6B) | 41.19($\downarrow 11.32\%$) | 94.8 $\downarrow$ | 91.0 $\downarrow$ | 75.2 $\downarrow$ |
| $f_q^{\#1,2,3}$ (GPT-J-6B) | 38.78($\downarrow 16.51\%$) | 90.6 $\downarrow$ | 91.6 $\uparrow$ | 74.3 $\downarrow$ |
| $f_v^{\#1}$ (GPT-J-6B) | 42.14($\downarrow 9.28\%$) | 91.5 $\downarrow$ | 88.7 $\downarrow$ | 74.9 $\downarrow$ |
| $f_v^{\#1,2}$ (GPT-J-6B) | 33.97($\downarrow 26.87\%$) | 84.6 $\downarrow$ | 81.8 $\downarrow$ | 70.3 $\downarrow$ |
| $f_q^{\#1}$ (Qwen3-8B) | 52.61($\downarrow 9.67\%$) | 96.6 $\downarrow$ | 91.3 $\downarrow$ | 74.3 $\downarrow$ |
| $f_q^{\#1,2}$ (Qwen3-8B) | 47.34($\downarrow 18.71\%$) | 92.4 $\downarrow$ | 89.3 $\downarrow$ | 72.7 $\downarrow$ |
| $f_q^{\#1,2,3}$ (Qwen3-8B) | 45.26($\downarrow 22.29\%$) | 91.3 $\downarrow$ | 85.8 $\downarrow$ | 71.5 $\downarrow$ |
| $f_v^{\#1}$ (Qwen3-8B) | 51.39($\downarrow 11.76\%$) | 94.8 $\downarrow$ | 90.2 $\downarrow$ | 73.6 $\downarrow$ |
| $f_v^{\#1,2}$ (Qwen3-8B) | 34.68($\downarrow 40.45\%$) | 71.9 $\downarrow$ | 73.1 $\downarrow$ | 72.3 $\downarrow$ |

## 6.3 ABLATION STUDY

**Edited Layer.** We performed an ablation study on the edited query and value FFNs in ACE by sequentially skipping edits to the identified critical layers within the model, aiming to elucidate the impact of these critical layers within the ACE editor. As shown in Table 4, for both GPT-J-6B and Qwen3-8B, skipping the layers targeted for editing significantly impaired the editor's performance. After skipping the three most important query layers, model performance decreased by 16.51%, while skipping the two most important value layers led to a performance drop of 40.45%.

The performance degradation resulting from skipping query layers was slightly less pronounced than that from skipping value layers, which validates our takeaways: query layers transmit information by activating value layers. Skipping query layers results in incomplete activation of the corresponding value layers, whereas skipping the editing of value layers leads to less knowledge being incorporated.

**Few-Shots Prompts.** Selecting appropriate Few-Shot prompts for in-context learning within the locate-then-edit paradigm enables the model to learn from challenging multi-hop reasoning editing examples, thereby improving its ability to answer questions accurately. We conducted ablative experiments on the prompts employed in our study. We conducted Zero-Shot, One-Shot and OOD Few-Shots (no overlap between evaluation set and prompts) prompts when evaluating.

As shown in Table 5, even with less information, the performance degradation is only 9.4% (Zero-Shot, GPT-J), 5.7% (One-Shot, GPT-J), 0.4% (OOD Few-Shots, GPT-J), 9.5% (Zero-Shot, Qwen3), 3.1% (One-Shot, Qwen3), and 0.6% (OOD Few-Shots, Qwen3) compared to the original. This demonstrates that ACE maintains strong robustness under reduced in-context learning information, providing evidence that the effectiveness of knowledge editing is inherent rather than reliant on in-context learning.

## 6.4 ANALYSIS

Now we can explain why the existing KE methods failed in multi-hop reasoning. Existing KE methods overlooked the value layers in deeper locations, even regarded the output generation layers as the value layers, where the knowledge truly be stored in latter ones (details in Appendix G). And the worst is, these KE methods ignored the importance of editing the query layers.

Table 5: The results of ablation experiments on different CoT Prompts upon GPT-J and Qwen3.

| Editor | Avg. |
|---|---|
| GPT-J (Zero-Shot) | 42.08 |
| GPT-J (One-Shot) | 43.79 |
| GPT-J (OOD Few-Shots) | 46.27 |
| Qwen3 (Zero-Shot) | 53.19 |
| Qwen3 (One-Shot) | 56.46 |
| Qwen3 (OOD Few-Shots) | 57.92 |

**Fine-Grained Activation Pattern.** In Table 4, Qwen3-8B exhibits greater sensitivity to the number of layers being edited compared to GPT-J. Qwen3-8B demonstrates more fine-grained activation patterns during the forward pass, which imposes stricter requirements on the coherence of information within the reasoning chain.In GPT-J, the active query layers are predominantly located in the middle layers, while the deeper FFN value layers are activated by queries. There exists a consistent layer-wise separation between these two families of layers, and their positions $(f_{q16,q17,q18}, f_{v28,v29,v30})$ remain invariant across different domains. In contrast, in Qwen3-8B, the query layers are situated in middle-to-deeper layers and are closely aligned—and at times partially overlapping—with the value layers they activate (e.g., $f_{q27,q28,q29}, f_{v30,v31,v32}$). Moreover, the absolute positions of these query and value layers are not statically fixed, they shift dynamically depending on the knowledge domain.

**Case Study.** As a case study, we examined the residual stream for the single-hop request: "Tim Duncan plays the sport of". The model exhibited the largest increase in importance at tokens where semantics converge, such as "plays" and "of", with reaching 0.9932 and 0.8469, respectively. The top predicted tokens were highly interpretable (e.g., "basketball", "ball", "NBA"). Meanwhile, at other transitional tokens, the model demonstrated stronger exploratory capabilities by predicting incoherent tokens. We select 27 neurons whose associated vocabulary included the correct target token and performed 1000 sampling trials with these neurons removed. The model's accuracy under this condition dropped to merely 3.2%, indicating the crucial role these interpretable neurons play in generating correct responses. In contrast, when we ablated 27 neurons with high importance scores but lacking correct semantic interpretability, the model maintained 59.4% accuracy. This suggests the final generation of correct output tokens depends on a sparse set of highly specialized, interpretable neurons.We also observe some neurons serve as q-v shared neurons at the same time, which are highly interpretable, the details of this residual stream see in Appendix F.

We consider that this alternating pattern of semantic divergence and convergence during in-text token prediction constitutes a more fine-grained activation behavior, which is modulated by the knowledge domain. Furthermore, these semantically convergent tokens, and neurons which serve as shared neurons also are aligned in recent RL research concerning token entropy (Wang et al., 2025), represents a promising focus for future studies of critical tokens in RL.

# 7 CONCLUSION

In this study, we systematically investigate the q-v activation mechanism underlying multi-hop factual recall in LMs. Through extensive experiments on both GPT-J and Qwen3-8B, our analysis reveals a sophisticated neural coordination pattern: query neurons—whether representing implicit subjects or components of the final answer—orchestrate the sequential activation of semantically interpretable value neurons throughout the reasoning chain. This mechanistic understanding resolves long-standing questions about how information propagates in multi-hop scenarios. Our work contributes to the broader understanding of how LLMs organize and process knowledge. These insights open new avenues for developing more interpretable LMs.

## ACKNOWLEDGEMENT

This work was supported by Guangzhou-HKUST(GZ) Joint Funding Program(Grant No.2023A03J0008), Education Bureau of Guangzhou Municipality. This work was supported by Jiangsu Industrial Technology Research Institute (JITRI) and Wuxi National High-Tech District (WND).

REPRODUCIBILITY STATEMENT

We place strong emphasis on the transparency and reproducibility of our work. We have uploaded the codes we used for reproducibility in the Supplementary Material. To facilitate independent verification, the complete implementation has been provided in the supplementary materials, allowing readers to directly reproduce the reported experiments. In addition, Section 6 and Appendix C, I of the main text outlines the experimental pipeline, including dataset preparation, model configurations, prompts we used and training procedures. For further clarity, Appendix H documents the full set of hyperparameter choices and auxiliary details. Together, these resources ensure that our results can be reliably replicated and extended in future research.

ETHICS STATEMENT

This work complies with the ICLR Code of Ethics. All authors of this work have committed to its adherence. The datasets used in this study are publicly available benchmarks. Our research does not involve any private or sensitive personal data. The code developed for experiments will be made publicly available to ensure reproducibility. We have followed standard practices in the field to ensure the fairness and reproducibility of our experiments. Efforts have been made to mitigate potential biases in the evaluation process.

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

APPENDIX CONTENTS

## A USE OF LARGE LANGUAGE MODELS

During manuscript preparation, a large language model (LLM) was occasionally employed as an auxiliary assistant to refine language expression, such as improving sentence fluency and enhancing readability. The model was not involved in generating original research contributions: it did not participate in formulating research questions, designing methodologies, conducting experiments, analyzing results, or drafting substantive scientific content. All core intellectual work, including the development of ideas, execution of experiments, and interpretation of findings, was carried out independently by the authors. Any linguistic suggestions offered by the LLM were critically reviewed and selectively incorporated, ensuring that accuracy, originality, and scholarly integrity were fully maintained. The authors alone bear responsibility for the research content and conclusions, and the LLM is not listed as a contributor or author.

## B RELATED WORK IN DETAILS

**Knowledge Editing in LLMs.** The inherent difficulty and computational expense of retraining LLMs to incorporate new or corrected information (Zhang et al., 2024a; Yao et al., 2023) have spurred significant research in KE. The dominant paradigm, locate-then-edit, aims to identify specific model parameters responsible for storing a target fact and modify them precisely. Seminal works like ROME Meng et al. (2022a) employed causal tracing to locate critical states in FFN layers and updated their weights via rank-one modifications. MEMIT Meng et al. (2022b) extended this by efficiently editing thousands of facts simultaneously using a closed-form optimization objective. Building on these, PMET Li et al. (2024) proposed that Multi-Head Self-Attention (MHSA) layers encode general knowledge extraction patterns, while FFNs store specific factual details, thus focusing updates on FFNs. Other approaches include fine-tuning a small set of parameters (Mitchell et al., 2021) or training external hypernetworks to predict parameter updates (Gupta et al., 2024). While these methods have shown strong performance in single-hop factual editing, their effectiveness often declines in multi-hop scenarios, as they overlook how knowledge is chained and how intermediate reasoning steps are utilized. Our work, AcE, directly addresses this limitation by focusing on neuron-level mechanisms underlying multi-hop knowledge retrieval.

**Mechanistic Interpretability and Knowledge Localization.** Effective knowledge editing is intrinsically linked to understanding where and how LLMs store knowledge. A growing body of work in mechanistic interpretability aims to unravel these internal workings. Geva et al. (2020) and Geva et al. (2020) identified FFN layers as key-value memories, where keys correspond to input patterns and values represent output distributions. More pertinent to our approach, Dai et al. (2021) and Hernandez et al. (2023) explored techniques to locate factual knowledge at a finer granularity.

The work by Yu & Ananiadou (2023) provides a crucial foundation for our methodology. They demonstrated that both attention and FFN layers operate via query neurons that activate specific value neurons to produce final predictions. These value neurons were found to be semantically interpretable when projected into the vocabulary space, and similar types of relations (e.g., "capital of X," "birthplace of Y") were shown to activate neurons in structurally consistent locations across different subjects. While IFMET focused on attribution and interpretation, AcE leverages these neuron-level insights to develop a more targeted and mechanistically grounded KE strategy, particularly for the complex dynamics of multi-hop reasoning where intermediate implicit subjects act as activators.

**Multi-hop Reasoning and Knowledge Editing.** Multi-hop reasoning, requires LLMs to chain multiple pieces of information to arrive at an answer, poses a significant challenge for KE (Yao et al., 2023; Cohen et al., 2024). Editing a single fact involved in a multi-hop chain can inadvertently disrupt the model's ability to perform the entire reasoning chain. The MQuAKE benchmark Zhong et al. (2023) highlighted the poor performance of existing KE methods on multi-hop requests, revealing that edits often fail to propagate effectively when the edited fact is an intermediate step.

AcE advances IFMET (Zhang et al., 2024b) by proposing a neuron-level editing mechanism: While IFMET observed that multi-hop editing via deeper FFN layers enhances performance, it lacked mechanistic insights into why deeper layers matter. Through query-value activation dynamics analysis, we reveal that deeper layers host value neurons processing multi-hop reasoning triggered by implicit subjects (acting as query-like activators) resolved in earlier layers. Unlike IFMET's

layer-level heuristic approach, AcE establishes an interpretable neuron-editing framework guided by attribution analysis — it achieves robust multi-hop knowledge updates by precisely identifying and modulating query-value activation pathways. This shift from layer-centric to neuron-centric mechanism interpretation constitutes our core innovation.

## C    SUBSET OF MQUAKE DATASET

### C.1    SUBSET CONSTRUCTION

This investigation into the cognitive mechanisms underlying single-hop and multi-hop fact retrieval employed a controlled experimental paradigm using cloze-style query templates. Our methodology involved curating knowledge triples from MQuAKE that were demonstrably answerable by GPT-J-6B in zero-shot configurations. This rigorous selection protocol achieved dual objectives: (1) establishing a baseline for knowledge recall under maximally constrained conditions, and (2) systematically mitigating potential confounding effects from response ambiguity in experimental outcomes.

### C.2    DATASET LABELING

We use GPT-4o to label the knowledge in the dataset. We show the prompts we used in Appendix I.3. We labeled the dataset related to the knowledge, which are Geographic location, Organization, Personal attributes, Sports, Entertainment, Language and culture, Education, Religion, Literature and Event, which consists of more detailed smaller classes.

## D    AcE OPTIMIZATION

**Input:**  Requested edits $E = \{(s_i, r_i, o_i \rightarrow o_i^*)\}_{i=1}^N$,
model $\mathcal{M}$,
all layers $l_{all}$,
value layers $l_v$
**Output:**  Modified model $\mathcal{M}_E$ containing edits from $E$
**for** $(s_i, r_i, o_i^*) \in E$ **do**
    Generate the single edit prompt $T_{r_i}(s_i)$
    Optimize $v_i^* \leftarrow \text{Search}(T_{r_i}(s_i))$ ;
    ;                                                    // $v^*$ for every new knowledge
**end**
**for** $l \in l_v$ **do**
    $\Delta^l \leftarrow \text{Calculate}([v_1^*, \dots, v_N^*])$ ;                        // edit value layers
    $W^l \leftarrow W^l + \Delta^l$ ;
    ;                                                    // update new weights
**end**
**for** $l \in l_{all}$ **do**
    $l_q \leftarrow \text{Search in residual}([l_1, \dots, l_L], l_v)$ ;        // Find critical query layers
**end**
**for** $l \in l_q$ **do**
    $\Delta^l \leftarrow \text{Calculate}([v_1^*, \dots, v_N^*])$ ;                        // edit query layers
    $W^l \leftarrow W^l + \Delta^l$ ;
    ;                                                    // update new weights
**end**

**Algorithm 1:** AcE Algorithm

Our method primarily consists of a first edit (value neurons edit) and a furtherance edit (query neurons edit). Each single edit process obtains target weights through optimization of the knowledge preservation and editing objective:

$$\underset{\hat{W}}{\arg\min} \left( \lambda \underbrace{\|\hat{W}K_0 - W_{fc2}^l K_0\|^2}_{\text{Preservation}} + \underbrace{\|\hat{W}K_E - V_E\|^2}_{\text{Editing}} \right), \tag{11}$$

where $K_0 = [k_0^1 \mid k_0^2 \mid \cdots \mid k_0^N]$ and $V_0 = W_{fc2}^l K_0$ encapsulate preserved knowledge, $K_E = [k_e^1 \mid k_e^2 \mid \cdots \mid k_e^E]$ represents edit matrices, and $V_e = [v_{e_1}^* \mid \ldots \mid v_{e_E}^*]$ denotes target representations for new knowledge. The edited fact set corresponds to $\{(s_i, r_i, o_i^*) \mid i = 1, 2, \cdots, E\}$.

Following the parameterization $\hat{W} = W_{fc2}^l + \Delta$, the closed-form solution for incremental weights is derived as:

$$\Delta = RK_E^\top (C_0 + K_E K_E^\top)^{-1}, \quad R := (V_E - W_{fc2}^l K_E), \quad C_0 := K_0 K_0^\top. \tag{12}$$

The optimization of value vector perturbations $\delta$ follows:

$$\delta = \underset{\delta}{\arg\min} \mathcal{L}(\delta) = \mu D_{\text{KL}} \left( P_{\mathcal{M}_e}[t' \mid T] \,\|\, P_{\mathcal{M}}[t' \mid T] \right) + \varphi \frac{1}{P} \sum_{j=1}^{P} - \log \mathbb{P}_{\mathcal{M}_e}[o^* \mid \text{pref}_j \oplus T_e], \tag{13}$$

where $T$ denotes KL prompts (e.g., "$s\_is\_a$"), $t'$ excludes answer tokens $o^*$, and $T_e$ represents editing prompts (e.g., "The capital of Spain is ").

## E  KNOWLEDGE BOTTLENECK

In all knowledge editing frameworks based on the locate-then-edit paradigm, the number of critical layers constitutes a crucial parameter. However, extensively editing a large number of model layers proves infeasible and often encounters a knowledge bottleneck, which is particularly pronounced in methods focusing solely on value neurons editing. Therefore, we conduct a sensitivity analysis on the number of edited layers for AcE and other baselines, demonstrating that query neurons editing is both highly efficient and necessary. Since editing query and value neurons incurs equivalent latency in AcE, we begin our comparison of methods' sensitivity to the knowledge bottleneck with an AcE variant restricted to skipping editing a single query layer, and then we extend the skipping numbers.

As shown in Table 6, we compare the performance of various editors under the condition of an equal number of edited layers, where "#N - editor" denotes the quantity of layers edited by each respective editor. It can be observed that a pronounced knowledge bottleneck exists across all editors except AcE. This manifests specifically as follows: even when significantly increasing the number of edited layers compared to the original baselines, editors focusing solely on value neurons exhibit only marginal performance gains. This phenomenon can be interpreted as evidence of an editing bottleneck inherent to value neurons, wherein excessive modifications fail to enhance the model's reasoning capability and may even prove detrimental. These results underscore the critical importance of editing query neurons.

## F  DETAILS OF INTERNAL FINE-GRAINED REASONING Q-V PAIRS FLOW IN QWEN3-8B

We examined the residual stream for the single-hop request: "Tim Duncan plays the sport of". We show the importance score increase with forward process here. In table 7, we identify distinct patterns of semantic divergence and convergence in the residual stream. At the tokens "plays" and "of", the model exhibits a stronger tendency to conclude the current prediction, narrowing the semantic flow to end the sentence, while demonstrating high interpretability. These tokens also correspond to the largest increase in importance scores. In contrast, at other token positions, the model exhibits greater potential for exploration and reasoning, favoring semantic divergence and losing almost all interpretability. We hypothesize that this intra-sentence semantic activation pattern is a capability

Table 6: Analysis of number of edited layers on GPT-J and Qwen3-8B.

| Editor | Avg. | Editor | Avg. |
|---|---|---|---|
| #9 - ROME (GPT-J) | 37.98 | #8 - ROME (Qwen3-8B) | 30.16 |
| #8 - ROME (GPT-J) | 37.56 | #7 - ROME (Qwen3-8B) | 29.72 |
| #7 - ROME (GPT-J) | 37.27 | #6 - ROME (Qwen3-8B) | 29.52 |
| #9 - MEMIT (GPT-J) | 39.47 | #8 - MEMIT (Qwen3-8B) | 21.95 |
| #8 - MEMIT (GPT-J) | 39.14 | #7 - MEMIT (Qwen3-8B) | 21.47 |
| #7 - MEMIT (GPT-J) | 38.98 | #6 - MEMIT (Qwen3-8B) | 20.16 |
| #9 - PMET (GPT-J) | 38.29 | #8 - PMET (Qwen3-8B) | 21.14 |
| #8 - PMET (GPT-J) | 38.16 | #7 - PMET (Qwen3-8B) | 21.08 |
| #7 - PMET (GPT-J) | 38.10 | #6 - PMET (Qwen3-8B) | 20.92 |
| #9 - AcE (GPT-J) | 46.45 | #8 - AcE (Qwen3-8B) | 58.24 |
| #8 - AcE (GPT-J) | 45.28 | #7 - AcE (Qwen3-8B) | 57.19 |
| #7 - AcE (GPT-J) | 43.58 | #6 - AcE (Qwen3-8B) | 54.10 |

conferred by post-training reinforcement learning, enabling the model to rapidly predict correct tokens at critical positions while maintaining strong exploratory behavior elsewhere. In studies related to token entropy in RL, such highly interpretable and semantically convergent regions are likely to play a decisive role during training.

Table 8 demonstrates the neurons' interpretability in vocabulary space. In query neurons, we could not find much interpretabilities after projecting neurons, most logits are unrelated to the request, but much interpretable neurons appears in the value neurons. Based on this observation, we could claim that the query layers enhance knowledge editing not through direct modification of knowledge representations or token embeddings, but by amplifying activations in value neurons through q-v pairs.

Table 7: Token Increase in Qwen3-8B on Residual Stream.

| Token | Importance increase | Top tokens in vocabulary space |
|---|---|---|
| Tim | FFN: 0.0021, attn: 0.0014 | 'an', 'os', 'ise', 'Exactly', 'R', 'ore', 'at', 'rot' |
| Duncan | FFN: 0.0014, attn: 0.0009 | 'era', 'allen', 'stad', 'oret', 'hit', '-led' |
| plays | FFN: 0.9932, attn: 0.8167 | **'basketball', 'NBA', 'career', 'ball'**, '-playing' |
| the | FFN: 0.4894, attn: 0.4159 | 'epit', 'inaugural', 'bidding', 'dream', 'etr' |
| sport | FFN: 0.1478, attn: 0.0948 | 'tennis', 'of', 'ful', 'arena', 'basketball', 'ball' |
| of | FFN: 0.8469, attn: 0.6198 | **'basketball', 'NBA', 'balls', 'ball'**, 'Olympia' |

Table 8: Interpretable neurons in vocabulary space, **bold** refers to the interpretable tokens.

| Neurons | Top tokens in vocabulary space |
|---|---|
| $f_{15} - 5495$ (query neuron) | outwe, expries, LESS, retaliate, $<$, Himself, ALSO |
| $f_{18} - 3584$ (value neuron) | **football**, **sports**, **players**, **soccer player**, **baseball**, **sport** |
| $f_{31} - 2097$(shared neuron in Qwen3) | **basketball**, **sports**, **balls**, **players soccer**, **baseball**, **NBA** |

# G    MORE DETAILS OF THE ANALYSIS

This section provides additional experimental details supporting the analysis in Section 4.2, using the query "Tim Duncan plays the sport of" as a case study. We analyze the attribution patterns across both FFN and attention layers throughout the forward pass. Figure 5 presents the layer-level log increase across all layers, revealing that in Qwen3-8B, activated layers are concentrated in deeper regions of

the network. Notably, the activation levels in the final layers remain relatively low, indicating that knowledge is **not** primarily stored in these terminal layers. Instead, these layers appear primarily responsible for generating model outputs rather than knowledge storage.

A key observation is the rapid decrease in attention log increase around layer 25, followed by a corresponding drop in FFN layer 27. This pattern supports the conclusion that attention mechanisms (and their associated query-value activations) facilitate factual recall by activating progressively deeper FFN layers. The attention heatmap in Figure 6 further corroborates this finding, with darker coloration indicating higher importance scores. These results demonstrate how knowledge accumulation begins in shallower attention layers and culminates in complete activation within deeper network regions.

We observe that during the forward pass of the query "Tim Duncan plays the sport of" in Qwen3-8B, unlike GPT-J which distributes query-value activation across shallower layers, Qwen3-8B executes this process continuously within deeper layers. As previously analyzed, this pattern arises from Qwen3-8B's more fine-grained intra-sentence activation behavior, where active query layers are positioned immediately preceding their corresponding value layers, which typically reside in deeper network regions.

Table 9 shows the top value neurons details in Qwen3-8B while processing this forward case. We select four neurons, representing different important position in the model. Our analysis reveals a clear distinction in the interpretability of neurons across different layers. Neurons in shallower and less critical regions typically exhibit minimal semantic interpretability, whereas those with the highest importance scores demonstrate strong correspondence to meaningful vocabulary tokens. To quantify the functional significance of these interpretable neurons, we conducted a systematic ablation study.

We selected 27 neurons whose associated vocabulary included the correct target token and performed 1000 sampling trials with these neurons removed. The model's accuracy under this condition dropped to merely 3.2%, indicating the crucial role these interpretable neurons play in generating correct responses. In contrast, when we ablated 27 neurons with high importance scores but lacking correct semantic interpretability, the model maintained 59.4% accuracy.

This striking disparity suggests that while the reasoning chain involves dense information propagation and accumulation across numerous neurons throughout the network, the final generation of correct output tokens depends on a sparse set of highly specialized, interpretable neurons. These findings have important implications for future work on token entropy in reinforcement learning, particularly regarding how models allocate computational resources during reasoning processes.

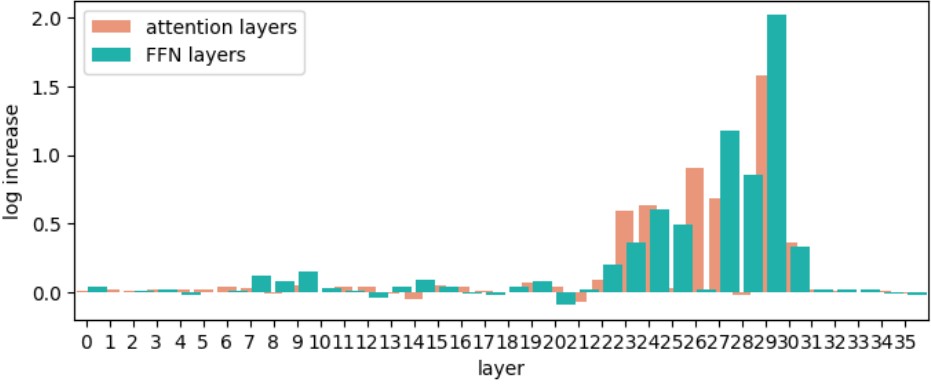

Figure 5: The layer-level log increase through all layer upon one case on Qwen3-8B.

## H EXPERIMENTAL SETTINGS

The critical layers for GPT-J-6B and Qwen3-8B have been identified as $\mathcal{R}_q = \{3, 4, 5, 6, 7, 8\}, \mathcal{R}_v = \{26, 27, 28\}$ and $\mathcal{R}_q = \{25, 26, 27\}, \mathcal{R}_v = \{28, 29, 30, 31, 32\}$. Therefore, we mainly update the FFNs components of these critical layers of GPT-J and Qwen3-8B.

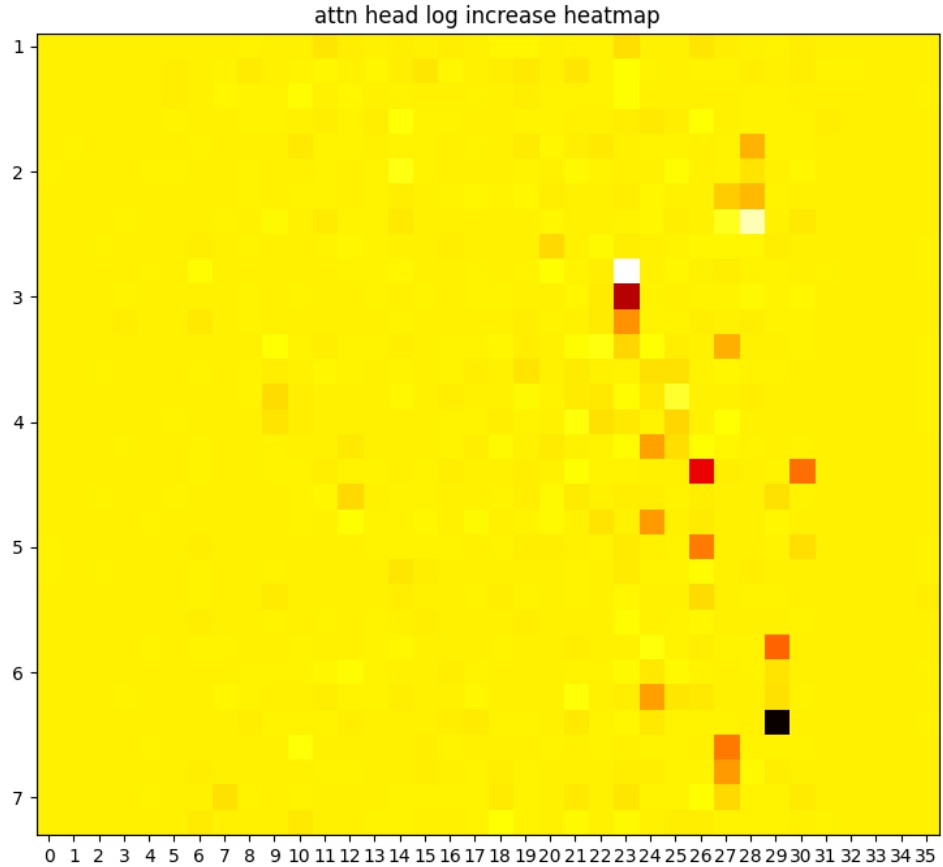

Figure 6: The attenton head heatmap through all layer upon one case on Qwen3-8B.

Table 9: FFN Value Neuron Increase and Vocabulary in Qwen3-8B on Residual Stream.

| Neuron | Importance increase | Top tokens in vocabulary space |
|---|---|---|
| $f_{v29} - 5709$ | 1.1542 | 'basketball', 'baskets', 'Basket', 'Baskets', 'ball', '-BASKET' |
| $f_{v27} - 4542$ | 0.4716 | 'fire', 'licer', 'phu', 'shutdown', 'IAM', 'arez' |
| $f_{v29} - 7550$ | 0.4421 | 'information', 'INFORMATION', '_information', 'informação' |
| $f_{v28} - 8055$ | 0.1866 | 'school', 'sniff', 'originals', 'baseball', 'balls' |

In our edits, our configuration for PMET adheres to the settings specified by (Li et al., 2024). Initially, we set $\varphi = 1$ and $0 \leq \mu \leq 1$ to manage the retention of the model's original knowledge. As $\mu$ increases, the retention level also increases, while $\varphi$ exhibits the opposite trend. After maximizing the probability of the target knowledge, we reduce $\varphi$ to 0.1 to preserve the original knowledge as much as possible. Optimization is halted when $D_{KL} < 0.01$. On GPT-J and Qwen3-8B, for estimating the covariance matrix (i.e., the set of previously memorized keys $C_0$), we sample 10,0000 times on Wikitext in fp32 precision and set $\lambda = 6000$. When optimizing, we limit the total optimization steps to 30 with a learning rate of 0.2. All our experiments were conducted using the MQuAKE dataset. To test the accuracy of answers to multi-hop questions, we adhered to the few-shot and Chain of Thought (CoT) templates in Appendix I.1 and procedures.

## I  PROMPTS

This appendix details the various prompt templates used in our experiments. These templates are used to evaluate the model's multi-hop fact recall ability after knowledge editing, as well as for automatic classification and annotation of the dataset.

### I.1  FEW-SHOT AND CHAIN-OF-THOUGHT EVALUATION PROMPTS

The following is an example of a few-shot and Chain of Thought (CoT) prompt used for the main experiment evaluation. We guide the model to answer complex multi-hop questions by showing a reasoning process containing "Thoughts".

```
Question: What is the capital of the country where Plainfield Town Hall is located?
Thoughts: Plainfield Town Hall is located in the country of the United States of America.
The capital of United States is Washington, D.C.
Answer: Washington, D.C.

Question: In which country is the company that created Nissan 200SX located?
Thoughts: Nissan 200SX was created by Nissan. Nissan is located in the country of Japan.
Answer: Japan

Question: Which continent is the country where the director of "My House Husband: Ikaw Na
!" was educated located in?
Thoughts: The director of "My House Husband: Ikaw Na!" is Jose Javier Reyes. Jose Javier
Reyes was educated at De La Salle University. De La Salle University is located in the
country of Philippines. Philippines is located in the continent if Asia.
Answer: Asia

Question: Who is the spouse of the US president?
Thoughts: The US president is Joe Biden. The spouse of Joe Biden is Jill Biden
Answer: Jill Biden

Question: Who has ownership of the developer of the Chevrolet Corvette (C4)?
Thoughts: The developer of Chevrolet Corvette (C4) is Chevrolet. Chevrolet is owned by
General Motors.
Answer: General Motors
Question:{Multi-hop questions...}
```

### I.2  PROMPTS TO RECALL SINGLE-HOP FACT

The following prompt templates are used in our dataset screening and construction process, as described in Appendix C.1. Before formally conducting multi-hop knowledge editing experiments, we use these straightforward, single-hop factual questions to probe the inherent knowledge reserves of base models (such as GPT-J-6B). This step allows us to identify knowledge chains from the MQuAKE dataset where the model has accurately grasped the ground truth, ensuring the reliability of subsequent experiments and avoiding interference caused by the model's inherent lack of foundational knowledge.

```
Q: What is the country of citizenship of Fernando Santos? A: Portugal
Q: What is the name of the current head of state in Portugal? A: Marcelo Rebelo de Sousa
Q: Who was Aslan created by? A: C. S. Lewis
Q: Which city was C. S. Lewis born in? A: Belfast
Q: Which city was Hari Kunzru born in? A: London
Q: Which continent is London located in? A: Europe
Q: Who was Nick Bottom created by? A: William Shakespeare
Q: What kind of work does William Shakespeare do? A: playwright
Q: Who is the head coach of Iran national football team? A: Carlos Queiroz
Q: Which sport is Carlos Queiroz associated with? A: association football
Q:{Single-hop questions...}
```

### I.3  DATASET ANNOTATION PROMPT

The following is a prompt template for requesting GPT-4o to semantically classify questions in the MQuAKE dataset, as described in Appendix C.2. This template ensures consistency and automation of data annotation by providing strict format requirements.

Table 10: Multi-hop accuracy comparison of different KE methods on the MQuAKE-3K dataset in a few-shot setting, Base shows the model's performance on the unedited answers and edited model's performance on edited answers. Our model outperformances than other models significantly.

| Editor | Avg.(GPT/Qwen) | GPT-J / # Edits = | | | | Qwen3-8B / # Edits = | | | |
|---|---|---|---|---|---|---|---|---|---|
| | | 1-edit | 2-edit | 3-edit | 4-edit | 1-edit | 2-edit | 3-edit | 4-edit |
| Base | 98.42 / 99.17 | 99.7 | 95.48 | 97.51 | 97.23 | 99.81 | 97.46 | 98.14 | 97.64 |
| FT | 3.54 / 2.18 | 4.17 | 2.63 | 0.00 | 0.00 | 3.14 | 2.79 | 0.00 | 0.00 |
| ROME | 35.04 / 28.79 | 44.51 | 38.93 | 17.52 | 5.06 | 35.09 | 32.48 | 18.97 | 7.08 |
| MEMIT | 38.58 / 18.67 | 64.30 | 16.87 | 17.25 | 8.16 | 29.84 | 19.18 | 12.91 | 4.20 |
| PMET | 37.01 / 20.78 | 49.26 | 36.30 | 24.34 | 17.01 | 28.64 | 14.08 | 12.56 | 11.20 |
| AlphaEdit | 39.27 / 45.59 | 47.24 | 37.69 | 29.59 | 28.98 | 54.58 | 48.62 | 27.54 | 27.31 |
| AcE (PMET) | 46.45 / 58.24 | 45.26 | 50.24 | 36.17 | 43.29 | 60.22 | 59.48 | 51.62 | 47.61 |
| AcE (AlphaEdit) | **49.24 / 61.54** | 52.91 | **53.74** | **39.08** | **43.82** | **62.48** | **64.76** | **63.04** | **51.32** |

```
Please analyze the type of the following question and return two categories strictly in
the following format, separated by '|':

[Main category]|[Subcategory]

Question: {question}
Category:
```

## J  AcE Latency

In our experiments, we utilized four A800 (80G) GPUs for computation. AcE can be decoupled into two distinct stages: identifying and editing; consequently, its latency is discussed by dividing it into these two components. Quite straightforwardly, the latency consumed by AcE in stage 1 can be approximated as the model's inference time. Our primary focus in the discussion will be on the latency consumed by the model in stage 2. In AcE, we have the luxury of being able to pre-compute and cache the knowledge values, since they are inserted to the model parallel. If all knowledge vectors are already computed, AcE takes $3.10$ & $3.14 \, \text{sec}$ for per update on GPT-J and Qwen3-8B, respectively. The most computationally expensive step is inverting a large square matrix $\Delta$ in Equation 12. To get all knowledge vectors, we need $26,474.35$ & $27,195.26 \, \text{sec}$ on GPT-J and Qwen3, however, this expensive computing result could be cache for incoming edits.

## K  AcE Extension

In this section, we extend our AcE framework to AlphaEdit version. Alphaedit Fang et al. (2024), a novel solution that projects perturbation onto the null space of the preserved knowledge before applying it to the parameters. As shown in Table 10, we add AlphaEdit as one new baseline for comparison. We also replace the AlphaEdit as our new backbone. After replacing the backbone, our method outperforms 9.97% and 15.95% compared to AlphaEdit. Moreover, this extended experiment claims that our Identify-Locate-then-Edit framework could be extended to more later KE methods.

## L  Human Understanding Alignment

In this section, we present the human alignment validation for GPT-4o-based dataset labeling. To ensure both methodological rigor and demographic diversity in the alignment assessment, the selected participants comprised an undergraduate student in literature, a graduate student in social sciences, a Ph.D. candidate in STEM, and an undergraduate student in STEM. As shown in Table 11, we computed the overlap rate between human labeling and GPT-4o labeling, we find that most of the human participants reach an acceptable understanding overlap with GPT-4o on our random sample batches (80 multi-hop questions).

Table 11: The result of human understanding alignment with GPT-4o labeling.

| Human Participants | Overlap Rate |
| --- | --- |
| Human A | 83.75% |
| Human B | 88.75% |
| Human C | 93.75% |
| Human D | 80.00% |

# M MORE EXPERIMENTS

## M.1 LOCALITY PERFORMANCE

To comprehensively demonstrate the generalizability of our editing framework and its ability to preserve model capabilities, we have supplemented our evaluation with experiments on four general reasoning benchmarks that assess the model's performance on unrelated tasks after editing. As demonstrated in Table 12, edited models retain their core capabilities without significant degradation on unrelated tasks.

## M.2 ATTRIBUTION ROBUSTNESS

To evaluate the robustness of our attribution mechanism, we performed multiple inference runs and compared the average importance scores of critical layers across different sampling trials. As shown in Table 13 below, the importance scores for our identified critical modules demonstrate strong stability across different Pass@k settings. The minimal variance in importance scores across different Pass@k trials (e.g., $f_{27}$ varies by only $\tilde{0}.47$ points across all settings) confirms that our attribution method identifies consistently significant neural pathways, rather than capturing random or unstable activation patterns.

We also quantify the robustness and stability of attributed query and value layers' rankings across different prompt templates. To quantitatively assess this stability, we conducted a new experiment where we re-ran our layer importance attribution under different in-context learning settings: Zero-Shot and One-Shot, in addition to our original Few-Shot setting. We then compared the Top-9 important attention and FFN layers across these settings for all semantic categories. The results for the GPT-J model are presented as Table 14 and Table 15. For clarity, we have underlined any layer where its rank changed compared to the Few-Shot setting used in the main paper. As the tables clearly demonstrate, the rankings of the most critical layers exhibit remarkable stability across different prompting settings:

- **Consistency of Top Layers:** The core set of highly important layers remains virtually unchanged. For instance, attention layers $a_{27}$, $a_{26}$, and $a_7$ consistently rank in the top positions across nearly all categories and settings. Similarly, FFN layers like $f_{26}$, $f_{27}$ and $f_{24}$ are stably identified as critical.

- **Minor Nature of Changes:** The few rank changes that do occur (underlined) are predominantly in the lower half of the Top-9 list.

- **Practical Implication for ACE:** This stability is crucial for ACE's practicality. It means that the critical Q/V layers for a given type of knowledge can be reliably identified using a standard prompting setup (e.g., Few-Shot). The subsequent edits targeting these layers are then likely to be effective across various phrasings of queries involving that knowledge, as the underlying neural pathways remain the primary routing and storage sites.

## M.3 VERIFYING INTERMEDIATE REASONING PROCESS CHANGES

Now we verify that the edited value propagates through the intended implicit subjects, rather than being injected late. To address this, we conducted a new experiment designed to trace the flow of information through the reasoning chain before and after a critical intervention: masking the final deep value editing. The core logic is as follows: If the edited knowledge is merely "injected" at the final generation step, then masking the final value edit should not significantly affect the activation of

Table 12: Locality Performance on several general benchmarks of ACE and other editing methods.

| Editor | CSQA | BBH | MMLU | GSM8k |
|---|---|---|---|---|
| ROME (Qwen3-8B) | 75.41 | 35.62 | 65.28 | 75.80 |
| MEMIT (Qwen3-8B) | 82.49 | 32.68 | 72.64 | 83.32 |
| PMET (Qwen3-8B) | 82.26 | 34.20 | 71.80 | 83.00 |
| ACE (Qwen3-8B) | 82.30 | 38.54 | 72.30 | 83.84 |

Table 13: The average importance score of Pass@k in identifying critical layers.

| Layer | Pass@1 | Pass@2 | Pass@3 | Pass@5 |
|---|---|---|---|---|
| $a_{27}$ (Qwen3-8B) | 110.48 | 109.29 | 109.45 | 109.69 |
| $a_{26}$ (Qwen3-8B) | 97.58 | 98.21 | 97.94 | 98.02 |
| $a_7$ (Qwen3-8B) | 64.29 | 64.80 | 64.75 | 64.78 |
| $f_{27}$ (Qwen3-8B) | 137.48 | 137.95 | 137.66 | 137.74 |
| $f_{24}$ (Qwen3-8B) | 104.20 | 104.39 | 104.28 | 104.23 |
| $f_{16}$ (Qwen3-8B) | 42.58 | 42.60 | 42.64 | 42.52 |

intermediate, implicit subjects. Conversely, if the edited knowledge is genuinely propagated through the chain via implicit subjects, then the intermediate activations should remain robust even when the final value edit is masked, while the final answer activation alone should drop precipitously. As the results demonstrated in Table 16 and 17, the activation patterns at critical intermediate tokens representing the implicit subject (e.g.,'plays' to 'basketball') remain stable and highly interpretable after masking the final value edit. The importance scores for 'plays' show minimal change, and its top vocabulary tokens remain strongly associated with the correct sport. This indicates that the query-layer edits successfully guide the model to the correct implicit subject ('basketball'), independent of the final answer's direct manipulation. In contrast, the activation at the final answer token 'from' experiences a severe drop (over 40%) after masking the value edit. Furthermore, the semantic specificity in its vocabulary projection degrades from concrete country names ('USA', 'America') to generic and uncertain terms ('country', 'many'). This clear dissociation—preserved intermediate reasoning but disrupted final answer generation—provides definitive evidence that ACE's edits propagate the updated knowledge through the intended implicit subjects along the reasoning chain. The final value edit then truthfully enhances the prediction based on this correctly propagated context ('basketball' to 'USA'), rather than injecting the answer "USA" directly at the end.

### M.4 COUNTER-FACTUAL EDIT

To demonstrate that our ACE has effectively edited the model, we have conducted experiments on counterfactual editing. We mismatched the relationship labels in the knowledge triplet to construct a counterfactual editing dataset. The results, presented in Table 18, demonstrate ACE's behavior under adversarial conditions. We also add the experiments the general benchmarks to claim that the

Table 14: Top 9 important attention layers (left block) and FFN layers (right block) in GPT-J using zero-shot implementation. Underline terms to the rank of the layer occurs changes.

| | \multicolumn{17}{c}{Top 9 important attention layers and FFN layers (zero-shot)} | | | | | | | | | | | | | | | | |
|---|---|---|---|---|---|---|---|---|---|---|---|---|---|---|---|---|---|---|
| NN | $a_{27}$ | $a_{26}$ | $a_7$ | $a_{10}$ | $a_9$ | $\underline{a_8}$ | $\underline{a_{25}}$ | $a_{11}$ | $a_5$ | $f_{20}$ | $f_{24}$ | $f_{16}$ | $\underline{f_{22}}$ | $\underline{f_{18}}$ | $\underline{f_{15}}$ | $f_{23}$ | $f_{26}$ | $f_{25}$ |
| CT | $a_{27}$ | $a_{26}$ | $a_7$ | $a_{10}$ | $a_8$ | $a_5$ | $a_9$ | $a_{11}$ | $a_{25}$ | $f_{22}$ | $f_{24}$ | $f_{16}$ | $f_{21}$ | $f_{15}$ | $f_{17}$ | $f_{26}$ | $f_{25}$ | $f_{23}$ |
| LG | $a_{27}$ | $\underline{a_5}$ | $\underline{a_7}$ | $a_6$ | $a_8$ | $a_4$ | $a_1$ | $a_{26}$ | $a_9$ | $f_{27}$ | $f_7$ | $f_5$ | $f_6$ | $f_8$ | $f_4$ | $f_1$ | $f_{26}$ | $f_9$ |
| CP | $a_{27}$ | $a_{26}$ | $a_7$ | $a_{10}$ | $a_8$ | $a_9$ | $a_{12}$ | $a_5$ | $a_6$ | $f_{27}$ | $f_{26}$ | $f_7$ | $f_{10}$ | $f_{12}$ | $f_8$ | $f_9$ | $f_5$ | $f_6$ |
| LS | $a_{27}$ | $a_{26}$ | $a_{25}$ | $a_7$ | $a_{10}$ | $a_6$ | $a_8$ | $a_6$ | $a_5$ | $f_{27}$ | $f_{26}$ | $f_{25}$ | $f_{24}$ | $f_7$ | $f_{10}$ | $f_6$ | $f_8$ | $f_5$ |
| AT | $a_{27}$ | $a_{26}$ | $a_{25}$ | $a_7$ | $a_9$ | $a_8$ | $a_{10}$ | $a_5$ | $a_6$ | $f_7$ | $f_9$ | $f_8$ | $f_{10}$ | $f_{27}$ | $f_{26}$ | $f_{25}$ | $f_5$ | $f_6$ |
| ST | $a_{26}$ | $a_{27}$ | $a_7$ | $a_{10}$ | $a_8$ | $a_6$ | $a_5$ | $a_{25}$ | $a_4$ | $f_{26}$ | $f_{27}$ | $f_7$ | $f_{10}$ | $f_8$ | $f_6$ | $f_5$ | $f_{25}$ | $f_4$ |
| CF | $a_{26}$ | $a_{27}$ | $\underline{a_{25}}$ | $\underline{a_{24}}$ | $a_7$ | $a_8$ | $a_9$ | $a_6$ | $a_5$ | $f_{26}$ | $f_{27}$ | $f_{24}$ | $f_{25}$ | $f_7$ | $f_8$ | $f_9$ | $f_6$ | $f_5$ |

Table 15: Top 9 important attention layers (left block) and FFN layers (right block) in GPT-J using one-shot implementation. Underline terms to the rank of the layer occurs changes.

| | | | | | | | | | | | | | | | | | | |
|---|---|---|---|---|---|---|---|---|---|---|---|---|---|---|---|---|---|---|
| **Top 9 important attention layers and FFN layers (one-shot)** | | | | | | | | | | | | | | | | | | |
| NN | $a_{27}$ | $a_{26}$ | $a_7$ | $a_{10}$ | $a_9$ | $a_{25}$ | $a_8$ | $a_{11}$ | $a_5$ | $f_{20}$ | $f_{24}$ | $f_{16}$ | $f_{18}$ | $f_{15}$ | $f_{22}$ | $f_{23}$ | $f_{26}$ | $f_{25}$ |
| CT | $a_{27}$ | $a_{26}$ | $a_7$ | $a_{10}$ | $a_8$ | $a_5$ | $a_9$ | $a_{11}$ | $a_{25}$ | $f_{22}$ | $f_{24}$ | $f_{16}$ | $f_{21}$ | $f_{15}$ | $f_{17}$ | $f_{26}$ | $f_{25}$ | $f_{23}$ |
| LG | $a_{27}$ | $a_7$ | $a_5$ | $a_6$ | $a_8$ | $a_4$ | $a_1$ | $a_{26}$ | $a_9$ | $f_{27}$ | $f_7$ | $f_5$ | $f_6$ | $f_8$ | $f_4$ | $f_1$ | $f_{26}$ | $f_9$ |
| CP | $a_{27}$ | $\underline{a_7}$ | $\underline{a_{26}}$ | $a_{10}$ | $a_8$ | $a_9$ | $a_{12}$ | $a_5$ | $a_6$ | $f_{27}$ | $f_{26}$ | $f_7$ | $f_{10}$ | $f_{12}$ | $f_8$ | $f_9$ | $f_5$ | $f_6$ |
| LS | $a_{27}$ | $a_{26}$ | $a_{25}$ | $a_7$ | $a_{10}$ | $a_6$ | $a_8$ | $a_6$ | $a_5$ | $f_{27}$ | $f_{26}$ | $f_{25}$ | $f_{24}$ | $f_7$ | $f_{10}$ | $f_6$ | $f_8$ | $f_5$ |
| AT | $a_{27}$ | $a_{26}$ | $a_{25}$ | $a_7$ | $a_9$ | $a_8$ | $a_{10}$ | $a_5$ | $a_6$ | $f_7$ | $f_9$ | $f_8$ | $f_{10}$ | $f_{27}$ | $f_{26}$ | $f_{25}$ | $f_5$ | $f_6$ |
| ST | $a_{26}$ | $a_{27}$ | $a_7$ | $a_{10}$ | $a_8$ | $a_6$ | $a_5$ | $a_{25}$ | $a_4$ | $f_{26}$ | $f_{27}$ | $f_7$ | $f_{10}$ | $f_8$ | $f_6$ | $f_5$ | $f_{25}$ | $f_4$ |
| CF | $a_{26}$ | $a_{27}$ | $a_{24}$ | $a_{25}$ | $a_7$ | $a_8$ | $a_9$ | $a_6$ | $a_5$ | $f_{26}$ | $f_{27}$ | $f_{24}$ | $f_{25}$ | $f_7$ | $f_8$ | $f_9$ | $f_6$ | $f_5$ |

Table 16: The Original Token Increase in Qwen3-8B on Residual Stream.

| Token | Importance increase | Top tokens in vocabulary space |
|---|---|---|
| Tim | FFN: 0.0014, attn: 0.0014 | 'an', 'os', 'ise', 'Exactly', 'R', 'ore', 'at', 'rot' |
| Duncan | FFN: 0.0009, attn: 0.0009 | 'era', 'allen', 'stad', 'oret', 'hit', '-led' |
| plays | FFN: 0.9846, attn: 0.8167 | **'basketball', 'NBA', 'career', 'ball'**, '-playing' |
| the | FFN: 0.4109, attn: 0.4159 | 'epit', 'inaugural', 'bidding', 'dream', 'etr' |
| sport | FFN: 0.0848, attn: 0.0948 | 'tennis', 'of', 'ful', 'arena', 'basketball', 'ball' |
| of | FFN: 0.8671, attn: 0.6198 | **'basketball', 'NBA', 'balls', 'ball'**, 'Olympia' |
| originates | FFN: 0.3508, attn: 0.3058 | 'kati', 'the', 'from', 'oret', 'orig', 'ball' |
| from | FFN: 0.9483, attn: 0.8720 | **'USA', 'America' , 'the', 'US', 'U.S.A.'** |

Table 17: The Token Increase in Qwen3-8B on Residual Stream After Masking Value Editing.

| Token | Importance increase | Top tokens in vocabulary space |
|---|---|---|
| Tim | FFN: 0.0019, attn: 0.0010 | 'an', 'os', 'ise', 'Exactly', 'R', 'ore', 'at', 'rot' |
| Duncan | FFN: 0.0008, attn: 0.0012 | 'era', 'allen', 'stad', 'oret', 'hit', '-led' |
| plays | FFN: 0.9779, attn: 0.8498 | **'basketball', 'NBA', 'career', 'ball'**, '-playing' |
| the | FFN: 0.4093, attn: 0.4715 | 'epit', 'inaugural', 'bidding', 'dream', 'etr' |
| sport | FFN: 0.0913, attn: 0.0142 | 'tennis', 'of', 'ful', 'arena', 'basketball', 'ball' |
| of | FFN: 0.9252, attn: 0.7090 | **'basketball', 'NBA', 'balls', 'ball'**, 'Olympia' |
| originates | FFN: 0.2308, attn: 0.3194 | 'kati', 'the', 'from', 'oret', 'orig', 'ball' |
| from | FFN: 0.5603, attn: 0.4764 | **'country', 'many' , '-Info', 'bot', 'US', 'USA'** |

Table 18: The detailed results of more metrics in experiments on GPT-J and Qwen3-8B in counter-fact scenarios.

| Editor (Counter-Fact) | Efficacy | Paraphrase | Specificity | Avg. |
|---|---|---|---|---|
| FT (GPT-J/Qwen3) | **98.1 / 97.9** | 69.4 / 64.2 | **82.7** / 77.4 | 1.59 / 1.04 |
| ROME (GPT-J/Qwen3) | 54.1 / 45.2 | 54.3 / 42.9 | 61.4 / 53.7 | 27.48 / 24.08 |
| MEMIT (GPT-J/Qwen3) | 57.0 / 50.3 | 51.9 / 53.6 | 66.2 / 66.4 | 30.09 / 10.27 |
| PMET (GPT-J/Qwen3) | 74.6 / 70.7 | 63.2 / 61.7 | 64.1 / 51.9 | 31.07 / 17.26 |
| AcE (GPT-J/Qwen3) | 89.7 / 91.2 | **83.6 / 80.7** | 70.6 / **74.6** | **43.58 / 54.27** |

locality performances of edited model are not corrupted highly. As demonstrated in Table 19, the result shows that after counter-factual editing in the model, the locality of the model keeps stable among four general benchmarks, showing the robustness and safety in our AcE framework. AcE effectively edits knowledge even in incorrect reasoning chains, demonstrating that it modifies the model's internal representations rather than merely amplifying correct reasoning patterns. AcE also maintains strong locality performance across general benchmarks (CSQA, BBH, MMLU, GSM8k), showing minimal negative impact on unrelated capabilities.

Table 19: Locality Performance on several general benchmarks of AcE and other editing methods in counter-fact scenarios.

| Editor (Counter-Fact) | CSQA | BBH | MMLU | GSM8k |
|---|---|---|---|---|
| ROME (Qwen3-8B) | 68.54 | 29.47 | 60.71 | 69.48 |
| MEMIT (Qwen3-8B) | 71.46 | 29.72 | 66.40 | 72.45 |
| PMET (Qwen3-8B) | 73.09 | 26.58 | 62.49 | 74.29 |
| AcE (Qwen3-8B) | 76.41 | 33.79 | 69.97 | 79.02 |

# N    AcE ATTRIBUTION DERIVATION

In this section, we provide a step-by-step formal derivation of the AcE attribution metrics. We begin by modeling the algebraic structure of the Transformer Residual Stream, prove the strict additivity of Logits, and derive the Importance Score via first-order Taylor approximation. Finally, we prove that the additivity assumption is mathematically isomorphic to the update mechanism of the PMET editor.

## N.1    ALGEBRAIC STRUCTURE OF THE TRANSFORMER

Let $\mathcal{M}$ be an autoregressive Transformer model with $L$ layers. We define the hidden state propagation through the residual stream.

**Definition 1 (Residual Stream Dynamics).** *Let $h^{(l)} \in \mathbb{R}^d$ denote the hidden state input to layer $l$. The output of layer $l$ is the sum of the Multi-Head Self-Attention (MHSA) output and the Feed-Forward Network (FFN) output, added to the residual stream:*

$$h^{(l+1)} = h^{(l)} + MHSA^{(l)}(h^{(l)}) + FFN^{(l)}(h^{(l)})$$

*By recursive expansion, the final hidden state $h^{(L)}$ is the cumulative sum of the initial embedding and all layer outputs:*

$$h^{(L)} = h^{(0)} + \sum_{l=0}^{L-1} MHSA^{(l)}(h^{(l)}) + \sum_{l=0}^{L-1} FFN^{(l)}(h^{(l)})$$

**Definition 2 (FFN as Key-Value Memories).** *Let $K^{(l)}$ and $V^{(l)}$ be the parameter matrices (also denoted as $W_{fc1}$ and $W_{fc2}$). The output is a weighted sum of value vectors:*

$$FFN^{(l)}(x) = \sum_{i=1}^{N} m_{l,i}(x) \cdot v_{l,i}$$

*where $v_{l,i} \in \mathbb{R}^d$ is the $i$-th column of the second weight matrix $W_{fc2}^{(l)}$, and $m_{l,i}(x) = \sigma(x^T k_{l,i})$ is the scalar activation coefficient.*

## N.2    THEOREM OF LOGIT LINEARITY

The probability distribution over the vocabulary is computed via the Softmax function applied to the logits $z \in \mathbb{R}^{|\mathcal{V}|}$. Let $E \in \mathbb{R}^{d \times |\mathcal{V}|}$ be the unembedding matrix, and $e_w$ be the column vector corresponding to token $w$.

**Lemma 1 (Logits).** *The logit $z_w$ for a target token $w$ is the inner product of the final state and the token embedding:*

$$z_w = \langle h^{(L)}, e_w \rangle$$

**Theorem 2 (Strict Linearity of Neuron Contribution).** *The contribution of any single FFN neuron $(l, i)$ to the target logit $z_w$ is strictly additive and independent of other neurons in the logit space.*

*Proof.* Substituting Definition 2 into Definition 1, and then into Lemma 1:

$$z_w = \left\langle h^{(0)} + \sum \text{MHSA} + \sum_{l,j} m_{l,j} v_{l,j}, e_w \right\rangle$$

$$= \langle h^{(0)} + \sum \text{MHSA}, e_w \rangle + \sum_{l,j} \langle m_{l,j} v_{l,j}, e_w \rangle$$

Let $z_{base} = \langle h^{(0)} + \sum \text{MHSA}, e_w \rangle$ be the base logit. The equation simplifies to:

$$z_w = z_{base} + \sum_{l=j} m_{l,j} (v_{l,j}^T e_w)$$

Define the marginal contribution of neuron $(l, i)$ as $\Delta z_{l,i} = m_{l,i}(v_{l,i}^T e_w)$. It follows that:

$$z_w(S) = z_{base} + \sum_{(l,i) \in S} \Delta z_{l,i}$$

where $S$ is any set of active neurons. The interaction terms are zero in the logit space. $\square$

### N.3 DERIVATION OF IMPORTANCE SCORE

ACE defines the importance score $\mathcal{I}$ based on the change in Log-Probability. While Log-Probability is non-linear, we prove that Equation 9 is the First-Order Taylor Approximation of the causal effect.

Let the objective function be $\mathcal{L}(z) = \log P(w) = \log \left( \frac{e^{z_w}}{\sum_k e^{z_k}} \right)$. Consider the activation of a neuron $v$ as a perturbation vector $\mathbf{u} = m \cdot v$. This causes a perturbation in the logits $\Delta z = \mathbf{u}^T E$. We perform a first-order Taylor expansion of

$$\mathcal{L}(z + \Delta z)$$

around the current logit $z$:

$$\mathcal{L}(z + \Delta z) \approx \mathcal{L}(z) + \nabla_z \mathcal{L}(z)^T \cdot \Delta z$$

The gradient of the Log-Softmax function w.r.t the logits $z$ is:

$$\frac{\partial \mathcal{L}}{\partial z_k} = \delta_{kw} - P(k)$$

where $\delta_{kw}$ is the Kronecker delta. Substituting the gradient and the logit perturbation:

$$\mathcal{I}(v) := \mathcal{L}(z + \Delta z) - \mathcal{L}(z)$$

$$\approx \sum_{k \in \mathcal{V}} (\delta_{kw} - P(k)) \cdot \Delta z_k$$

$$= (1 - P(w))\Delta z_w - \sum_{k \neq w} P(k)\Delta z_k$$

This derivation confirms that Eq. 9 is formally the Marginal Contribution to the Logit ($\Delta z$), projected onto the tangent space of the probability simplex.

### N.4 IMPORTANCE ADDITIVITY

**Theorem 3** (**Additivity of First-Order Attribution**). *Under the assumption that the local curvature of the Log-Likelihood manifold is negligible (first-order approximation), the importance score of a set of neurons is the sum of their individual scores.*

*Proof.* From Theorem 1, the total logit perturbation is the sum of individual perturbations:

$$\Delta z_{total} = \sum_i \Delta z_i$$

The importance score is a linear map of the logit perturbation:

$$\mathcal{I}(\Delta z) \approx \nabla_z \mathcal{L}^T \cdot \Delta z$$

By the linearity of the inner product:

$$\mathcal{I}_{total} \approx \nabla_z \mathcal{L}^T \cdot \left( \sum_i \Delta z_i \right)$$
$$= \sum_i \left( \nabla_z \mathcal{L}^T \cdot \Delta z_i \right)$$
$$\approx \sum_i \mathcal{I}(v_i)$$

Thus, layer importance is mathematically valid within the local neighborhood defined by the Taylor expansion. $\square$

### N.5    ALIGNMENT WITH PMET

Finally, we prove that the additivity of ACE is a necessary condition imposed by the update mechanism of the PMET editor.

**Theorem 4** (**Linearity of PMET Updates**). *The PMET editor updates the FFN weights $W$ by solving a linear least-squares problem(6). The closed-form solution for the weight update $\Delta W$ is:*

$$\Delta W = RK^T (C_0 + KK^T)^{-1}$$

*This results in a linear shift in the output of the FFN for a given input $k_{in}$:*

$$\delta FFN = \Delta W \cdot k_{in}$$

*Proof.* Let us decompose the FFN output shift $\delta$FFN into the basis of neurons. Since $W$ is composed of columns $v_i$, the update $\Delta W$ corresponds to updating individual value vectors $v_i \leftarrow v_i + \delta v_i$.

$$\delta\text{FFN} = \sum_i m_i \cdot (v_i + \delta v_i) - \sum_i m_i \cdot v_i = \sum_i m_i \cdot \delta v_i$$

PMET operates by injecting a sum of linear updates into the residual stream. If ACE were to use a non-additive metric, it would identify neurons crucial for non-linear inference. However, PMET's linear update mechanism $\Delta W$ is mathematically incapable of manipulating such non-linearities. $\square$

