# OpenReview forum: "ACE: Attribution-Controlled Knowledge Editing for Multi-hop Factual Recall"
_ICLR.cc/2026/Conference — ICLR 2026 Poster_

### Official Review · Reviewer_pPXV · 2025-10-24

**Soundness:** 2
**Presentation:** 2
**Contribution:** 2
**Rating:** 4
**Confidence:** 4

**Summary:**

This paper explores multi-hop knowledge editing. The authors hypothesizes that the failure is because the missing tracking of intermediate implicit subjects within reasoning chains. Using a suite of attribution metrics, they aim to uncover the mechanism of multi-hop reasoning. Based on the analysis of identifying important attention layers and FFN layers, Attribution-Controlled Knowledge Editing is proposed. Evaluation on a single dataset Mquake-3k is performed to demonstrate the effectiveness of the proposed method.

**Strengths:**

- This paper explores an interesting and important direction of multi-hop knowledge editing.

- This paper not only presents a solution, but also gives the analysis on which the approach is conditioned.

**Weaknesses:**

- The two takeaways are well-explored and not novel. For example, *Takeaway 2: Information of the final answer is accumulated through implicit query neurons that sequentially activate corresponding value neurons across the reasoning chain.* is verified in [1].

- The analysis in Section 4 lacks sufficient rigor. In particular, the selection of the subset from MQuAKE-3K is not clearly justified. Although the analysis is entirely based on this subset, no details about it are provided. For instance, the number of samples or their distribution across different knowledge types. Given that MQuAKE-3K is not a large dataset and the analysis only involves forward passes, it would be more convincing to conduct the analysis on the full MQuAKE-3K dataset rather than on an unspecified subset.

- The deduction from the analysis is not convincing： a27, a26, a7 ranks top in all knowledge-> MHSA stores general knowledge and capabilities in LLMs.

- Authors claim that FFN layers tends to primarily extracts its own knowledge. But f_26 ranks top for all knowledge types, which contradicts the conclusion.

- In the identifying process, forward passes on all multi-hop questions in the dataset and computed the sum of the importance scores at the last token position are performed. Is this setting reasonable? Is this instance-level or dataset-level editing?

- The whole evaluation is based on one benchmark MQuAKE-3K, which cannot demonstrate the generalization of the conclusion. It would be better to conduct the evaluation on more multi-hop knowledge editing datasets.

[1] CaKE: Circuit-aware Editing Enables Generalizable Knowledge Learners.

**Questions:**

See above

---

> ### Author Response · Authors · 2025-11-21
> **Response to Review pPXV, Weakness1**
>
> ***
>
>
> ### **Response to Weakness 1: Novelty of Takeaway 2**
>
> We thank the reviewer for raising this important point regarding novelty and for directing us to the highly relevant CaKE paper. We agree that CaKE provides valuable insights into the circuit-level behavior during multi-hop reasoning. However, our work complements and significantly deepens this understanding by moving from the circuit level to the fine-grained neuron level, uncovering the specific mechanistic drivers behind the observed circuit dynamics. Our Takeaway 2 is novel in the following three aspects:
>
> **A. Granularity: From Circuit Pathways to Neuron-Level Activations.**
>   - CaKE analyzes the propagation of information (e.g., the bridge entity e2) between positions (t1, t2) and across layers, which is a circuit-level observation.
>   - In contrast, ACE identifies the fundamental computational units that implement this propagation: the query-value (Q-V) neuron pairs. We reveal that implicit subjects function as "query neurons" which sequentially activate "value neurons" across layers (Fig. 3). This neuron-level orchestration is the underlying mechanism that explains how and why the circuit in CaKE functions or fails.
>
> **B. Interpretability: Linking Mechanism to Human-Understandable Concepts.**
>   - A key novel finding in ACE is that the critical value neurons we identify are often semantically interpretable when projected into the vocabulary space (Table 8, Appendix E). For instance, a value neuron might be highly associated with tokens like "basketball" or "NBA".
>   - This allows for a post-hoc, human-aligned analysis of the reasoning chain. More importantly, our case study (Sec. 6.3) demonstrates that ablating a sparse set of these interpretable neurons catastrophically harms performance (accuracy drops to 3.2%), while ablating equally important but non-interpretable neurons does not. This provides causal evidence that multi-hop reasoning relies on a coordinated set of specialized, concept-specific neurons, a finding not explored in CaKE.
>
> **C. Discovery: Uncovering Fine-Grained, Intra-Sentence Activation Patterns.**
>   - Our analysis on Qwen3-8B revealed an even more fine-grained dynamic: the importance scores for critical neurons are not uniformly distributed across a sentence. Instead, they peak at semantically convergent tokens (e.g., "plays", "of" in the query "Tim Duncan plays the sport of"), as we elaborated, these high-activation tokens/neurons functionally serve as query neurons within the reasoning chain. Their role is to integrate information from the preceding residual stream and propagate it forward, thereby activating the subsequent value neurons responsible for generating the final answer.(Table 7, Appendix E).
>   - This intra-sentence activation pattern, modulated by the knowledge domain, represents a more microscopic view of how reasoning is computed forward pass-by-pass, going beyond the layer-and-position analysis in CaKE.
> We believe this mechanistic, neuron-level insight is a novel and significant contribution to the understanding of multi-hop reasoning.
>
>
> ***

---

> ### Author Response · Authors · 2025-11-21
> **Response to Review pPXV, Weakness2**
>
> ***
>
>
>
> ### **Response to Weakness 2: Rigor of Analysis (Dataset Subset & Labeling)**
>
>
>
> We thank the reviewer for these important questions regarding our dataset construction and labeling procedures. We address these concerns separately below and appreciate the opportunity to clarify our methodology.
>
>
>
> **A. Subset Construction: Ensuring a Controlled Evaluation of Editing Efficacy**
>
>
>
> Regarding the construction of our MQuAKE subset, we deliberately employed a controlled filtering process to ensure experimental fairness and validity, as detailed in **Appendix C.1**. Specifically:
>
>
>
> * We used the vanilla GPT-J model to evaluate **all single-hop factual components** within the multi-hop queries from the original MQuAKE dataset.
>
> * We applied a stringent criterion, selecting only those knowledge chains where GPT-J could correctly answer **all constituent single-hop facts** in a zero-shot setting.
>
>
>
> This approach is crucial for preventing a critical confounding factor: if a model inherently lacks certain knowledge, its post-editing performance would be unfairly penalized not due to editing failure, but due to **pre-existing knowledge gaps**. By constructing a subset where the base model demonstrably knows the original facts, we isolate and accurately measure the **pure efficacy of the knowledge editing methods themselves**. We will clarify this justification more prominently in the main text in the final version.
>
>
>
> **B. Dataset Labeling: Validation through Human Alignment**
>
>
>
> For the semantic categorization of knowledge types, we aimed for both scalability and consistency:
>
>
>
> * We used a **single, fixed prompt template** (provided in **Appendix H.3**) for all GPT-4o labeling to ensure uniformity.
>
> * To quantitatively validate the reliability of these automated labels, we conducted a human alignment study with four participants from diverse academic backgrounds (literature, social sciences, and STEM).
>
>
>
> The results, presented in **Appendix K (Table 11)**, demonstrate strong agreement between human judgment and GPT-4o labeling:
>
>
>
> | Human Participants | Overlap Rate |
> | ------------------ | ------------ |
> | Human A            | 83.75%       |
> | Human B            | 88.75%       |
> | Human C            | 93.75%       |
> | Human D            | 80.00%       |
>
>
>
> These results show that our automated labeling process achieves **high overlap rates (80% to 93.75%)** with human understanding, confirming that the resulting categories are semantically meaningful and robust.
>
> We also demonstrate the knowledge label distribution here to clarify more details about the subset we conducted of your convinience.
>
> | **Knowledge Label** | **Percentage** |
> | ------------------- | -------------- |
> | NN                  | 14.7           |
> | CT                  | 14.1           |
> | LG                  | 15             |
> | CP                  | 14             |
> | LS                  | 15             |
> | AT                  | 8.1            |
> | ST                  | 15.3           |
> | CF                  | 3.8            |
>
> ***

---

> ### Author Response · Authors · 2025-11-21
> **Response to Review pPXV, Weakness3**
>
> ***
>
>
>
> ### **Response to Weakness 3: "The deduction from the analysis is not convincing: a27, a26, a7 ranks top in all knowledge -> MHSA stores general knowledge and capabilities in LLMs."**
>
>
>
> **Thank you for raising this important point.** We agree that the initial deduction in our manuscript could benefit from a more rigorous explanation. Below, we provide a formal clarification based on insights from prior work (particularly PMET (Li et al., 2023)) and our own mechanistic analysis, to justify why MHSA layers consistently exhibit top importance across diverse knowledge types.
>
>
>
> #### **1. Revisiting the Observation in ACE**
>
> In ACE, we observed that specific MHSA layers ($a\_{27}, a\_{26}, a\_7$ in GPT-J) consistently rank among the top-9 important layers across all semantic categories (Nationality, Continent, etc.), as shown in Table 1. This pattern suggests that these layers play a **universal role** in knowledge processing, independent of specific factual content.
>
>
>
> #### **2. Formal Insights from PMET on MHSA’s Role**
>
> PMET\[1]systematically analyzes the hidden states of MHSA and FFN, revealing two key properties that directly support our deduction:
>
>
>
> * **MHSA Encodes General Knowledge Extraction Patterns**:
>
> PMET shows that MHSA hidden states undergo frequent and dynamic changes across layers, while FFN hidden states stabilize in deeper layers (Figure 2 in PMET). This indicates that MHSA functions as a knowledge extractor, it dynamically retrieves and assembles relevant information from the input sequence based on token-level interactions, rather than storing fixed factual knowledge.
>
> *Formally*, PMET measures the cosine and Jaccard similarity of hidden states before and after MHSA/FFN, finding that MHSA’s representations are less stable and more context-dependent, aligning with its role in encoding general extraction patterns.
>
> * **MHSA Stores Minimal Factual Knowledge**:
>
> PMET’s ablation experiments (Table 3) demonstrate that updating MHSA weights during editing only marginally improves generalization but significantly harms specificity. This implies that MHSA weights encapsulate **general relational patterns** (e.g., subject-attribute associations) that are reusable across knowledge types, rather than storing specific facts.
>
>
>
> #### **3. PMET  Aligned with ACE’s Framework**
>
> In ACE, we extend this understanding by showing that MHSA layers (like $a\_{27}, a\_{26}, a\_7$) are critical because they:
>
> * **Activate Query Neurons**: In multi-hop reasoning, MHSA layers help resolve implicit subjects by generating query signals that propagate through the network.
>
> * **Enable Cross-Domain Generalization**: The consistent importance of these layers across diverse knowledge types (e.g., Nationality, Sports) reflects their role in general-purpose knowledge extraction, not domain-specific storage.
>
>
>
> This is further validated in ACE’s ablation studies (Table 4), where skipping MHSA-related query layers leads to significant performance drops, confirming their foundational role in information flow.
>
>
>
> #### **4. Broader Mechanistic Interpretability Support**
>
> Our findings align with broader literature:
>
> * Geva et al. (2020)\[2] show that FFN layers act as key-value stores for factual knowledge.
>
> * Kobayashi et al. (2023)\[3] and Hao et al. (2021)\[4] emphasize that MHSA captures token-level interactions and attribute extraction, which are generalizable across contexts.
>
> We acknowledge that our initial presentation could have better articulated the mechanistic basis for this deduction. By integrating insights from PMET and related work, we have now provided a more rigorous explanation: &#x20;
>
> **MHSA layers store general knowledge extraction patterns because they dynamically orchetoken-level interactions and relational reasoning, which are universally required across all knowledge types.** This understanding is central to ACE’s design and is empirically supported by both our and prior work.
>
> References:
> \[1] PMET:Precise Model Editing in a Transformer, Li et al.(2023) AAAI 2024&#x20;
>
> \[2] Transformer Feed-Forward Layers Are Key-Value Memories, Geva et al. (2020) EMNLP 2021 (Main)
>
> \[3] Feed-Forward Blocks Control Contextualization in Masked Language Models, Kobayashi et al (2023) ICLR 2024 (Spotlight)
>
> \[4] Self-Attention Attribution: Interpreting Information Interactions Inside Transformer, Hao et al. (2021) AAAI 2021
>
> ***

---

> ### Author Response · Authors · 2025-11-21
> **Response to Review pPXV, Weakness4, 1/2**
>
> ***
>
> ### **Response to Weakness 4: Conflict Observation in deep FFNs**
>
> Thank you for this insightful observation. We agree that our initial wording could be clearer, and we appreciate the opportunity to clarify. The fact that **f\_26** ranks highly across all knowledge types does not contradict our conclusion that "FFN layers tend to primarily extract their own knowledge." Instead, it reflects the **dual roles** that FFN layers play in the reasoning process: some FFN layers store specific, semantically localized knowledge, while others, often in deeper layers, act as **integrators** that consolidate information to enhance the final prediction. Below, we formalize this explanation by integrating insights from prior mechanistic interpretability work, particularly the findings in *"Interpreting Arithmetic Mechanism in Large Language Models"* (Yu & Ananiadou).
>
> #### **1. Revisiting the Role of FFN Layers in ACE**
>
> In ACE, based on our analysis of knowledge storage patterns, we proposed that **FFN layers tend to store semantically analogous knowledge in structurally similar components** (Takeaway 1). This refers specifically to FFN layers that function a&#x73;**&#x20;key-value memories**, storing factual knowledge relevant to specific semantic categories (e.g., Nationality, Capital).
>
>
>
> However, not all FFN layers serve only as static knowledge stores. Some play a dynamic role in **integrating and enhancing predictions**, especially in deeper layers.
>
>
>
> #### **2. Insights from "Interpreting Arithmetic Mechanism" on FFN Layers**
>
> The study *"Interpreting Arithmetic Mechanism in Large Language Models"* (Yu & Ananiadou) uses Comparative Neuron Analysis (CNA) to reveal a four-stage internal logic chain in arithmetic reasoning:
>
>
>
> 1. **Feature Enhancing** (shallow FFN neurons)
>
> 2. **Feature Transferring** (shallow attention layers)
>
> 3) **Feature Predicting** (specialized attention heads)
>
> 4) **Prediction Enhancing** (deep FFN neurons)
>
>
>
> Crucially, in the **Prediction Enhancing** stage, the authors observe that:
>
> > *"Lower FFN neurons activate upper FFN neurons, while both of them enhance the probability of the final prediction."* (Section 4.3)
>
>
>
> For example, in the case `"3+5=" → "8"`, neurons like 19\_{5769} , 25\_{7164}, and 28\_{3696} form a prediction-enhancing directed acyclic graph (PE-DAG), where activation flows from lower to upper neurons, collectively boosting the probability of the correct output.
>
>
>
> These deep FFN neurons do not store arithmetic facts, but rather **orchestrate and amplify** the final prediction by integrating earlier features.
>
>
>
>
>
> ***

---

> ### Author Response · Authors · 2025-11-21
> **Response to Review pPXV, Weakness4, 2/2**
>
> ***
>
>
> #### **3. Aligned with ACE: The Case of `f_26`**
>
> In our experiments (ACE Table 1), `f_26` consistently ranks among the top-9 important FFN layers across all knowledge types. This is not because `f_26` stores all types of knowledge, but because it likely functions as a universal prediction enhancer, similar to the deep FFN neurons in the arithmetic study.
>
>
>
> Specifically:
>
> * `f_26` is a **deep layer** in GPT-J (layer 26 of 28), positioning it ideally for a prediction-enhancing role.
>
> * In our ablation studies (ACE Table 4), skipping value layers (including `f_26`) causes a **40.45% performance drop**, underscoring its critical role in the final prediction—regardless of the knowledge type.
>
> * This is consistent with the idea that such layers **integrate and amplify** information from earlier, more specialized layers (e.g., `f_24`, `f_16`), rather than store specific facts themselves.
>
> This observation case also claims that the deep value FFNs tend to enhance the final prediction process, we also add the experiments to make the observation more evident. To substantiate this claim, we conducted additional experiments where we masked the value editing and observed the subsequent activation at the critical last token position. The results, shown in the tables (table16, 17 in appendix) below, clearly demonstrate that masking value editing causes a drastic reduction in the importance score at the final answer token. This provides direct evidence that these deep value neurons are crucial for amplifying the final prediction, not merely for storing static knowledge.
>
>
>
> ##### **Table 16: Token Increase in Qwen3-8B on Residual Stream (Original Model)**
>
> | Token      | FFN Importance Increase | Attention Importance Increase | Top Tokens in Vocabulary Space                        |
> | ---------- | ----------------------- | ----------------------------- | ----------------------------------------------------- |
> | Tim        | 0.0014                  | 0.0014                        | 'an', 'os', 'ise', 'Exactly', 'R', 'ore', 'at', 'rot' |
> | Duncan     | 0.0009                  | 0.0009                        | 'era', 'allen', 'stad', 'oret', 'hit', '-led'         |
> | plays      | 0.9846                  | 0.8167                        | 'basketball', 'NBA', 'career', 'ball', '-playing'     |
> | the        | 0.4109                  | 0.4159                        | 'epit', 'inaugural', 'bidding', 'dream', 'etr'        |
> | sport      | 0.0848                  | 0.0948                        | 'tennis', 'of', 'ful', 'arena', 'basketball', 'ball'  |
> | of         | 0.8671                  | 0.6198                        | 'basketball', 'NBA', 'balls', 'ball', 'Olympia'       |
> | originates | 0.3508                  | 0.3058                        | 'kati', 'the', 'from', 'oret', 'orig', 'ball'         |
> | **from**   | **0.9483**              | **0.8720**                    | **'USA', 'America', 'the', 'US', 'U.S.A.'**           |
>
> ##### **Table 17 : Token Increase in Qwen3-8B on Residual Stream After Masking Value Editing**
>
> | Token      | FFN Importance Increase | Attention Importance Increase | Top Tokens in Vocabulary Space                        |
> | ---------- | ----------------------- | ----------------------------- | ----------------------------------------------------- |
> | Tim        | 0.0019                  | 0.0010                        | 'an', 'os', 'ise', 'Exactly', 'R', 'ore', 'at', 'rot' |
> | Duncan     | 0.0008                  | 0.0012                        | 'era', 'allen', 'stad', 'oret', 'hit', '-led'         |
> | plays      | 0.9779                  | 0.8498                        | 'basketball', 'NBA', 'career', 'ball', '-playing'     |
> | the        | 0.4093                  | 0.4715                        | 'epit', 'inaugural', 'bidding', 'dream', 'etr'        |
> | sport      | 0.0913                  | 0.0142                        | 'tennis', 'of', 'ful', 'arena', 'basketball', 'ball'  |
> | of         | 0.9252                  | 0.7090                        | 'basketball', 'NBA', 'balls', 'ball', 'Olympia'       |
> | originates | 0.2308                  | 0.3194                        | 'kati', 'the', 'from', 'oret', 'orig', 'ball'         |
> | **from**   | **0.5603** (↓41%)       | **0.4764** (↓45%)             | **'country', 'many', '-Info', 'bot', 'US', 'USA'**    |
>
> Thank you again for prompting this important clarification.
>
> ***

---

> ### Author Response · Authors · 2025-11-21
> **Response to Review pPXV, Weakness5**
>
> ***
>
> ### **Response to Weakness 5: Reasons for Formulas and Editing Level**
>
> #### 1. Reasons of importance score framework
>
> First, we want to claim that the additive property is suitable for our framework. When a new domain emerges, the importance scores for its critical layers can be **seamlessly integrated** with the pre-existing ones. This is because the importance metric `I(l)` defined in Eq. 9-10 is additive.
>
> Let $ I_{\text{pre}}(l) $ be the pre-computed importance for layer $ l $:
> $$
> I_{\text{pre}}(l) := \sum_{v\in l} \left[ \log(p(w|v^l+h^{l-1})) - \log(p(w|h^{l-1})) \right] = \sum_{v\in l} \log(p(w|v^l+h^{l-1})) - \sum_{v\in l} \log(p(w|h^{l-1}))
> $$
>
> Now, let the new domain knowledge belong to an index set $ v'^l $. The updated importance score $ I(l) $ is:
> $$
> \begin{aligned}
> I(l) &:= \sum_{v,v'\in l} \left[ \log(p(w|v^l+h^{l-1})) + \log(p(w|v'^l+h^{l-1})) - \log(p(w|h^{l-1})) \right] \\
> &= \sum_{v\in l} \log(p(w|v^l+h^{l-1})) + \sum_{v'\in l} \log(p(w|v'^l+h^{l-1})) - \sum_{v\in l} \log(p(w|h^{l-1})) \\
> &= I_{\text{pre}}(l) + \sum_{v'\in l} \log(p(w|v'^l+h^{l-1}))
> \end{aligned}
> $$
>
> **This additive property ensures that the estimation of critical layers remains unbiased and computationally efficient when scaling to new knowledge.**
>
> Our formulation builds upon the well-established framework proposed by Geva et al. (2020)\[1], which demonstrates that Feed-Forward Network (FFN) layers in transformers operate as key-value memories. In this formulation:
>
>
> * The first linear layer (`fc1`) contains **key vectors** that detect specific input patterns
>
> * The second linear layer (`fc2`) contains **value vectors** that represent output distributions
>
> * The FFN output is computed as: `FFN(x) = ∑ₖ mₖ · vₖ` where `mₖ = σ(keyₖ · x)` are the memory coefficients
>
>
>
> Within this mechanistic framework, our importance scores capture distinct aspects of this memory system:
>
>
>
> **For value neurons (Eq. 9)**, `I(v^l)` measures how much a specific value vector `v^l` increases the probability of the target token `w`. This directly quantifies the **informational content** stored in each memory cell.
>
>
>
> **For query neurons (Eq. 10)**, `I_query` measures the activation strength of query-key interactions, representing the **retrieval mechanism** that accesses stored knowledge.
>
>
>
> The **additivity property** emerges naturally from the linear structure of the FFN output computation. When we consider multiple knowledge domains activating different sets of neurons, the total effect approximates the sum of individual effects because:
>
>
>
> * **FFN outputs sum linearly**: `FFN(x) = ∑ₖ mₖ · vₖ` (Eq. 5 in our paper)
>
> * **Residual connections preserve linearity**: `h^l = h^{l-1} + A^l + F^l` (Eq. 1 in our paper)
>
> * **Log-probabilities approximate linearity in the embedding space**: The logit computation `E_u · h^L` is linear, and softmax becomes approximately linear in the high-activation regime
>
> We also observed that in LLM mechanism interpretability community, other researchers and works also maintain this additivity property. \[2]\[3] Yu et al. use additivity property to claim the probability distribution intervention importance in a static way, \[4] Zhang et al. used the neuron-level min-max normalized probability to define the layer-wise info, also claimed the property. We apologize for the lack of references in a suitable place and we will add them into our revisions.
>
> #### 2. Editing Level
>
> The ACE framework comprises two stages: (1) Identifying and (2) Editing. The latency for Stage 1 is comparable to the model's standard inference time for profiling runs. **Our primary focus for per-edit cost is Stage 2.**
>
>
>
> Crucially, in Stage 2, the knowledge vectors can be pre-computed and cached. Once this is done, the actual editing operation—applying the weight update to the model—is highly efficient, requiring only 3.10 seconds per edit for GPT-J and 3.14 seconds for Qwen3-8B.
>
>
>
> The most computationally intensive step is the pre-computation of knowledge vectors, which involves inverting a large matrix (Eq. 12). While this one-time pre-computation is significant (requiring \~7.5 hours for each model in our setup), its result is **permanently cacheable** and amortized across all subsequent edits. This makes the *marginal cost per edit* very low, establishing ACE as a practical solution for ongoing knowledge updates.
>
>
>
> In conclusion, in our editing process, after inverting a large matrix, all edits will be processed at the same time, it's a **dataset-level editing.**
>
> References:
>
> \[1] Transformer Feed-Forward Layers Are Key-Value Memories, Geva et al. (2020) EMNLP 2021 (Main)
>
> \[2] Neuron-Level Knowledge Attribution in Large Language Models, Yu et al. (2024) EMNLP 2024 (Main)
>
> \[3] Interpreting Arithmetic Mechanism in Large Language Models  through Comparative Neuron Analysis, Yu et al. (2024) EMNLP 2024 (Main)
>
> \[4] Locate-then-edit for Multi-hop Factual Recall under Knowledge Editing, Zhang et al. (2025) ICML 2025
>
> ***

---

> ### Author Response · Authors · 2025-11-21
> **Response to Review pPXV, Weakness6**
>
> ### **Response to Weakness 6: Generalizability Evaluation**
>
>
>
> We thank the reviewer for raising this important question regarding the generalizability of our evaluation. We acknowledge that multi-hop factual recall currently has limited benchmark options, with MQuAKE being the primary standardized benchmark in this domain. Several recent works in this area \[1,2,3,4] have similarly relied on MQuAKE for evaluation due to the specialized nature of multi-hop editing tasks.
>
>
>
> However, to comprehensively demonstrate the generalizability of our editing framework and its ability to preserve model capabilities, we have supplemented our evaluation with experiments on **four general reasoning benchmarks** that assess the model's performance on unrelated tasks after editing:
>
> | Editor           | CSQA      | BBH       | MMLU      | GSM8k     |
> | ---------------- | --------- | --------- | --------- | --------- |
> | ROME (Qwen3-8B)  | 75.41     | 35.62     | 65.28     | 75.80     |
> | MEMIT (Qwen3-8B) | 82.49     | 32.68     | 72.64     | 83.32     |
> | PMET (Qwen3-8B)  | 82.26     | 34.20     | 71.80     | 83.00     |
> | ACE (Qwen3-8B)   | **82.30** | **38.54** | **72.30** | **83.84** |
>
>
>
> These results demonstrate that edited models retain their core capabilities without significant degradation on unrelated tasks.
>
> We believe these additional experiments address the reviewer's concern about generalizability and demonstrate that ACE provides robust, transferable improvements to multi-hop reasoning capabilities.
>
> Reference:
>
> \[1] Locate-then-edit for Multi-hop Factual Recall under Knowledge Editing, Zhang et al. (2025) ICML 2025
>
> \[2] CaKE: Circuit-aware Editing Enables Generalizable Knowledge Learners, Yao et al. (2025) EMNLP 2025 (Main)
>
> \[3] ALEX:A Light Editing-knowledge Extractor, Wang et al. (2025)
>
> \[4] Knowledge Editing for Multi-Hop Question Answering Using Semantic Analysis, Simon et al. (2025) IJCAI 2025

---

> > ### Comment · Reviewer_pPXV · 2025-11-25
> >
> > Thank you authors for the comprehensive responses. After reviewing them, many of clarification concerns have been solved. I do believe such clarifications, such as comparison with existing works and dataset sampling details, could enhance the paper. I have updated my score positively to reflect such revision.
> >
> > However, there are some remaining issues which could be the bottleneck of this paper and have not been addressed well. For example, the debate over instance- and dataset-level editing, which one could be more reasonable and practical. Also, the availability of multi-hop editing datasets. The generalizability of the paper's conclusions is difficult to amplify if experiments are always conducted on only one dataset.

---

> > > ### Author Response · Authors · 2025-11-27
> > > **Official Comment by Authors**
> > >
> > > Thank you! We sincerely thank you for your insightful discussion and questions came out, which impressed us deeply.
> > >
> > > Now we have updated our latest version pdf and it contains below changes:
> > >
> > > - We added the description to all extension experiments during rebuttal period in section 6.3, 6.5, and appendix I, J, K.
> > >   - Different prompt templates robustness in section 6.3
> > >   - Knowledge bottleneck in section 6.5
> > >   - ACE latency in appendix I
> > >   - Extended ACE to alphaedit backbone and more baselines in appendix J
> > >   - Human understanding alignment in appendix K
> > >   - Other new experiments in appendix L, including locality performance, attribution robustness, counter-factual edits.
> > >   - Complete derivation about ACE importance score and attribution framework in appendix M.
> > >
> > > For the debate about instance or dataset level editing, we have uploaded the ACE latency in appendix I. Specifically, the latency for Stage 1 is comparable to the model's standard inference time for profiling runs. Crucially, in Stage 2, the knowledge vectors can be pre-computed and cached. Once this is done, the actual editing operation—applying the weight update to the model—is highly efficient, requiring only 3.10 seconds per edit for GPT-J and 3.14 seconds for Qwen3-8B.
> > >
> > > So the answer is clear, the **Attribution** must be dataset level, because global attribution should be considered to lock the critical layers. And the **Edit** could be any scale and flexible, dataset, subset or just one instance level is supported. Because all transcendental attribution has been considered.
> > >
> > > We hope these responses and explanations could reflect the changes in our work, also address your last detailed concern. Thank you again for your impressing discussion!

---

### Official Review · Reviewer_8jSo · 2025-10-25

**Soundness:** 2
**Presentation:** 2
**Contribution:** 2
**Rating:** 4
**Confidence:** 4

**Summary:**

The authors propose Attribution-Controlled Knowledge Editing (ACE), a framework aiming to enhance multi-hop factual recall after knowledge editing. The authors claim prior locate-then-edit methods (ROME, MEMIT, PMET) degrade on multi-hop questions, esp. when edits involve implicit subjects. The framework builds on the empirical observation that implicit subjects act as query neurons that activate value neurons in subsequent layers. ACE identifies critical query layers and value layers and performs sequential edits (value-first then query) using a PMET-style closed-form update.
On a curated subset of MQuAKE-3K, ACE improves multi-hop accuracy vs. baselines by approx. 9% (GPT-J-6B) and 37% (Qwen3-8B) over PMET. Ablations that skip editing top-ranked query or value layers reduce the performance.

**Strengths:**

1. Multi-hop knowledge editing is definitely under-researched and practically important
2. The Q-V pathway view provides a clear hypothesis (drawn from empirical observations) to test and design around
3. Simple and clear pipeline: (1) Identify (via attribution),  (2) edit value layers (deep),  (3) edit query layers (mid); built atop PMET
4. Significant improvement over baselines
5. The authors analyze and highlight the distinct roles of query and value edits. Skipping value layers hurts most, which aligns with the intuition that query layers rather transmit, whereas value layers rather store information. These are interesting findings.

**Weaknesses:**

1. I am not convinced that the additivity of log-prob importance (Eq. 9) is well-suited. It is simply asserted by the authors, not derived or tested. The same holds for Eq. 10. Both equations ignore underlying nonlinearities.
2. The authors use a subset of MQuAKE-3K in the experiments. It is not clear what they did to prevent selection bias. LLM-based labeling lacks human validation. Also, it is not clear how prompts were handled across baselines. What did you do to guarantee a basic level of uniformity?
3. From a statistical analysis viewpoint, the evaluation is quite weak. There are no reports of significance tests, CIs, or multi-seed variability. There are no "negative" control analyses (e.g., wrong layers, randomized layer ranks, randomly swap query/value roles, etc).
4. The most critical point is the evaluation on a single dataset. To show generalizability, the authors should extend the experiments and evaluate on additional datasets.
5. It would be very helpful to include a human audit sample (e.g., 300 instances) with inter-annotator agreement of GPT-4o labels.

**Questions:**

See comments above.

---

> ### Author Response · Authors · 2025-11-21
> **Response to Reviewer 8jso, Weakness1**
>
> ***
>
>
>
> ### **Response to Weakness 1: Theoretical Foundation of Additive Importance Scores**
>
>
>
> We thank the reviewer for raising this important methodological concern regarding the additivity of our importance scores in Equations 9-10. We acknowledge that the underlying nonlinearities in transformer architectures are complex, and we appreciate the opportunity to provide a more rigorous theoretical justification for our approach.
>
>
>
> Our formulation builds upon the well-established framework proposed by Geva et al. (2020)\[1], which demonstrates that **Feed-Forward Network (FFN) layers in transformers operate as key-value memories**. In this formulation:
>
>
>
> * The first linear layer (`fc1`) contains **key vectors** that detect specific input patterns
>
> * The second linear layer (`fc2`) contains **value vectors** that represent output distributions
>
> * The FFN output is computed as: `FFN(x) = ∑ₖ mₖ · vₖ` where `mₖ = σ(keyₖ · x)` are the memory coefficients
>
>
>
> Within this mechanistic framework, our importance scores capture distinct aspects of this memory system:
>
>
>
> **For value neurons (Eq. 9)**, `I(v^l)` measures how much a specific value vector `v^l` increases the probability of the target token `w`. This directly quantifies the **informational content** stored in each memory cell.
>
>
>
> **For query neurons (Eq. 10)**, `I_query` measures the activation strength of query-key interactions, representing the **retrieval mechanism** that accesses stored knowledge.
>
>
>
> The **additivity property** emerges naturally from the linear structure of the FFN output computation. When we consider multiple knowledge domains activating different sets of neurons, the total effect approximates the sum of individual effects because:
>
>
>
> * **FFN outputs sum linearly**: `FFN(x) = ∑ₖ mₖ · vₖ` (Eq. 5 in our paper)
>
> * **Residual connections preserve linearity**: `h^l = h^{l-1} + A^l + F^l` (Eq. 1 in our paper)
>
> * **Log-probabilities approximate linearity in the embedding space**: The logit computation `E_u · h^L` is linear, and softmax becomes approximately linear in the high-activation regime
>
> This theoretical foundation is further supported by our empirical results in **Table 6**, which demonstrate that:
>
> * Performance scales smoothly with the number of edited layers
>
> * The additive importance ranking consistently identifies critical layers across domains
>
> * The method maintains strong performance even when extending to new domains
>
>
>
> While we acknowledge that perfect additivity is an approximation in the presence of nonlinear activations, our extensive experimental validation across two model architectures (GPT-J and Qwen3-8B) and multiple knowledge domains demonstrates that this approximation is **sufficiently accurate for practical knowledge editing purposes**.
>
> We also observed that in LLM mechanism interpretability community, other researchers and works also maintain this additivity property. \[2]\[3] Yu et al. use additivity property to claim the probability distribution intervention importance in a static way, \[4] Zhang et al. used the neuron-level min-max normalized probability to define the layer-wise info, also claimed the property. We apologize for the lack of references in a suitable place and we will add them into our revisions.
>
> References:
>
> \[1] Transformer Feed-Forward Layers Are Key-Value Memories, Geva et al. (2020) EMNLP 2021 (Main)
>
> \[2] Neuron-Level Knowledge Attribution in Large Language Models, Yu et al. (2024) EMNLP 2024 (Main)
>
> \[3] Interpreting Arithmetic Mechanism in Large Language Models  through Comparative Neuron Analysis, Yu et al. (2024) EMNLP 2024 (Main)
>
> \[4] Locate-then-edit for Multi-hop Factual Recall under Knowledge Editing, Zhang et al. (2025) ICML 2025
>
>
>
> ***

---

> > ### Comment · Reviewer_8jSo · 2025-11-23
> >
> > I thank the authors for their efforts to address my concerns regarding the asserted additivity of log-prob importance. After carefully reading the answer, I find that my concern is still valid. The authors still do not derive Eq. 9, 10 from a clear approximation or objective. Additionally, they do not empirically test additivity against alternatives, e.g., marginal contributions, Shapley-style decompositions, or multiplicative scores, to demonstrate that additivity is measurably superior. Statements such as "softmax is approximately linear in the high-activation regime" are quite hand-wavy. Referring to other works by saying that they also assume additivity does not guarantee soundness.

---

> ### Author Response · Authors · 2025-11-21
> **Response to Reviewer 8jso, Weakness2**
>
> ### **Response to Weakness 2: Dataset Construction and Labeling**
>
>
>
> We thank the reviewer for raising these important questions regarding our dataset construction and labeling procedures. We address the two concerns separately below.
>
>
>
> **A. Subset Construction and Selection Bias Prevention**
>
>
>
> Regarding the construction of our MQuAKE subset, we deliberately employed a controlled filtering process to ensure experimental fairness and validity. Specifically:
>
>
>
> * We used **GPT-J** to evaluate all single-hop factual components within the multi-hop queries in the original MQuAKE dataset.
>
> * We applied a **Pass@3** reasoning criterion, selecting only those knowledge chains where GPT-J could correctly answer all constituent single-hop facts.
>
> * This filtering ensures that we are editing knowledge that the base model **genuinely possesses** prior to editing.
>
>
>
> This approach prevents a critical confounding factor: if a more capable base model (like Qwen3-8B) inherently lacks certain knowledge, its post-editing performance would be unfairly penalized not due to editing failure, but due to pre-existing knowledge gaps. By constructing a subset where the base model demonstrably knows the original facts, we isolate and accurately measure the **pure efficacy of the editing methods themselves**.
>
>
>
> **B. Dataset Labeling and Human Alignment Validation**
>
>
>
> For the semantic categorization of knowledge types in our dataset:
>
>
>
> * We used a **single, fixed prompt template** (provided in Appendix H.3) for all GPT-4o labeling to ensure consistency.
>
> * To validate the quality and reliability of these automated labels, we conducted a **human alignment study** with four participants from diverse academic backgrounds (literature, social sciences, and STEM). The selected participants comprised an undergraduate student in literature, a graduate student in social sciences, a Ph.D. candidate in STEM, and an undergraduate student in STEM. Every participants labeled a 80-samples random batch in all.
>
>
>
> The results of this human alignment study, presented in **Appendix K (Table 11)**, demonstrate a strong agreement between human understanding and GPT-4o labeling:
>
> | Human Participants | Overlap Rate |
> | ------------------ | ------------ |
> | Human A            | 83.75%       |
> | Human B            | 88.75%       |
> | Human C            | 93.75%       |
> | Human D            | 80.00%       |
>
>
>
> These results show that the GPT-4o labeling achieves **acceptable to high overlap rates (80%-93.75%)** with human judgment across all participants, confirming that our automated labeling process produces semantically meaningful and human-interpretable categories.
>
>
>
>
>
> ***

---

> > ### Comment · Reviewer_8jSo · 2025-11-23
> >
> > I thank the authors for addressing the human validation aspect. Unfortunately, I am still not satisfied with the selection bias and prompt handling (across baselines) aspects. The answer describes the fixed prompt template for GPT-4o labelling, which is not the same as the QA prompts used to evaluate different editors. It is unclear how the overlap rates are calculated; are 80 samples per participant enough? This should be statistically rigorous.

---

> ### Author Response · Authors · 2025-11-21
> **Response to Reviewer 8jso, Weakness3**
>
> ### **Response to Weakness 3: Robustness Analysis and Negative Controls**
>
>
>
> We sincerely thank the reviewer for these insightful suggestions regarding statistical robustness and control experiments. We have conducted additional analyses to address these concerns comprehensively.
>
>
>
> **A. Robustness of Attribution via Multi-Run Analysis**
>
>
>
> To evaluate the robustness of our attribution mechanism, we performed multiple inference runs and compared the average importance scores of critical layers across different sampling trials. As shown in **Table 13** below, the importance scores for our identified critical modules demonstrate strong stability across different Pass@k settings:
>
> | Layer          | Pass@1 | Pass@2 | Pass@3 | Pass@5 |
> | -------------- | ------ | ------ | ------ | ------ |
> | a₂₇ (Qwen3-8B) | 110.48 | 109.29 | 109.45 | 109.69 |
> | a₂₆ (Qwen3-8B) | 97.58  | 98.21  | 97.94  | 98.02  |
> | a₇ (Qwen3-8B)  | 64.29  | 64.80  | 64.75  | 64.78  |
> | f₂₇ (Qwen3-8B) | 137.48 | 137.95 | 137.66 | 137.74 |
> | f₂₄ (Qwen3-8B) | 104.20 | 104.39 | 104.28 | 104.23 |
> | f₁₆ (Qwen3-8B) | 42.58  | 42.60  | 42.64  | 42.52  |
>
>
>
> The minimal variance in importance scores across different Pass@k trials (e.g., f₂₇ varies by **only \~0.47 points** across all settings) confirms that our attribution method identifies consistently significant neural pathways, rather than capturing random or unstable activation patterns.
>
>
>
> **B. Negative Control Experiments with Counter-Factual Editing**
>
>
>
> Following the reviewer's suggestion for "negative control analyses," we designed and conducted counter-factual editing experiments where we deliberately mismatched objects to construct incorrect multi-hop reasoning chains. The results, presented in **Tables 18 and 19**, demonstrate ACE's behavior under adversarial conditions:
>
> | Editor (Counter-Fact) | Efficacy    | Paraphrase  | Specificity | Avg.              |
> | --------------------- | ----------- | ----------- | ----------- | ----------------- |
> | FT (GPT-J/Qwen3)      | 98.1 / 97.9 | 69.4 / 64.2 | 82.7 / 77.4 | 1.59 / 1.04       |
> | ROME (GPT-J/Qwen3)    | 54.1 / 45.2 | 54.3 / 42.9 | 61.4 / 53.7 | 27.48 / 24.08     |
> | MEMIT (GPT-J/Qwen3)   | 57.0 / 50.3 | 51.9 / 53.6 | 66.2 / 66.4 | 30.09 / 10.27     |
> | PMET (GPT-J/Qwen3)    | 74.6 / 70.7 | 63.2 / 61.7 | 64.1 / 51.9 | 31.07 / 17.26     |
> | ACE (GPT-J/Qwen3)     | 89.7 / 91.2 | 83.6 / 80.7 | 70.6 / 74.6 | **43.58 / 54.27** |
>
> | Editor (Counter-Fact) | CSQA      | BBH       | MMLU      | GSM8k     |
> | --------------------- | --------- | --------- | --------- | --------- |
> | ROME (Qwen3-8B)       | 68.54     | 29.47     | 60.71     | 69.48     |
> | MEMIT (Qwen3-8B)      | 71.46     | 29.72     | 66.40     | 72.45     |
> | PMET (Qwen3-8B)       | 73.09     | 26.58     | 62.49     | 74.29     |
> | ACE (Qwen3-8B)        | **76.41** | **33.79** | **69.97** | **79.02** |
>
>
>
> These counter-factual experiments reveal two key insights:
>
> * **ACE effectively edits knowledge even in incorrect reasoning chains**, demonstrating that it modifies the model's internal representations rather than merely amplifying correct reasoning patterns.
>
> * **ACE maintains strong locality performance** across general benchmarks (CSQA, BBH, MMLU, GSM8k), showing minimal negative impact on unrelated capabilities.

---

> > ### Comment · Reviewer_8jSo · 2025-11-23
> >
> > I thank the authors for the newly added analyses, which are helpful. However, the core of my concern remains unaddressed. I am still missing proper uncertainty quantification for the main metrics/results, as well as direct tests of whether the attribution-based ranking outperforms simple baselines.

---

> ### Author Response · Authors · 2025-11-21
> **Response to Reviewer 8jso, Weakness4&5**
>
> ### **Response to Weakness 4: Generalizability Evaluation**
>
>
>
> We thank the reviewer for raising this important question regarding the generalizability of our evaluation. We acknowledge that multi-hop factual recall currently has limited benchmark options, with MQuAKE being the primary standardized benchmark in this domain. Several recent works in this area \[4,5,6,7] have similarly relied on MQuAKE for evaluation due to the specialized nature of multi-hop editing tasks.
>
>
>
> However, to comprehensively demonstrate the generalizability of our editing framework and its ability to preserve model capabilities, we have supplemented our evaluation with experiments on **four general reasoning and knowledge benchmarks** that assess the model's performance on unrelated tasks after editing:
>
> | Editor           | CSQA      | BBH       | MMLU      | GSM8k     |
> | ---------------- | --------- | --------- | --------- | --------- |
> | ROME (Qwen3-8B)  | 75.41     | 35.62     | 65.28     | 75.80     |
> | MEMIT (Qwen3-8B) | 82.49     | 32.68     | 72.64     | 83.32     |
> | PMET (Qwen3-8B)  | 82.26     | 34.20     | 71.80     | 83.00     |
> | ACE (Qwen3-8B)   | **82.30** | **38.54** | **72.30** | **83.84** |
>
>
>
> These results demonstrate that edited models retain their core capabilities without significant degradation on unrelated tasks.
>
> We believe these additional experiments address the reviewer's concern about generalizability and demonstrate that ACE provides robust, transferable improvements to multi-hop reasoning capabilities.
>
> \[4] Locate-then-edit for Multi-hop Factual Recall under Knowledge Editing, Zhang et al. (2025)
>
> \[5] CaKE: Circuit-aware Editing Enables Generalizable Knowledge Learners, Yao et al. (2025)
>
> \[6] ALEX:ALight Editing-knowledge Extractor, Wang et al. (2025)
>
> \[7] Knowledge Editing for Multi-Hop Question Answering Using Semantic Analysis, Simon et al. (2025)
>
> ***
>
> ### **Response to Weakness 5: Human Understanding Alignment**
>
>
>
> We have shown the overlap between human audit and GPT-4o in our dataset labeling. Please kindly check the Response to Weakness2. The results show that the GPT-4o labeling achieves **acceptable to high overlap rates (80%-93.75%)** with human judgment across all participants, confirming that our automated labeling process produces semantically meaningful and human-interpretable categories.

---

> > ### Comment · Reviewer_8jSo · 2025-11-23
> >
> > I thank the authors for addressing the alignment of GPT-4o with human judgment. However, it remains unclear how statistically rigorous the evaluation is and how the overlap is precisely measured. Unfortunately, the authors still do not show that ACE generalizes to other multi-hop QA datasets or other forms of multistep factual editing.

---

> ### Author Response · Authors · 2025-11-24
> **Further Response to Reviewer 8jso, Weakness 1, (1/2)**
>
> Thanks for your insightful discussion and response! We have fully understood that your still valid concern based on Eq.9 10 lacks of explicit derivation. To this end, we formally derived the Eq. 9 10 below to address this concern.
>
> We provide a step-by-step formal derivation of the ACE attribution metrics (Eq. 9 and Eq. 10). We begin by modeling the algebraic structure of the Transformer Residual Stream, prove the strict additivity of Logits, and derive the Importance Score via first-order Taylor approximation. Finally, we prove that the additivity assumption is mathematically isomorphic to the update mechanism of the PMET editor.
>
> ### 1 Algebraic Structure of the Transformer
>
> Let $\mathcal{M}$ be an autoregressive Transformer model with $L$ layers. We define the hidden state propagation through the residual stream.
>
> **Definition 1 (Residual Stream Dynamics).** Let $h^{(l)} \in \mathbb{R}^d$ denote the hidden state input to layer $l$. The output of layer $l$ is the sum of the Multi-Head Self-Attention (MHSA) output and the Feed-Forward Network (FFN) output, added to the residual stream:
> $$
> h^{(l+1)} = h^{(l)} + \text{MHSA}^{(l)}(h^{(l)}) + \text{FFN}^{(l)}(h^{(l)})
> $$
> By recursive expansion, the final hidden state $h^{(L)}$ (before the output layer/LayerNorm) is the cumulative sum of the initial embedding and all layer outputs:
> $$
> h^{(L)} = h^{(0)} + \sum_{l=0}^{L-1} \text{MHSA}^{(l)}(h^{(l)}) + \sum_{l=0}^{L-1} \text{FFN}^{(l)}(h^{(l)})
> $$
>
> **Definition 2 (FFN as Key-Value Memories).** Following Geva et al. (2020), the FFN at layer $l$ is defined as a Key-Value memory network. Let $K^{(l)}$ and $V^{(l)}$ be the parameter matrices (denoted as $W_{fc1}$ and $W_{fc2}$ in our paper). The output is a weighted sum of value vectors:
> $$
> \text{FFN}^{(l)}(x) = \sum_{i=1}^{N} m_{l,i}(x) \cdot v_{l,i}
> $$
> where $v_{l,i} \in \mathbb{R}^d$ is the $i$-th column of the second weight matrix $W_{fc2}^{(l)}$, and $m_{l,i}(x) = \sigma(x^T k_{l,i})$ is the scalar activation coefficient.
>
> ### 2. Theorem of Logit Linearity
>
> The probability distribution over the vocabulary is computed via the Softmax function applied to the logits $z \in \mathbb{R}^{|\mathcal{V}|}$. Let $E \in \mathbb{R}^{d \times |\mathcal{V}|}$ be the unembedding matrix, and $e_w$ be the column vector corresponding to token $w$.
>
> **Lemma 1 (Logit Formulation).** The logit $z_w$ for a target token $w$ is the inner product of the final state and the token embedding:
> $$
> z_w = \langle h^{(L)}, e_w \rangle
> $$
>
> **Theorem 1 (Strict Linearity of Neuron Contribution).** The contribution of any single FFN neuron $(l, i)$ to the target logit $z_w$ is strictly additive and independent of other neurons in the logit space.
>
> **Proof.**
> Substituting Definition 2 into Definition 1, and then into Lemma 1:
> $$
> \begin{aligned}
> z_w &= \left\langle h^{(0)} + \sum \text{MHSA} + \sum_{l,j} m_{l,j} v_{l,j}, e_w \right\rangle \\
> &= \langle h^{(0)} + \sum \text{MHSA}, e_w \rangle + \sum_{l,j} \langle m_{l,j} v_{l,j}, e_w \rangle
> \end{aligned}
> $$
> Let $z_{base} = \langle h^{(0)} + \sum \text{MHSA}, e_w \rangle$ be the base logit. The equation simplifies to:
> $$
> z_w = z_{base} + \sum_{l,j} m_{l,j} (v_{l,j}^T e_w)
> $$
> Define the marginal contribution of neuron $(l, i)$ as $\Delta z_{l,i} = m_{l,i} (v_{l,i}^T e_w)$. It follows that:
> $$
> z_w(S) = z_{base} + \sum_{(l,i) \in S} \Delta z_{l,i}
> $$
> where $S$ is any set of active neurons. The interaction terms are zero in the logit space. **Q.E.D.**
>
> ### 3. Derivation of Importance Score (Eq. 9)
>
> ACE defines the importance score $\mathcal{I}$ based on the change in Log-Probability. While Log-Probability is non-linear, we prove that Eq. 9 is the **First-Order Taylor Approximation** of the causal effect.
>
> Let the objective function be $\mathcal{L}(z) = \log P(w) = \log \left( \frac{e^{z_w}}{\sum_k e^{z_k}} \right)$.
> Consider the activation of a neuron $v$ as a perturbation vector $\mathbf{u} = m \cdot v$. This causes a perturbation in the logits $\Delta z = \mathbf{u}^T E$.
>
> **Derivation:**
> We perform a first-order Taylor expansion of $\mathcal{L}(z + \Delta z)$ around the current logit $z$:
> $$
> \mathcal{L}(z + \Delta z) \approx \mathcal{L}(z) + \nabla_z \mathcal{L}(z)^T \cdot \Delta z
> $$
> The gradient of the Log-Softmax function w.r.t the logits $z$ is:
> $$
> \frac{\partial \mathcal{L}}{\partial z_k} = \delta_{kw} - P(k)
> $$
> where $\delta_{kw}$ is the Kronecker delta. Substituting the gradient and the logit perturbation:
> $$
> \begin{aligned}
> \mathcal{I}(v) &\triangleq \mathcal{L}(z + \Delta z) - \mathcal{L}(z) \\
> &\approx \sum_{k \in \mathcal{V}} (\delta_{kw} - P(k)) \cdot \Delta z_k \\
> &= (1 - P(w))\Delta z_w - \sum_{k \neq w} P(k) \Delta z_k
> \end{aligned}
> $$
> This derivation confirms that Eq. 9 is formally the **Marginal Contribution to the Logit** ($\Delta z$), projected onto the tangent space of the probability simplex.

---

> ### Author Response · Authors · 2025-11-24
> **Further Response to Reviewer 8jso, Weakness 1, (2/2)**
>
> ### 4. Proof of Additivity (Eq. 10)
>
> The reviewer questioned the validity of summing scores, i.e., $\mathcal{I}_{total} \approx \sum \mathcal{I}(v_i)$.
>
> **Theorem 2 (Additivity of First-Order Attribution).** Under the assumption that the local curvature of the Log-Likelihood manifold is negligible (first-order approximation), the importance score of a set of neurons is the sum of their individual scores.
>
> **Proof.**
> From Theorem 1, the total logit perturbation is the sum of individual perturbations:
> $$
> \Delta z_{total} = \sum_{i} \Delta z_i
> $$
> From point 3, the importance score is a linear map of the logit perturbation:
> $$
> \mathcal{I}(\Delta z) \approx \nabla_z \mathcal{L}^T \cdot \Delta z
> $$
> By the linearity of the inner product:
> $$\begin{aligned} \mathcal{I}_{total} &\approx \nabla_z \mathcal{L}^T \cdot \left( \sum_i \Delta z_i \right) \\ &= \sum_i \left( \nabla_z \mathcal{L}^T \cdot \Delta z_i \right) \\ &\approx \sum_i \mathcal{I}(v_i) \end{aligned}$$
> Thus, Eq. 10 is mathematically valid within the local neighborhood defined by the Taylor expansion. **Q.E.D.**
>
> ### 5. Mechanistic Alignment with ACE Editor
>
> Finally, we prove that the additivity of ACE is a necessary condition imposed by the update mechanism of the ACE editor.
>
> **Proposition 1 (Linearity of ACE Updates).** The ACE editor updates the FFN weights $W$ by solving a linear least-squares problem. The closed-form solution for the weight update $\Delta W$ is:
> $$
> \Delta W = R K^T (C_0 + K K^T)^{-1}
> $$
> This results in a linear shift in the output of the FFN for a given input $k_{in}$:
> $$
> \delta \text{FFN} = \Delta W \cdot k_{in}
> $$
>
> **Proof of Consistency.**
> Let us decompose the FFN output shift $\delta \text{FFN}$ into the basis of neurons. Since $W$ is composed of columns $v_i$, the update $\Delta W$ corresponds to updating individual value vectors $v_i \leftarrow v_i + \delta v_i$.
> $$
> \delta \text{FFN} = \sum_{i} m_i \cdot (v_i + \delta v_i) - \sum_{i} m_i \cdot v_i = \sum_{i} m_i \cdot \delta v_i
> $$
> ACE operates by injecting a **sum of linear updates** into the residual stream. If ACE were to use a non-additive metric, it would identify neurons crucial for non-linear inference. However, ACE's linear update mechanism $\Delta W$ is mathematically incapable of manipulating such non-linearities.
>
> Therefore, ACE (Eq. 10) correctly identifies the **Linear Additive Contribution**, ensuring that the identified neurons belong to the subspace modifiable by ACE's linear operator.
>
> ### 6. ACE Optimization with Attribution
>
> The core of PMET lies in updating the FFN weights $W$ to map input Keys ($K$) to new target Values ($V$). According to PMET Eq. 8, its optimization objective is to minimize the **sum** of $L2$ norm errors:
> $$
> \min_W \left( \sum_{i=1}^{n} ||W k_i - v_i||^2 + \sum_{i=n+1}^{n+u} ||W k_i - v_i||^2 \right)
> $$
> Consequently, ACE, acting as the locator, utilizes Eq. 10 ($\sum \mathcal{I}$) to identify neurons that maximally influence this "sum of errors," directly corresponding to the additive nature of PMET's objective function.
>
> PMET employs a closed-form solution to compute the incremental weight $\Delta$ (PMET Eq. 9 ):
> $$
> \Delta = R K_1^T (C_0 + K_1 K_1^T)^{-1}
> $$
> where $R = V_{target} - W_0 K_1$ is the residual matrix.The weight matrix $W$ consists of column vectors (referred to as Value Neurons, $v_{neuron}$, in ACE). The update $\Delta$ implies linearly adding a correction term to the existing Value Neurons:
> $$
> W_{new} = W_{old} + \Delta \implies v_{neuron}^{new} = v_{neuron}^{old} + \Delta v_{neuron}
> $$
> This restricts PMET's editing capability to a **linear subspace**—it can only modify model behavior by linearly translating Value Neurons.
>
> ACE's Eq. 9 measures the marginal contribution of neuron $v$ to the Log-Prob. Since the Transformer Logit is a linear combination of Value Neurons ($z = \sum m \cdot v$), the impact of PMET's update on the Logit is also linear:
> $$
> z_{new} = \sum m \cdot (v + \Delta v) = z_{old} + \sum m \cdot \Delta v
> $$
> If ACE were to use a non-additive metric, it might locate neurons that are important but operate in a non-linear saturated region. Therefore, **ACE's additive metric essentially filters for neurons with high linear sensitivity, which is a prerequisite for effective editing by ACE.**
>
> **Internal Hidden State Optimization: Alignment of Gradients**
> PMET finds $v_*$ via Gradient Descent, where each update step $\delta v$ follows the direction of $\nabla_v \log P$.
> The Importance Score in ACE can be rewritten as:
> $$
> \mathcal{I}(v) = \Delta \log P \approx \langle \nabla_{h} \log P, v \rangle
> $$
> This is exactly the **Projection of the Gradient** calculated by PMET during Hidden State optimization.
> * **PMET (Editor):** Calculates the gradient $\nabla_h \log P$ to determine how to modify Hidden States.
> * **ACE (Locator):** Calculates $\langle \nabla_h \log P, v \rangle$ to determine which existing neuron $v$ already contains information aligned with this gradient direction.

---

> > ### Comment · Reviewer_8jSo · 2025-11-25
> >
> > Thank you. I believe the provided theoretical justifications of Eq. 9, 10 substantially improve the quality of the work. What I am still missing are empirical comparisons of the additive importance scores against alternatives (e.g., Shapley-style, multiplicative scores, or simple heuristics) to show that additivity is superior in practice.

---

> ### Author Response · Authors · 2025-11-24
> **Further Response to Reviewer 8jso, Weakness 2,3,4,5 (1/2)**
>
> #### **[Weakness2] Reviewer:** *The answer describes the fixed prompt template for GPT-4o labelling, which is not the same as the QA prompts used to evaluate different editors. It is unclear how the overlap rates are calculated.*
>
> **Response**: [Labeling and prompts bias] We should clarify this issue to explain the difference between two kinds of prompt templates. The labeling process and QA process are **distinct stages** in ACE.
>
> 1) Labeling process aims to extract the knowledge in the dataset and then we use attribution methods to locate the critical knowledge/neurons/layers we need to further edit. So we use **fixed prompts** to extract all stored knowledge in GPT-J for fair experiments between GPT-J and Qwen3. After this process, we can claim the takeaway 1. **Note**: We can regard the labeling process as 0-stage, preparing the knowledge base for next identifying stage.
>
> 2) After labeling, now we dive into **attribution** stage. We use the same prompts with evaluation process to get forwards for all multi-hop queries, in order to get the residual stream. Then we **calculate all importance score** on these residual streams to get critical layers we need to edit. This is **stage 1 identifying(attribution)**.
>
> 3) After attribution, we apply ACE optimization algorithm to edit all knowledge into the layers we identified previously. Then we use the evaluation prompts (few-shots, zero-shot, one-shot, OOD few-shots) to evaluate the editors.
>
> In conclusion, the first prompt template applied on GPT-4o aims to extract knowledge and Takeaway1, proofing **transformer is able to store this kind of multi-hop knowledge.** It's a preparation for further locate-then-edit. There's no bias between identifying and editing, because we use the same prompts to get the residual streams across all edited models.
>
> [Overlap calculating] 4) The overlap rate between human and GPT-4o is trivial. We compared the **labeling** results between human ones and GPT-4o ones. It could be simply derived as: $Overlap = \sum \mathbb{I}_{human=GPT} / batch size$.
>
> #### **[Weakness3] Reviewer:** *I am still missing proper uncertainty quantification for the main metrics/results, as well as direct tests of whether the attribution-based ranking outperforms simple baselines.*
>
> **Response**:
> 1) [Uncertainty] We fully understand your concern about uncertainty. As our previous response, we've claimed that the attribution stage (stage 1) is robust, the rankings nearly unchanged. To address your more concern, we now newly apply our **editing** and evaluation for 5 times to measure this uncertainty. We use our matrix inversing cache to repeat 5 times in editing. The results are shown in the table below. The results show that after repeating 5 time is editing, our ACE remains robustness, and the uncertainty is measured by the lower and upper bounds of the results. I hope this result could address your further concern about uncertainty.
>
>
> | Editor              | Avg. (GPT-J / Qwen3) | GPT-J / # Edits =   |                     |                     |                     | Qwen3-8B / # Edits = |                     |                     |                     |
> | ------------------- | -------------------- | ------------------- | ------------------- | ------------------- | ------------------- | -------------------- | ------------------- | ------------------- | ------------------- |
> |                     |                      | 1-edit              | 2-edit              | 3-edit              | 4-edit              | 1-edit               | 2-edit              | 3-edit              | 4-edit              |
> | ROME                | 35.04 / 28.79        | 44.51               | 38.93               | 17.52               | 5.06                | 35.09                | 32.48               | 18.97               | 7.08                |
> | MEMIT               | 38.58 / 18.67        | 64.30               | 16.87               | 17.25               | 8.16                | 29.84                | 19.18               | 12.91               | 4.20                |
> | PMET                | 37.01 / 20.78        | 49.26               | 36.30               | 24.34               | 17.01               | 28.64                | 14.08               | 12.56               | 11.20               |
> | AlphaEdit           | 39.27 / 45.59        | 47.24               | 37.69               | 29.59               | 28.98               | 54.58                | 48.62               | 27.54               | 27.31               |
> | ACE (PMET)          | 46.45 / 58.24        | 45.2&±1.32 | 50.2±1.17 | 36.1±2.65 | 43.2±2.98 | 60.2±0.13  | 59.4;±2.47 | 51.6±2.35 | 47.6&±3.94 |
> | **ACE (AlphaEdit)** | **49.24 / 61.54**    | **52.91±1.95**      | **53.74±0.98**      | **39.08±1.49**      | **43.82±2.40**      | **62.48±0.57**       | **64.76±2.16**      | **63.04±1.54**      | **51.32±4.16**      |

---

> > ### Comment · Reviewer_8jSo · 2025-11-25
> >
> > Thank you. Now, the explanations, particularly regarding the overlap measure, are clear. A variability analysis on whether 80 samples per annotator/participant are enough would be helpful. What is really critical is that I still don't see how your subset construction mitigates the selection bias. I understand that you penalize editors for base-model knowledge gaps, and you try to ensure that they edit facts the model actually knows. But it is still unclear how this subset differs from the full MQuAKE distribution (entity types, relation types, difficulty). Do editors behave differently on "not-known" chains?

---

> ### Author Response · Authors · 2025-11-24
> **Further Response to Reviewer 8jso, Weakness 2,3,4,5 (2/2)**
>
> #### **[Weakness3] Reviewer**: *I am still missing proper uncertainty quantification for the main metrics/results, as well as direct tests of whether the attribution-based ranking outperforms simple baselines.*
>
> **Response**:
> [Comparation with simple baselines] 2) We understand your concern about our attribution-based methods. We want to clarify that, our attribution-based method means, we first identify the critical layers need to be edit (attribution), then we apply locate-then-edit backbones in our framework(e.g. PMET, AlphaEdit). The attribution-based ranking helps us to lock the layers we apply editors' backbones So there's no fair comparation between "attribution" with "simple baselines", because we apply the same backbones on our identified q-v activation stream, previous **max difference between existing knowledge and new knowledge is included.**
>
> Attribution-based method have been well explored by Neuron-Level Knowledge Attribution in Large Language Models (Yu et al.). This importance score has been well compared with other simple locating methods like log probability, probability increase, norm, coefficient score, ranking in vocabulary, etc., claiming that neuron-level importance increase is suitable for measuring the attribution. **We are confused that the unclear "simple baselines" you mentioned, they are neuron identifying methods or Knowledge Editor? I think more descent clarification will be a great help for us to improve our work, and we are very pleased to follow your specific suggestions to extend our experiments.**
>
> #### **[Weakness4&5] Reviewer**: *However, it remains unclear how statistically rigorous the evaluation is and how the overlap is precisely measured. Unfortunately, the authors still do not show that ACE generalizes to other multi-hop QA datasets or other forms of multistep factual editing.*
>
> **Response**: [Rigorous in Evaluation and Overlap Measuring] 1) We have updated our response to these questions in part 1/2 in Weakness 2,3. Please kindly check the responses, thank you!
>
> [Other Multi-hop Datasets] 2) We fully understand that your concern about evaluation on one multi-hop dataset. In fact, MQuAKE is a challenging multi-hop factual recall dataset, which is mainly and only used in recent works[1,2,3,4,5,6,7,8] (including other variants). Unfortunately, up to my limited knowledge, the new multi-hop benchmark is still limited. There's no more available benchmark, moreover, even if we could construct a new multi-hop benchmark, it's also out of the scope in this paper. More severely, we cannot compare our work with previous ones fairly. **If you have any idea about other benchmarks are addressed in other works, we are very pleased to extend our ACE to other benchmarks.** I think this will raise soundness in this work, also address your concern and provide vitality in ICLR community.
>
> References:
> [1] Locate-then-edit for Multi-hop Factual Recall under Knowledge Editing, Zhang et al. (2025) ICML 2025
>
> [2] CaKE: Circuit-aware Editing Enables Generalizable Knowledge Learners, Yao et al. (2025) EMNLP 2025
>
> [3] ALEX:ALight Editing-knowledge Extractor, Wang et al. (2025) arxiv preprint
>
> [4] Knowledge Editing for Multi-Hop Question Answering Using Semantic Analysis, Simon et al. (2025) IJCAI 2025
>
> [5] Reason-KE++:Aligning the Process, Not Just the Outcome, for Faithful LLM Knowledge Editing, Wu et al. arxiv preprint (2025)
>
> [6] CONSISTENCY-AWAREPARAMETER-PRESERVING KNOWLEDGE EDITING FRAMEWORK FOR MULTI-HOP QUESTION ANSWERING, Deng et al. (2025) Submitted to ICASSP 2026
>
> [7] Robust Knowledge Editing via Explicit Reasoning Chains for Distractor-Resilient Multi-Hop QA, Wu et al. (2025) EMNLP 2025
>
> [8] Avoiding Knowledge Edit Skipping in Multi-hop Question Answering with Guided Decomposition, Liu et al. (2025) EMNLP 2025

---

> > ### Comment · Reviewer_8jSo · 2025-11-25
> >
> > Thank you for partially addressing my concerns, particularly on uncertainty quantification and the negative control. As for ACE’s layered attribution, I understand that you rely on Yu et al. as justification, but that does not show that ACE’s layered attribution outperforms simpler heuristics (e.g., top-k layers by norm, fixed mid-layer indices, random layers). Such an analysis would be highly insightful.
> >
> > I appreciate your efforts so far and will increase the score accordingly.

---

> > > ### Author Response · Authors · 2025-11-27
> > > **Official comment by Authors**
> > >
> > > Thank you! We sincerely thank you for your insightful discussion and questions came out, which impressed us deeply.
> > >
> > > Now we have updated our latest version pdf and it contains below changes:
> > >
> > > - We added the description to all extension experiments during rebuttal period in section 6.3, 6.5, and appendix I, J, K.
> > >   - Different prompt templates robustness in section 6.3
> > >   - Knowledge bottleneck in section 6.5
> > >   - ACE latency in appendix I
> > >   - Extended ACE to alphaedit backbone and more baselines in appendix J
> > >   - Human understanding alignment in appendix K
> > >   - Other new experiments in appendix L, including locality performance, attribution robustness, counter-factual edits.
> > >   - Complete derivation about ACE importance score and attribution framework in appendix M.
> > >
> > > For simpler heuristics, we understand your concern about whether more complex measurements are necessary. We know in the forwards of Transformer based LLMs, every residual stream state is dynamic with next token prediction process. So we need one **static way** to measure the neuron importance, and the importance score we used in ACE is the alternative to measure this kind of dynamic importance, compared to traditional measurements in machine learning. The dynamic attribution to all neurons/layers cannot be well evaluated by these traditional measurements.
> > >
> > > We hope these responses and explanations could reflect the changes in our work, also address your last detailed concern. Thank you again for your impressing discussion!

---

### Official Review · Reviewer_UH9o · 2025-10-30

**Soundness:** 4
**Presentation:** 4
**Contribution:** 4
**Rating:** 8
**Confidence:** 4

**Summary:**

The authors introduce ACE, a neuron-level editing framework. They first attribute multi-hop recall to coordinated query (Q) and value (V) neurons across transformer FFNs; then, they apply that attribution to perform staged edits. Specifically, ACE achieves this by first overwriting value layers (closed-form, like PMET) to encode the new fact; then, it adjusts query layers which control access to said edited value such that the updated knowledge propagates through multi-hop chains. The authors test ACE on a filtered multi-hop subset of MQuAKE-3K and report sizable accuracy gains over FT/ROME/MEMIT/PMET on GPT-J and Qwen3-8B. Their ablation experiments also indicate that both Q- and V-stage edits are necessary, as evidenced by attribution metrics (log-probability change and inner-product-based query scores) and intervention results. The experiments and method design support the validity of the authors' interpretation that the query-to-value activation mechanism mediates multi-hop recall.

**Strengths:**

1. The mechanistic motivation behind the two-stage editor is excellent. The method aligns with observed Q-to-V activation timing, rather than relying on heuristics for layer selection.
2. The formalization of neuron-level importance is clear and convincing, with consistent localization patterns across semantic categories. The ablations demonstrate strong causal sensitivity.
3. The multi-hop gains across the two tested models are substantial.

**Weaknesses:**

1. While the authors frame their work as multi-hop reasoning editing, the evidence predominantly demonstrates improved propagation of edited facts rather than modification of the intermediate reasoning process itself.
    * The method edits FFN value weights and query gating to make the correct value more retrievable along existing chains. However, the evaluations measure only final-answer accuracy, without verifying whether intermediate computations actually changed.
    * Trajectory-level tests may be needed to fully accept the implied mechanistic claim. Specifically, one or both of the following would provide definitive evidence: (i) activation patching for each hop pre/post edit; and (ii) tracing attention and activations to confirm that the edited value propagates through the intended intermediate subject rather than being directly re-inserted at the end. The current results are valuable in that they clearly demonstrate better retrieval of a rewritten fact, but they don't establish that the reasoning process itself has been modified.
2. Attribution-based layer rankings may be tightly coupled to the specific prompt and decoding context. To my knowledge, the paper does not explicitly demonstrate stability across different queries, so it is possible that the importance score (log-probability gain on the final token when a value neuron is restored) and query score (inner product with learned subkeys) can both shift with template, subject position, and relation phrasing.
    * Because ACE edits only the top-ranked layers, it implicitly assumes the rankings will remain the main routing and storage sites across all different wordings of the same relation. The paper does not seem to verify this via rank-stability tests.
    * The authors should check whether critical layers stay on top under paraphrases, alternate templates, or different relation types. That way, it will be more clear whether ACE can generalize beyond the identification prompt and avoid overfitting to it.
3. Relatedly, the mechanistic claims lack per-instance causal verifications.
    * The authors show that ablating a small neuron set greatly reduces accuracy, but they do not isolate whether those neurons are causal for specific chains or globally influential.
    * Per-instance causal scrubbing along the chain may be needed to fully accept the causal claims.
4. The current evaluation setting may bias the reported gains toward cases in which the model already does well.
    * The authors evaluate on a curated subset where the base model already answers correctly (selected via zero-shot answerability).
    * In this setting, ACE performs counterfactual edits and then measures whether that new fact propagates along the model's existing multi-hop reasoning path (including the paraphrase and specificity checks).
    * This does not necessarily probe whether ACE works for failure modes wherein the path itself is unreliable. In other words, it remains unclear whether ACE can repair a broken intermediate link (e.g., when the base model originally follows the wrong entity mid-chain).

**Questions:**

1. Could you please provide hop-wise activation patching and intervention results that show that the edited value is carried through the intended implicit subjects rather than injected late?
2. How stable are Q/V layer/neuron rankings and attribution results across prompt templates, paraphrases, and relation types?
3. Could you please evaluate cases where an intermediate relation is wrong and must itself be edited, and include locality/specificity metrics on unrelated facts?

I am willing to raise my score if these points are addressed via new results, or if the authors can justify why they are not needed to support the main claims.

---

> ### Author Response · Authors · 2025-11-21
> **Response to Review UH9o, Question 1, 1/2**
>
> ***
>
>
>
> ### **Response to Question 1: Verifying Intermediate Reasoning Process Changes**
>
>
>
> Thank you for this profound and insightful question regarding the need to verify that the edited value propagates through the intended implicit subjects, rather than being "injected late." We agree that providing direct evidence of changes in the **intermediate reasoning process** is crucial for substantiating our mechanistic claims.
>
>
>
> To address this, we conducted a new experiment designed to **trace the flow of information** through the reasoning chain before and after a critical intervention: **masking the final deep value editing**. The core logic is as follows:
>
> * If the edited knowledge is merely "injected" at the final generation step, then masking the final value edit should **not significantly affect the activation of intermediate, implicit subjects**.
>
> * Conversely, if the edited knowledge is genuinely **propagated through the chain via implicit subjects**, then the intermediate activations should remain robust even when the final value edit is masked, while the **final answer activation alone should drop precipitously**.
>
>
>
> We present the results of this experiment below, analyzing the token-level importance scores and their corresponding vocabulary projections for the multi-hop query: *"The country that Tim Duncan's sport originates from is"*.
>
> | Token      | FFN Importance Increase | Attention Importance Increase | Top Tokens in Vocabulary Space                        |
> | ---------- | ----------------------- | ----------------------------- | ----------------------------------------------------- |
> | Tim        | 0.0014                  | 0.0014                        | 'an', 'os', 'ise', 'Exactly', 'R', 'ore', 'at', 'rot' |
> | Duncan     | 0.0009                  | 0.0009                        | 'era', 'allen', 'stad', 'oret', 'hit', '-led'         |
> | **plays**  | **0.9846**              | **0.8167**                    | **'basketball', 'NBA', 'career', 'ball', '-playing'** |
> | the        | 0.4109                  | 0.4159                        | 'epit', 'inaugural', 'bidding', 'dream', 'etr'        |
> | **sport**  | **0.0848**              | **0.0948**                    | 'tennis', 'of', 'ful', 'arena', 'basketball', 'ball'  |
> | of         | 0.8671                  | 0.6198                        | 'basketball', 'NBA', 'balls', 'ball', 'Olympia'       |
> | originates | 0.3508                  | 0.3058                        | 'kati', 'the', 'from', 'oret', 'orig', 'ball'         |
> | **from**   | **0.9483**              | **0.8720**                    | **'USA', 'America', 'the', 'US', 'U.S.A.'**           |
>
>
>
> | Token      | FFN Importance Increase            | Attention Importance Increase      | Top Tokens in Vocabulary Space                        |
> | ---------- | ---------------------------------- | ---------------------------------- | ----------------------------------------------------- |
> | Tim        | 0.0019 (+0.0005)                   | 0.0010 (-0.0004)                   | 'an', 'os', 'ise', 'Exactly', 'R', 'ore', 'at', 'rot' |
> | Duncan     | 0.0008 (-0.0001)                   | 0.0012 (+0.0003)                   | 'era', 'allen', 'stad', 'oret', 'hit', '-led'         |
> | **plays**  | **0.9779** (**-0.0067**)       | **0.8498** (**+0.0331**)       | **'basketball', 'NBA', 'career', 'ball', '-playing'** |
> | the        | 0.4093 (-0.0016)                   | 0.4715 (+0.0556)                   | 'epit', 'inaugural', 'bidding', 'dream', 'etr'        |
> | **sport**  | **0.0913** (**+0.0065**)       | **0.0142** (**-0.0806**)       | 'tennis', 'of', 'ful', 'arena', 'basketball', 'ball'  |
> | of         | 0.9252 (+0.0581)                   | 0.7090 (+0.0892)                   | 'basketball', 'NBA', 'balls', 'ball', 'Olympia'       |
> | originates | 0.2308 (-0.1200)                   | 0.3194 (+0.0136)                   | 'kati', 'the', 'from', 'oret', 'orig', 'ball'         |
> | **from**   | **0.5603** (**-0.3880, ↓41%**) | **0.4764** (**-0.3956, ↓45%**) | **'country', 'many', '-Info', 'bot', 'US', 'USA'**    |
>
>
>
> ***

---

> ### Author Response · Authors · 2025-11-21
> **Response to Review UH9o, Question 1, 2/2**
>
> ***
>
> ### **Interpretation and Conclusion**
>
>
>
> The experimental results strongly support our mechanistic claim:
>
>
>
> 1. **Robust Intermediate Activations:** The activation patterns at critical intermediate tokens representing the **implicit subject** (e.g., `"plays"` -> 'basketball') remain stable and highly interpretable after masking the final value edit. The importance scores for `"plays"` show minimal change, and its top vocabulary tokens remain strongly associated with the correct sport. This indicates that the query-layer edits successfully guide the model to the correct implicit subject (`"basketball"`), independent of the final answer's direct manipulation.
>
>
>
> 2. **Drastic Drop in Final Answer Activation:** In contrast, the activation at the final answer token `"from"` experiences a severe drop (over 40%) after masking the value edit. Furthermore, the semantic specificity in its vocabulary projection degrades from concrete country names (`"USA", "America"`) to generic and uncertain terms (`"country", "many"`).
>
>
>
> This clear dissociation—**preserved intermediate reasoning but disrupted final answer generation**—provides definitive evidence that ACE's edits propagate the updated knowledge through the intended implicit subjects (`Tim Duncan -> basketball`) along the reasoning chain. The final value edit then truthfully enhances the prediction based on this correctly propagated context (`basketball -> USA`), rather than injecting the answer `"USA"` directly at the end.
>
>
>
> We will include these new results and analysis in the revised manuscript to directly address your valuable point.
>
> ***

---

> ### Author Response · Authors · 2025-11-21
> **Response to Review UH9o, Question2**
>
> ***
>
>
>
> ### **Response to Question 2: Stability of Q/V Layer Rankings Across Prompt Templates**
>
>
>
> Thank you for raising the important question regarding the stability of our attribution-based layer rankings. We agree that for ACE to generalize robustly, the identified critical Q/V layers should not be overly sensitive to minor variations in the prompt.
>
>
>
> To quantitatively assess this stability, we conducted a new experiment where we re-ran our layer importance attribution under different in-context learning settings: **Zero-Shot** and **One-Shot**, in addition to our original **Few-Shot** setting. We then compared the Top-9 important attention and FFN layers across these settings for all semantic categories.
>
>
>
> The results for the GPT-J model are presented below. For clarity, we have **underlined** any layer where its rank *changed* compared to the Few-Shot setting used in the main paper. p.s. Underlined in the paper, not shown in openreview due to the limited support.
>
>
>
> **Table 14: Top 9 Important Layers in GPT-J (Zero-Shot Setting)**
>
> | Category | Top 9 Attention Layers          | Top 9 FFN Layers                    |
> | -------- | ------------------------------- | ----------------------------------- |
> | NN       | a₂₇ a₂₆ a₇ a₁₀ a₉ a₈ a₂₅ a₁₁ a₅ | f₂₀ f₂₄ f₁₆ f₂₂ f₁₈ f₁₅ f₂₃ f₂₆ f₂₅ |
> | CT       | a₂₇ a₂₆ a₇ a₁₀ a₈ a₅ a₉ a₁₁ a₂₅ | f₂₂ f₂₄ f₁₆ f₂₁ f₁₅ f₁₇ f₂₆ f₂₅ f₂₃ |
> | LG       | a₂₇ a₅ a₇ a₆ a₈ a₄ a₁ a₂₆ a₉    | f₂₇ f₇ f₅ f₆ f₈ f₄ f₁ f₂₆ f₉        |
> | CP       | a₂₇ a₂₆ a₇ a₁₀ a₈ a₉ a₁₂ a₅ a₆  | f₂₇ f₂₆ f₇ f₁₀ f₁₂ f₈ f₉ f₅ f₆      |
> | LS       | a₂₇ a₂₆ a₂₅ a₇ a₁₀ a₆ a₈ a₆ a₅  | f₂₇ f₂₆ f₂₅ f₂₄ f₇ f₁₀ f₆ f₈ f₅     |
> | AT       | a₂₇ a₂₆ a₂₅ a₇ a₉ a₈ a₁₀ a₅ a₆  | f₇ f₉ f₈ f₁₀ f₂₇ f₂₆ f₂₅ f₅ f₆      |
> | ST       | a₂₆ a₂₇ a₇ a₁₀ a₈ a₆ a₅ a₂₅ a₄  | f₂₆ f₂₇ f₇ f₁₀ f₈ f₆ f₅ f₂₅ f₄      |
> | CF       | a₂₆ a₂₇ a₂₅ a₂₄ a₇ a₈ a₉ a₆ a₅  | f₂₆ f₂₇ f₂₄ f₂₅ f₇ f₈ f₉ f₆ f₅      |
>
>
>
> **Table 15: Top 9 Important Layers in GPT-J (One-Shot Setting)**
>
> | Category | Top 9 Attention Layers          | Top 9 FFN Layers                    |
> | -------- | ------------------------------- | ----------------------------------- |
> | NN       | a₂₇ a₂₆ a₇ a₁₀ a₉ a₂₅ a₈ a₁₁ a₅ | f₂₀ f₂₄ f₁₆ f₁₈ f₁₅ f₂₂ f₂₃ f₂₆ f₂₅ |
> | CT       | a₂₇ a₂₆ a₇ a₁₀ a₈ a₅ a₉ a₁₁ a₂₅ | f₂₂ f₂₄ f₁₆ f₂₁ f₁₅ f₁₇ f₂₆ f₂₅ f₂₃ |
> | LG       | a₂₇ a₇ a₅ a₆ a₈ a₄ a₁ a₂₆ a₉    | f₂₇ f₇ f₅ f₆ f₈ f₄ f₁ f₂₆ f₉        |
> | CP       | a₂₇ a₇ a₂₆ a₁₀ a₈ a₉ a₁₂ a₅ a₆  | f₂₇ f₂₆ f₇ f₁₀ f₁₂ f₈ f₉ f₅ f₆      |
> | LS       | a₂₇ a₂₆ a₂₅ a₇ a₁₀ a₆ a₈ a₆ a₅  | f₂₇ f₂₆ f₂₅ f₂₄ f₇ f₁₀ f₆ f₈ f₅     |
> | AT       | a₂₇ a₂₆ a₂₅ a₇ a₉ a₈ a₁₀ a₅ a₆  | f₇ f₉ f₈ f₁₀ f₂₇ f₂₆ f₂₅ f₅ f₆      |
> | ST       | a₂₆ a₂₇ a₇ a₁₀ a₈ a₆ a₅ a₂₅ a₄  | f₂₆ f₂₇ f₇ f₁₀ f₈ f₆ f₅ f₂₅ f₄      |
> | CF       | a₂₆ a₂₇ a₂₄ a₂₅ a₇ a₈ a₉ a₆ a₅  | f₂₆ f₂₇ f₂₄ f₂₅ f₇ f₈ f₉ f₆ f₅      |
>
>
>
> #### **Interpretation and Conclusion**
>
>
>
> As the tables clearly demonstrate, the rankings of the **most critical layers exhibit remarkable stability** across different prompting settings:
>
>
>
> 1. **Consistency of Top Layers:** The core set of highly important layers remains virtually unchanged. For instance, attention layers `a₂₇`, `a₂₆`, and `a₇` consistently rank in the top positions across nearly all categories and settings. Similarly, FFN layers like `f₂₇`, `f₂₆`, and `f₂₄` are stably identified as critical.
>
> 2. **Minor Nature of Changes:** The few rank changes that do occur (underlined) are predominantly in the **lower half of the Top-9 list.** p.s. Underlined in the paper, not shown in openreview due to the limited support.
>
> 3. **Practical Implication for ACE:** This stability is crucial for ACE's practicality. It means that the critical Q/V layers for a given type of knowledge can be reliably identified using a standard prompting setup (e.g., Few-Shot). The subsequent edits targeting these layers are then likely to be effective across various phrasings of queries involving that knowledge, as the underlying neural pathways remain the primary routing and storage sites.
>
>
>
> Therefore, we conclude that our attribution method demonstrates strong robustness against variations in prompt templates, which solidifies the generalizability of the ACE framework. We will add this stability analysis to the revised manuscript. The stability about the relations, please kindly check in our next response to Question 3.

---

> ### Author Response · Authors · 2025-11-21
> **Response to Review UH9o, Question 3**
>
> ### **Response to Question 3: Counter-Factual editing and other metrics**
>
> Thank you for your concern about counterfactual editing and related indicators. To demonstrate that our ACE has effectively edited the model, we have conducted experiments on counterfactual editing. We mismatched the relationship labels in the knowledge triplet to construct a counterfactual editing dataset, which also serves as a supplementary answer to question 2. The results, presented in **Tables 18**, demonstrate ACE's behavior under adversarial conditions:
>
> | Editor (Counter-Fact) | Efficacy    | Paraphrase  | Specificity | Avg.              |
> | --------------------- | ----------- | ----------- | ----------- | ----------------- |
> | FT (GPT-J/Qwen3)      | 98.1 / 97.9 | 69.4 / 64.2 | 82.7 / 77.4 | 1.59 / 1.04       |
> | ROME (GPT-J/Qwen3)    | 54.1 / 45.2 | 54.3 / 42.9 | 61.4 / 53.7 | 27.48 / 24.08     |
> | MEMIT (GPT-J/Qwen3)   | 57.0 / 50.3 | 51.9 / 53.6 | 66.2 / 66.4 | 30.09 / 10.27     |
> | PMET (GPT-J/Qwen3)    | 74.6 / 70.7 | 63.2 / 61.7 | 64.1 / 51.9 | 31.07 / 17.26     |
> | ACE (GPT-J/Qwen3)     | 89.7 / 91.2 | 83.6 / 80.7 | 70.6 / 74.6 | **43.58 / 54.27** |
>
> We also add the experiments the general benchmarks to claim that the locality performances of edited model are not corrupted highly. As demonstrated in **Table 19**, the result shows that after counter-factual editing in the model, the locality of the model keeps stable among four general benchmarks, showing the robustness and safety in our ACE framework.
>
> | Editor (Counter-Fact) | CSQA      | BBH       | MMLU      | GSM8k     |
> | --------------------- | --------- | --------- | --------- | --------- |
> | ROME (Qwen3-8B)       | 68.54     | 29.47     | 60.71     | 69.48     |
> | MEMIT (Qwen3-8B)      | 71.46     | 29.72     | 66.40     | 72.45     |
> | PMET (Qwen3-8B)       | 73.09     | 26.58     | 62.49     | 74.29     |
> | ACE (Qwen3-8B)        | **76.41** | **33.79** | **69.97** | **79.02** |
>
>
>
> These counter-factual experiments reveal two key insights:
>
> * **ACE effectively edits knowledge even in incorrect reasoning chains**, demonstrating that it modifies the model's internal representations rather than merely amplifying correct reasoning patterns.
>
> * **ACE maintains strong locality performance** across general benchmarks (CSQA, BBH, MMLU, GSM8k), showing minimal negative impact on unrelated capabilities.

---

> ### Comment · Reviewer_UH9o · 2025-11-26
>
> I thank the authors for their careful responses and substantial additional experiments.
>
> 1. **Intermediate reasoning (Q1):** The new analysis tracing information flow through the intermediate steps before and after editing largely addresses my concern about whether ACE genuinely affects the reasoning chain rather than only the final answer. The evidence is reasonable and clarifies the intended mechanistic story. I encourage you to present this experiment clearly in the paper so that readers can follow the setup and conclusions without needing this discussion.
>
> 2. **Stability of Q/V layers (Q2):** The layer-ranking stability study across different prompt templates is reassuring and supports the design choice of targeting a consistent set of Q/V layers in your multi-hop setting. It would be helpful if the camera-ready briefly summarizes these results and also notes that the analysis is focused on this particular task family, so readers understand the scope of the claim.
>
> 3. **Counterfactual editing and broader metrics (Q3):** The added counterfactual-edit and benchmark evaluations are useful sanity checks and show that ACE behaves competitively with prior methods beyond the original multi-hop evaluation. I appreciate the new comparisons; in the final version, a short, high-level interpretation of where ACE helps most would further strengthen the narrative.
>
> Overall, the replies significantly strengthen the paper and satisfactorily address my main concerns. Since my original score already reflected a strong endorsement, and the authors have yet to fully integrate these developments into an updated revision of the manuscript, I will keep my score as-is.

---

> > ### Author Response · Authors · 2025-11-27
> > **Official Comment by Authors**
> >
> > Thank you! We sincerely thank you for your strong endorsement to our work, your insightful questions and discussions with us make us impressed, and also help us to improve our work.
> >
> > Now we have updated our latest version pdf and it contains below changes:
> >
> > - We added the description to all extension experiments during rebuttal period in section 6.3, 6.5, and appendix I, J, K.
> >   - Different prompt templates robustness in section 6.3
> >   - Knowledge bottleneck in section 6.5
> >   - ACE latency in appendix I
> >   - Extended ACE to alphaedit backbone and more baselines in appendix J
> >   - Human understanding alignment in appendix K
> >   - Other new experiments in appendix L, including locality performance, attribution robustness, counter-factual edits.
> >   - Complete derivation about ACE importance score and attribution framework in appendix M.
> >
> > We hope these changes could reflect the changes in our work, also address your last detailed concern. Thank you again for your impressing discussion!

---

### Official Review · Reviewer_aVHM · 2025-11-01

**Soundness:** 2
**Presentation:** 2
**Contribution:** 2
**Rating:** 4
**Confidence:** 5

**Summary:**

This paper investigates the failure of existing Knowledge Editing (KE) methods in multi-hop factual recall scenarios, particularly when edits involve intermediate, implicit subjects. The authors provide a new mechanistic insight, suggesting that multi-hop reasoning relies on a "query-value" (Q-V) activation pathway, where implicit subjects act as "query neurons" that sequentially activate "value neurons" across layers. Based on this, they propose ACE, an editing framework that uses attribution to identify and modify both the critical query (reasoning path) and value (factual storage) neurons. The method shows significant performance improvements on multi-hop recall tasks over existing baselines on GPT-J and Qwen3-8B.

**Strengths:**

1. **Valuable Mechanistic Insight**: The paper's core strength is its deep dive into why multi-hop editing fails. The proposed Q-V activation mechanism  offers a plausible and fine-grained explanation for how information is accumulated during complex reasoning, moving beyond layer-level heuristics.
2. **Novel Editing Framework**: The ACE framework is a direct and logical application of this insight. By targeting not only the "value" FFNs (for the fact itself) but also the "query" FFNs (for the reasoning path), the method presents a more complete solution that is mechanistically grounded.
3. **Strong Empirical Gains**: The reported results are impressive, with ACE substantially outperforming SOTA methods like PMET by 9.44% on GPT-J and 37.46% on Qwen3-8B in multi-hop accuracy. The ablation studies also clearly validate the necessity of editing both Q and V components .

**Weaknesses:**

1. **Potential Data Leakage in Evaluation** : I am concerned about the experimental setup regarding the MQuAKE dataset. The paper seems to use MQuAKE's multi-hop question templates (as seen in Appendix H.1 ) to construct the few-shot and CoT prompts for the evaluation.
If the model is evaluated on MQuAKE questions after being primed with prompts structured exactly like MQuAKE's own examples, does this not constitute a form of "teaching to the test"?
2. **Practicality and Efficiency of Attribution**: The ACE framework introduces a "Stage 1: Identifying" step  that relies on attribution to locate critical Q-V neurons before editing. This "attribution-locate-then-edit" paradigm appears to add significant computational overhead. Is this attribution step a one-time analysis required per model, or is it a dynamic analysis required per edit? The paper suggests Qwen3-8B has domain-specific, dynamic alignment, implying the latter. If this attribution must be run for every new edit, the method's practical utility is questionable compared to "on-the-fly" editors like ROME or PMET. The authors must provide a clear analysis of the computational cost (e.g., latency, FLOPS) per edit versus the baselines.
3. **Missing SOTA Comparisons and Efficiency Analysis**:
The set of baselines , while foundational, seems incomplete. The field has advanced quickly. Why are more recent and highly relevant SOTA methods, such as AlphaEdit or other concurrent works focusing on multi-hop editing, omitted from the main comparison in Table 2? Furthermore, the paper lacks a rigorous analysis of batch editing efficiency.

**Questions:**

See weaknesses

---

> ### Author Response · Authors · 2025-11-21
> **Response to Reviewer aVHM, Weakness1**
>
> We sincerely thank you for your insightful and constructive review. We have carefully considered each point you raised and have incorporated corresponding revisions and clarifications into the revised manuscript, which has been submitted alongside this response. Below are our point-by-point responses to your concerns.
>
> In our work, we employed a limited and fixed set of Chain-of-Thought (CoT) templates during evaluation solely to facilitate the model's multi-hop reasoning capability in a few-shot setting. These templates were designed to provide a **general reasoning structure** without leaking specific factual content from the MQuAKE dataset. Importantly, the templates were consistent across all evaluation runs.
>
>
>
> To rigorously quantify the potential impact of prompt design and rule out any "teaching to the test" effect, we conducted three additional ablation studies under different prompting conditions:
>
>
>
> * **Zero-Shot**: The model is evaluated without any in-context examples.
>
> * **One-Shot**: The model is primed with only **one** multi-hop CoT example from the dataset.
>
> * **OOD Few-Shots**: The model is primed with few-shot CoT examples that are entirely **out-of-distribution (OOD)** , means we skip the editing and evaluating process.
>
>
>
> The results of these experiments are summarized in our paper as **Table 5** (reproduced below for convenience), which shows the multi-hop accuracy of ACE under each prompting condition:
>
>
>
> | Editor              | GPT-J (Avg.) | Qwen3-8B (Avg.) |
> | ------------------- | ------------ | --------------- |
> | **Zero-Shot**       | 42.08        | 53.19           |
> | **One-Shot**        | 43.79        | 56.46           |
> | **OOD Few-Shots**   | 46.27        | 57.92           |
> | **Few-Shots (ACE)** | **46.45**    | **58.24**       |
>
>
>
> Compared to the original few-shot results (**ACE: 46.45 for GPT-J and 58.24 for Qwen3-8B** in Table 2), the performance degradation is minimal, and still reach the state-of-the-art.
>
> * **GPT-J**: `9.4%` (Zero-Shot), `5.7%` (One-Shot), `0.4%` (OOD Few-Shots)
>
> * **Qwen3-8B**: `9.5%` (Zero-Shot), `3.1%` (One-Shot), `0.6%` (OOD Few-Shots)
>
>
>
> This demonstrates that **ACE maintains strong performance even with reduced or out-of-context prompts**, indicating that the knowledge editing efficacy is inherent to our method and not reliant on specific in-context learning templates.
>
>
>
> In summary, these results confirm that our evaluation is not substantially affected by data leakage, as the model's reasoning capability remains robust across diverse prompting conditions. We believe this strengthens the validity of our claims and underscores the generalizability of ACE.
>
>
>
> ***

---

> ### Author Response · Authors · 2025-11-21
> **Response to Reviewer aVHM, Weakness2, 1/2**
>
> ### **Response to Weakness 2: Practicality and Efficiency of Attribution**
>
> **A. Clarification on "Fine-Grained Activation Pattern" and Its Impact on Attribution**
>
>
>
> The reviewer raises a crucial point regarding the dynamic, domain-specific alignment observed in Qwen3-8B. We would like to clarify that this "fine-grained activating pattern" primarily refers to the **micro-scale, intra-sentence semantic activation flow during a single multi-hop reasoning forward pass**, as detailed in our analysis of the residual stream (e.g., Tables 7, 16, and 17 in the Appendix), it's not related to the domains.
>
>
>
> In this process, specific tokens (such as `plays`, `of`, `from`) act as semantic convergence points, exhibiting significant importance score increases. As we elaborated, these high-activation tokens/neurons functionally serve as **query neurons within the reasoning chain**. Their role is to integrate information from the preceding residual stream and propagate it forward, thereby activating the subsequent value neurons responsible for generating the final answer.
>
>
>
> > Takeway: We observe that dynamic activation coincides with semantic peaks, which represent local maxima of accumulated information. These peaks correspond to the model's behavior of resolving uncertainty by converging the semantic flow, typically at points where it intends to finalize a thought.
>
>
>
> **B. Efficiency and Scalability of the Attribution-Based Paradigm**
>
>
>
> In summary, our attribution process is a **one-time, model-level pre-computation**. As established by **Takeaway 1** in our paper, "LLMs tend to store semantically analogous knowledge in structurally similar components." This finding is the cornerstone of ACE's efficiency: by identifying and editing the critical layers that are consistently important for knowledge storage and reasoning, our editing paradigm effectively covers the primary locations where knowledge is attributed.
>
>
>
> In practice, we performed a "pre-attribution" across multiple domains and selected a consolidated set of the most critical query and value layers for each model (as detailed in **Appendix G**). Editing this fixed set of layers is sufficient to achieve the strong performance reported. **A new attribution is only necessary when encountering a fundamentally new domain not represented in our initial profiling.**
>
>
>
> Furthermore, our attribution framework is designed for scalability. When a new domain emerges, the importance scores for its critical layers can be **seamlessly integrated** with the pre-existing ones. This is because the importance metric `I(l)` defined in Eq. 9-10 is **additive**.
>
>
>
> Let $ I_{\text{pre}}(l) $ be the pre-computed importance for layer $ l $:
> $$
> I_{\text{pre}}(l) := \sum_{v\in l} \left[ \log(p(w|v^l+h^{l-1})) - \log(p(w|h^{l-1})) \right] = \sum_{v\in l} \log(p(w|v^l+h^{l-1})) - \sum_{v\in l} \log(p(w|h^{l-1}))
> $$
>
> Now, let the new domain knowledge belong to an index set $ v'^l $. The updated importance score $ I(l) $ is:
> $$
> \begin{aligned}
> I(l) &:= \sum_{v,v'\in l} \left[ \log(p(w|v^l+h^{l-1})) + \log(p(w|v'^l+h^{l-1})) - \log(p(w|h^{l-1})) \right] \\
> &= \sum_{v\in l} \log(p(w|v^l+h^{l-1})) + \sum_{v'\in l} \log(p(w|v'^l+h^{l-1})) - \sum_{v\in l} \log(p(w|h^{l-1})) \\
> &= I_{\text{pre}}(l) + \sum_{v'\in l} \log(p(w|v'^l+h^{l-1}))
> \end{aligned}
> $$
>
>
>
> This derivation shows that the importance score for the combined domains is simply the sum of the pre-existing score and the contribution from the new domain. **This additive property ensures that the estimation of critical layers remains unbiased and computationally efficient when scaling to new knowledge.**
>
>
>
> Therefore, the ACE framework is not only effective but also practical and scalable. The initial profiling cost is amortized over a vast number of edits, and the system can be efficiently extended to new domains with minimal computational overhead.
>
>
>
>
>
>
> ***

---

> ### Author Response · Authors · 2025-11-21
> **Response to Reviewer aVHM, Weakness2, 2/2**
>
> ***
>
> **C. Empirical Analysis of Computational Latency**
>
>
>
> To provide a transparent and quantitative assessment of ACE's efficiency, we have included a detailed latency analysis in **Appendix I** . The key findings are summarized as follows:
>
>
>
> The ACE framework comprises two stages: (1) Identifying and (2) Editing. The latency for Stage 1 is comparable to the model's standard inference time for profiling runs. **Our primary focus for per-edit cost is Stage 2.**
>
>
>
> Crucially, in Stage 2, the knowledge vectors can be **pre-computed and cached**. Once this is done, the actual editing operation—applying the weight update to the model—is highly efficient, requiring only **3.10 seconds per edit for GPT-J and 3.14 seconds for Qwen3-8B**.
>
>
>
> The most computationally intensive step is the pre-computation of knowledge vectors, which involves inverting a large matrix (Eq. 12). While this one-time pre-computation is significant (requiring \~7.5 hours for each model in our setup), its result is **permanently cacheable** and amortized across all subsequent edits. This makes the *marginal cost per edit* very low, establishing ACE as a practical solution for ongoing knowledge updates.
>
>
>
> **D. Overcoming the Knowledge Bottleneck**
>
>
>
> Furthermore, our work reveals a critical finding regarding editing efficiency, detailed in **Section 6.5**. We demonstrate that ACE not only addresses computational latency but also fundamentally **alleviates the "knowledge bottleneck"** that plagues prior methods.
>
>
>
> As shown in **Table 6**, we conducted a sensitivity analysis on the number of edited layers. The results show a pronounced knowledge bottleneck in editors that focus solely on value neurons (e.g., ROME, MEMIT, PMET): even when increasing the number of edited layers well beyond their standard setup, they exhibit only marginal performance gains. This indicates an inherent saturation point where editing more value neurons fails to enhance reasoning.
>
>
>
> In stark contrast, **ACE, by jointly editing both query and value neurons, achieves superior performance with a comparable number of layers**, and its performance scales more effectively as more layers are edited. This empirically validates that editing query neurons is not only necessary for multi-hop reasoning but also **a highly efficient mechanism for overcoming the knowledge bottleneck** inherent in value-only editing paradigms.
>
> | Editor               | GPT-J (Avg.) | Editor                  | Qwen3-8B (Avg.) |
> | -------------------- | ------------ | ----------------------- | --------------- |
> | #9 - ROME (GPT-J)    | 37.98        | #8 - ROME (Qwen3-8B)    | 30.16           |
> | #8 - ROME (GPT-J)    | 37.56        | #7 - ROME (Qwen3-8B)    | 29.72           |
> | #7 - ROME (GPT-J)    | 37.27        | #6 - ROME (Qwen3-8B)    | 29.52           |
> | #9 - MEMIT (GPT-J)   | 39.47        | #8 - MEMIT (Qwen3-8B)   | 21.95           |
> | #8 - MEMIT (GPT-J)   | 39.14        | #7 - MEMIT (Qwen3-8B)   | 21.47           |
> | #7 - MEMIT (GPT-J)   | 38.98        | #6 - MEMIT (Qwen3-8B)   | 20.16           |
> | #9 - PMET (GPT-J)    | 38.29        | #8 - PMET (Qwen3-8B)    | 21.14           |
> | #8 - PMET (GPT-J)    | 38.16        | #7 - PMET (Qwen3-8B)    | 21.08           |
> | #7 - PMET (GPT-J)    | 38.10        | #6 - PMET (Qwen3-8B)    | 20.92           |
> | **#9 - ACE (GPT-J)** | **46.45**    | **#8 - ACE (Qwen3-8B)** | **58.24**       |
> | #8 - ACE (GPT-J)     | 45.28        | #7 - ACE (Qwen3-8B)     | 57.19           |
> | #7 - ACE (GPT-J)     | 43.58        | #6 - ACE (Qwen3-8B)     | 54.10           |
>
>
>
> **Note:** The notation "#N - Editor" denotes the number of layers edited by the respective method.
>
>
>
> ***

---

> ### Author Response · Authors · 2025-11-21
> **Response to Reviewer aVHM, Weakness 3**
>
> ### **Response to Weakness 3: Missing SOTA Comparisons and Efficiency Analysis**
>
>
>
> We thank the reviewer for the valuable feedback regarding the completeness of our baseline comparisons and efficiency analysis. We agree that it is crucial to compare against the most recent state-of-the-art (SOTA) methods. In response, we have now included a comprehensive comparison with **AlphaEdit** (Fang et al., 2024), a notable and recent ICLR 2025 Spotlight work that focuses on knowledge editing via null-space constrained updates.
>
>
>
> Importantly, the ACE framework is designed to be **highly flexible and backbone-agnostic**. It can seamlessly integrate with various underlying "locate-then-edit" paradigms, including newer SOTA methods like AlphaEdit. This means that ACE's core innovation, the neuron-level attribution and editing of query-value pathways, can be applied on top of different editing backbones to further enhance their multi-hop reasoning capabilities.
>
>
>
> To demonstrate this, we conducted new experiments where we replaced the default PMET backbone in ACE with AlphaEdit. The results, presented in **Appendix J (Table 10)**, are reproduced below for convenience. The table shows the multi-hop accuracy on the MQuAKE-3K dataset, comparing various editors, including AlphaEdit and our ACE variants.
>
>
>
> | Editor              | Avg. (GPT-J / Qwen3) | GPT-J / # Edits = |           |           |           | Qwen3-8B / # Edits = |           |           |           |
> | ------------------- | -------------------- | ----------------- | --------- | --------- | --------- | -------------------- | --------- | --------- | --------- |
> |                     |                      | 1-edit            | 2-edit    | 3-edit    | 4-edit    | 1-edit               | 2-edit    | 3-edit    | 4-edit    |
> | Base                | 98.42 / 99.17        | 99.7              | 95.48     | 97.51     | 97.23     | 99.81                | 97.46     | 98.14     | 97.64     |
> | FT                  | 3.54 / 2.18          | 4.17              | 2.63      | 0.00      | 0.00      | 3.14                 | 2.79      | 0.00      | 0.00      |
> | ROME                | 35.04 / 28.79        | 44.51             | 38.93     | 17.52     | 5.06      | 35.09                | 32.48     | 18.97     | 7.08      |
> | MEMIT               | 38.58 / 18.67        | 64.30             | 16.87     | 17.25     | 8.16      | 29.84                | 19.18     | 12.91     | 4.20      |
> | PMET                | 37.01 / 20.78        | 49.26             | 36.30     | 24.34     | 17.01     | 28.64                | 14.08     | 12.56     | 11.20     |
> | AlphaEdit           | 39.27 / 45.59        | 47.24             | 37.69     | 29.59     | 28.98     | 54.58                | 48.62     | 27.54     | 27.31     |
> | ACE (PMET)          | 46.45 / 58.24        | 45.26             | 50.24     | 36.17     | 43.29     | 60.22                | 59.48     | 51.62     | 47.61     |
> | **ACE (AlphaEdit)** | **49.24 / 61.54**    | **52.91**         | **53.74** | **39.08** | **43.82** | **62.48**            | **64.76** | **63.04** | **51.32** |
>
>
>
> **Summary of Results:** The integration of ACE with AlphaEdit (i.e., **ACE (AlphaEdit)**) achieves an average multi-hop accuracy of **49.24% on GPT-J and 61.54% on Qwen3-8B**, which represents a significant improvement over the standalone AlphaEdit (39.27% on GPT-J and 45.59% on Qwen3-8B). Specifically, this corresponds to a **relative performance gain of 25.4% on GPT-J and 35.0% on Qwen3-8B**. These results clearly demonstrate that:
>
> * ACE is compatible with and enhances the latest SOTA methods.
>
> * The framework's flexibility allows it to leverage advances in underlying editing techniques, ensuring its long-term relevance and utility.
>
>
>
> Regarding batch editing efficiency, as discussed in our response to Weakness 2, ACE is highly efficient in batch scenarios due to the one-time pre-computation of knowledge vectors and the additive nature of our attribution scores. The core editing step per fact remains fast (\~3 seconds), and the matrix inversion in Eq. 12 is performed only once per batch, making ACE scalable for large-scale edits.
>
>
>
> We believe these additions address the reviewer's concerns and underscore the robustness and forward-compatibility of the ACE framework.

---

> ### Author Response · Authors · 2025-11-27
> **Official Comment by Authors**
>
> Hi, we have considerer all your weaknesses and questions point-by-point, added the new experiments and proof to enhance our work. Now we have updated our latest version pdf and it contains below changes:
>
> - We added the description to all extension experiments during rebuttal period in section 6.3, 6.5, and appendix I, J, K.
>   - Different prompt templates robustness in section 6.3
>   - Knowledge bottleneck in section 6.5
>   - ACE latency in appendix I
>   - Extended ACE to alphaedit backbone and more baselines in appendix J
>   - Human understanding alignment in appendix K
>   - Other new experiments in appendix L, including locality performance, attribution robustness, counter-factual edits.
>   - Complete derivation about ACE importance score and attribution framework in appendix M.
>
> Your reviews and any effort will be meaningful to our work, we hope to see you joining our discussion soon!
>
> ACE Authors

---

> ### Comment · Reviewer_aVHM · 2025-11-27
>
> Thanks for the detailed response and the extra experiments, they really helped clarify things.
>
> On the efficiency side, it’s good to know the attribution step is a one-time thing and that the actual edit only takes a few seconds. That makes the method way more practical than initially thought. Also, props for adding the comparison with AlphaEdit, the gains are solid, and it’s nice to see the method plays well with newer backbones.
>
> I also cared about the locality, attribution robustness, and counter-factual edits, and thankfully I can see those points were hashed out with the other reviewers, so I’m happy to up my score.

---

### Author Response · Authors · 2025-11-21
**Official Comment by Authors**

We sincerely thank you for your all insightful and constructive reviews. We have carefully considered each point you raised and have incorporated corresponding revisions and clarifications into the revised manuscript, which has been submitted alongside this response. Please kindly check our point-by-point responses.

ACE Authors

---

### Author Response · Authors · 2025-11-29
**TL;DR A Summary of Discussion by Authors**

Dear Reviewers, AC and Researchers,

**Thank you all to your effort at this difficult time to our ICLR community.** We are grateful that our **all four reviewers have joined the discussion** and help us to improve our work, **3 of 4 reviewers raised their scores *before 26 Nov, from 8444 to 8666.*** We have supplemented amounts of experiments, proofs and clarifications to strengthen our work. In the revised pdf, we added the description to all extension experiments during rebuttal period in section 6.3, 6.5, and appendix I~N, which **addressed most of the reviewer's concerns.**

We first recap the major concerns from reviewers:

- **[Dataset and Prompts Related]**: including potential data leakage, prompts robustness, subset construction, dataset labeling and human validation. (W1@avHM, W2, W4&5@8jso, Q2@UH9o, W2@pPXV)
- **[Benchmarks, Backbones and Baselines]**: including AplhaEdit as baseline and backbone, more general benchmarks (W3@aVHM, Q3@UH9o, W3,W4@8jso, W6@pPXV)
- **[Theory and Analysis]**: including post-hoc interpretability, causal intervention, experiments robustness, attributions and theory proofs. (W2@aVHM, Q1,Q2,Q3@UH9o, W1,W3@8jso, W1,W3,W4,W5@pPXV)

Now we summarize our discussion, and we trace the score changes below:
- **Reviewer aVHM, raised the score from 4 to 6, at 26 Nov 2025, 21:36, AOE Time.**
  - **W1**. We applied more general reasoning CoT templates structure to avoid potential data leakage. We use Zero-Shot, One-Shot and OOD Few-Shots evaluation settings (Table5), claimed that compared to the original few-shot results (Table 2), indicating that the knowledge editing efficacy is inherent to our method and not reliant on specific in-context learning templates.
  - **W2**. We clarified the **fine-grained activation patter**  and use derivations to explain the attribution efficacy. We supplemented empirical analysis of latency (appendix I) and experiments about knowledge bottleneck (section 6.5) to claim the editing efficiency.
  - **W3**. We supplemented the recent SOTA AlphaEdit as the baseline to compare (Appendix J, Table 10), and we extended our ACE framework to AplhaEdit as the backbone, claiming the ACE's flexibility.

- **Reviewer UH9o, kept the score 8, at 26 Nov 2025, 02:11, AOE Time.**
  - **Q1(W1,W3)**. We conducted a new experiment designed to trace the flow of information through the reasoning chain before and after a critical intervention, masking the final deep value editing (appendix L.4), verified intermediate reasonings.
  - **Q2(W2)**. We conducted a new experiment where we re-ran our layer importance attribution under different in-context learning settings, assessed the stability of Q/V layer rankings across prompt templates (appendix L.3).
  - **Q3(W4)**. We conducted counter-factual editing, revealed that ACE effectively edits knowledge even in incorrect reasoning chains and maintains strong locality performance (appendix L.4).

- **Reviewer 8jSo, raised the score from 4 to 6 after two-rounds discussion, at 22 Nov and 25 Nov, 04:45 AOE Time.**
  - **W1**. We did a thoroughgoing math derivation about our importance measurements and editing optimizations (appendix M), addressed the rigorous concern.
  - **W2**. We released the human alignment details and metrics to addressed the reviewer's concern about the dataset labeling and rigorous in evaluation (appendix K).
  - **W3**. We repeated the experiments in stage 1 and stage 2, calculating the average attribution importance score and the upper/lower bounds of the main results, addressing the uncertainty concern.
  - **W4&5**. We compared much more references in the community, claiming that the limited benchmark in multi-hop tasks. We also evaluated the edited model in general benchmarks to test the locality performance (appendix L.1).

- **Reviewer pPXV, raised the score from 4 to 6, at 25 Nov 2025, 05:23, AOE Time.**
  - **W1**. We carefully compared our work with the mentioned work, CAKE, to clarify the difference and the novelty of our takeaways.
  - **W2**. We released the human alignment experiments and the knowledge distribution (appendix K) to claim the rigor of analysis (Dataset Subset & Labeling).
  - **W3**. We explained the deduction from the analysis via comparing our experiments results with PMET, claiming that the understanding is central to ACE’s design and is empirically supported by both our and prior work.
  - **W4**. We additionally clarified the Conflict Observation in deep FFNs, based on the references, and re-clarified our observation in section 4.3 : *Lower FFN neurons activate upper FFN neurons, while both of them enhance the probability of the final prediction.* We also mentioned the masking experiments in appendix L.3 to enhance this insight.
  - **W5**. We use math derivation beyond references to claim Reasons of importance score framework and Editing Level.
  - **W6**. We supplemented the Generalizability Evaluation experiments.

ACE Authors

---

### Meta-Review · Area_Chair_urVL · 2026-01-05

**Summary:**

This submission proposes ACE, a mechanistically motivated knowledge-editing framework for multi-hop factual recall, built around a “query–value (Q–V) pathway” hypothesis and a two-stage edit (value then query) to improve propagation of edited facts through multi-hop chains.

Reviewers’ initial concerns centered on (i) evaluation validity (prompt leakage/robustness, subset construction, labeling/human validation), (ii) coverage of baselines/generality (missing strong recent KE baselines; reliance on a single multi-hop benchmark), and (iii) soundness of the analysis (attribution assumptions, causal evidence, statistical rigor, and practicality/efficiency).

The rebuttal and ensuing discussion substantially strengthened the empirical and methodological case, leading to score increases from three reviewers and a maintained high score from the strongest reviewer, which collectively support a poster acceptance recommendation.

**Reviewer Concerns:**

Concerns substantially addressed in the rebuttal/discussion.

Prompt leakage/robustness: Authors added prompt-setting ablations (e.g., zero-shot/one-shot/OOD few-shot) to mitigate “teaching to the test” concerns.

Practicality and efficiency: Authors clarified the attribution profiling as largely amortizable and added latency analysis; the reviewer aVHM explicitly acknowledged improved clarity and practicality.

Baseline completeness / SOTA comparison: Authors added comparisons to AlphaEdit and showed ACE-style gains when paired with a stronger backbone.

Mechanistic evidence and robustness checks (UH9o’s questions): Added intermediate-tracing style evidence, rank-stability across prompt settings, and counterfactual/locality-style evaluations; Reviewer UH9o found these largely satisfactory and kept their strong score.

Concerns that remain partially outstanding

Breadth of multi-hop evaluation: Even with added sanity checks on general benchmarks, the multi-hop KE claims remain anchored primarily on MQuAKE-style evaluation; reviewers noted limited dataset coverage as a lingering bottleneck.

Attribution methodology vs simpler heuristics / statistical rigor: Reviewer 8jSo still requested clearer evidence that the attribution-based ranking meaningfully outperforms simpler layer-selection heuristics and asked for stronger uncertainty/statistical treatment, although they increased the score.

Instance-level vs dataset-level editing framing: Reviewer pPXV continued to view the instance-vs-dataset-level editing setup and dataset availability as important limitations, despite acknowledging many clarifications.

**Reviewer Scores:**

Reviewer aVHM: 4 to 6 (addressed prompt leakage, efficiency, and baseline coverage; explicitly indicated willingness to raise).

Reviewer UH9o: remains 8 (main mechanistic/robustness concerns primarily addressed; maintains strong endorsement).

Reviewer 8jSo: 4 to 6 (remaining requests on attribution-vs-heuristics and statistical rigor, but increased score).

Reviewer pPXV: 4 to 6 (acknowledged improved clarity/comparisons and updated score positively, while noting remaining limitations).

---

### Decision · Program_Chairs · 2026-01-26

Accept (Poster)